# A numerical sensitivity study of how permeability, porosity, geological structure, and hydraulic gradient control the lifetime of a geothermal reservoir

Johanna F. Bauer [1,2], Michael Krumbholz[3], Elco Luijendijk[1], David C. Tanner[4]

[1]Department of Structural Geology and Geodynamics, Georg August University of Göttingen, 37077 Göttingen, Germany
[2]now at Department of Rock Physics & Borehole Geophysics, Leibniz Institute for Applied Geophysics, 30655 Hanover, Germany
[3]Independent researcher
[4]Department of Seismic, Gravimetry, and Magnetics, Leibniz Institute for Applied Geophysics, 30655 Hanover, Germany

*Correspondence to*: Johanna F. Bauer (Johanna.Bauer@leibniz-liag.de)

**Abstract.** Geothermal energy is an important and sustainable resource that has more potential than is currently utilized. Whether or not a deep geothermal resource can be exploited, mostly depends on, besides temperature, the utilizable reservoir volume over time, which in turn largely depends on petrophysical parameters. We show, using over one thousand (n = 1027) 4-dimensional finite-element models of a simple geothermal doublet, that the lifetime of a reservoir is a complex function of its geological parameters, their heterogeneity, and the background hydraulic gradient (BHG). In our models, we test the effects of porosity, permeability, and BHG in an isotropic medium. Furthermore, we simulate the effect of permeability contrast and anisotropy induced by layering, fractures, and a fault. We quantify the lifetime of the reservoir by measuring the time to thermal breakthrough, i.e., how many years pass before the temperature of the produced fluid falls below the 100°C threshold. The results of our sensitivity study attest to the positive effect of high porosity; however, high permeability and BHG can combine to outperform the former. Particular configurations of all the parameters can cause either early thermal breakthrough or extreme longevity of the reservoir. For example, the presence of high permeability fractures, e.g., in a fault damage zone, can provide initially high yields, but it channels fluid flow and therefore dramatically restricts the exploitable reservoir volume. We demonstrate that the magnitude and orientation of the BHG, provided permeability is sufficiently high, are the prime parameters that affect the lifetime of a reservoir. Our numerical experiments show also that BHGs (low and high) can be outperformed by comparatively small variations in permeability contrast ($10^3$) and fracture-induced permeability anisotropy ($10^1$) that thus strongly affect the performance of geothermal reservoirs.

## 1 Introduction

The amount of geothermal energy that can be extracted from a reservoir depends, to a first order, on reservoir temperature, permeability, and utilizable reservoir volume. While temperature is often well constrained, the latter two parameters are more difficult to predict (e.g., Bauer et al., 2017; Bauer, 2018; Kushnir et al., 2018; Laubach et al., 2009; Seeburger and Zoback,

1982). The most important parameters recognised in the literature are porosity and permeability (e.g., Agemar et al., 2014; Moeck, 2014; Tiab and Donaldson, 2004). They are often highly heterogeneous because of layering, localized fracturing, and diagenesis (e.g., Aragón-Aguilar et al., 2017; De Marsily, 1986; Manning and Ingebritsen, 1999; Zhang, 2013). The vast majority of hydrothermal systems can be considered dual-porosity systems, where porosity is provided by both pore space and

fractures (Gringarten, 1984; Warren and Root, 1963).

In sedimentary geothermal reservoirs, the matrix porosity can exceed 30% and is often highly variable even on a small scale, e.g., within or between different sedimentary layers (e.g., Bär, 2012; Bauer et al., 2017; Heap at al., 2017; Zhang, 2013). Fracture porosity of sedimentary rocks, in contrast, is commonly significantly lower than matrix porosity and rarely exceeds 0.001% (e.g., Snow, 1968; van Golf-Racht, 1982). Nevertheless, permeability, and therefore flow rate in geothermal reservoirs,

is dominantly controlled by fractures (e.g., Bear, 1993; De Marsily, 1986; Hestir and Long, 1990; Nelson, 1985). Intrinsic fracture permeability is determined by the cube of the fracture aperture, while the permeability of fractured and porous reservoirs is related to the fracture system and the connection between fractures and pore space (De Marsily, 1986; Odling et al., 1999; Ran et al., 2014). Importantly, geometry, spatial distribution of fractures, and the resulting permeability anisotropy of a fracture system, are difficult to predict (e.g., Laubach et al., 2014; Ortega and Marrett, 2000; Watkins et al., 2018). A

number of deep geothermal projects in southern Germany (e.g., Trebur (Erdwerk), Mauerstetten (iTG), and Geretsried (iTG)) were unsuccessful because they failed to predict the hydraulic properties of the fracture system(s). In addition, there have been cases where, after a successful initial phase, the temperature of the production fluid dropped unexpectedly. Several of these cases were observed in faulted reservoirs, where production and injection wells were placed within a highly-permeable fault damage zone (e.g., Bödvarsson and Tsang, 1982; Diaz et al., 2016; Horne, 1982a,b; MacDonald et al., 1992; Ocampo et al.,

1998; Parini et al., 1996; Tenma et al., 2008). These fault zone permeabilities are often highly variable, i.e., they have been reported to reach from about $10^{-20}$ to $10^{-11}$ m$^2$, with the lowest values typically found in the fault core (Evans et al., 1997; Lopez and Smith, 1996; Shipton et. al., 2002). In addition, deep-seated, highly permeable faults are of interest for exploration because they can constitute positive thermal anomalies (Sanjuan et al., 2014; Vidal and Genter, 2018).

Geothermal reservoirs are also affected by other factors, such as the background hydraulic gradient (BHG) and its interaction

with the artificial flow field caused by the production. The BHG is, to our best knowledge, often not considered in model studies of geothermal systems, even though it may strongly affect the reservoir's lifetime (Bense et al., 2013; Hochstein, 1988; Moeck, 2014). Hydraulic gradients generated by groundwater recharge and discharge at the land surface average 1% (10 mm m$^{-1}$; Fan et al., 2013, Gleeson et al., 2016). These gradients dissipate at depth, especially in systems that contain low-permeable units that overlie geothermal reservoirs. However, lateral gradients can also be generated by a multitude of other processes,

including sediment compaction, clay mineral diagenesis, and buoyancy caused by changes in temperature or salinity (Bachu, 1995; Ingebritsen et al., 2006). These are often an order of magnitude lower than the driving force generated by recharge and discharge at the land surface, but have been shown to affect groundwater flow in a diverse range of geological settings (Garven 1995, Ingebritsen et al. 2006).

Here, we present a non-site specific sensitivity analysis of a 4-dimensional finite-element model of fluid and heat flow in a reservoir that comprises more than a thousand individual model runs. The objective of our study is to quantify the effects of various parameters on the temperature development of a geothermal reservoir and to quantify to what extent these parameters should be known to allow for reliable estimates on the lifetime of a geothermal reservoir. In addition, we evaluate under which circumstances a closed geothermal system can be achieved, i.e., when the injection- and production well doublet is hydraulically connected.

To achieve these objectives, we examine the importance of porosity, permeability, and permeability anisotropy on reservoir lifetime. We systematically test the effects of all these parameters for homogenous, layered, fractured, and faulted reservoirs. Furthermore, we apply BHGs of different magnitude and orientation to each of the different reservoir configurations.

The values for porosity and permeability that were included covered a range that is considered desirable for geothermal reservoir, but also included values that lie above and below these values. Specifically, we used permeabilities of $10^{-15}$, $10^{-13}$, and $10^{-11}$ $m^2$ and porosities of 3, 14, and 25%. With these values, we cover a large range of lithologies that could host geothermal reservoirs, i.e., fractured igneous and metamorphic rocks, as well as densely and less densely fractured sandstones or limestones (Bear, 1972; Freeze and Cherry, 1979; Lee and Farmer, 1993; Moeck, 2014). The one-at-a-time sensitivity analysis resulted in some parameter combinations (e.g., of porosity and permeability) that, while uncommon in nature, nevertheless help to identify the impact of individual parameters. At the same time, other parameters such as temperature gradient, production- and injection rate were kept constant to avoid exponential growth of the number of modelled parameter combinations, also known as the "curse of dimensionality" (Bellman, 2003).

## 2 Methods

We simulated fluid- and heat flow for a geothermal doublet, with one injection- and one production well, over a time span of 200 years. The model results quantify the effect of different geological parameters on the lifetime of a geothermal reservoir, i.e., the time during which the temperature of the produced fluid is above a critical value. As benchmark for the reservoir's performance, we chose to record the time before the production temperature reaches 100°C. This is because 100°C is often taken as the minimum temperature that allows the generation of electrical energy with binary cycles (e.g., Bhatia, 2014; Buness et al., 2010; Erec, 2004; Huenges, 2010; Mergner et al., 2012). We performed a series of over one thousand (n = 1027) model experiments, during which we systematically varied the values of parameters considered critical (e.g., Agemar et al., 2014; DiPippo, 2005; Moeck, 2014; Tiab and Donaldson, 2004), such as porosity and permeability, within ranges typical for e.g., sandstones, limestones, igneous, and metamorphic rocks (Bear, 1972; Freeze and Cherry, 1979; Lee and Farmer, 1993; Moeck, 2014) and conditions desirable for geothermal reservoirs (e.g., Agemar et al., 2014; Stober et al., 2017). In addition, we increased the complexity of the geological structure and permeability distribution by including sedimentary layering, permeability anisotropy induced by fracture networks, and a fault zone. Furthermore, the performance of all 1027 models was tested under different BHGs.

## 2.1 Numerical model

The numerical model experiments were carried out using finite-element modelling (FEM) with the COMSOL Multiphysics®5.0 software package, including the sub-surface flow module for fluid flow in porous media and for heat flow (COMSOL Multiphysics®). Fluid and heat flow were modelled by solving the following equations:


$$(\rho C_p)_{eq} \frac{\partial T}{\partial t} + \rho C_p q \cdot \nabla T = \nabla \cdot (k_{eq} \nabla T) + Q \tag{1}$$

$$(\rho C_p)_{eq} = \theta_s (\rho c_p)_s + (1 - \theta_s)(\rho c_p)_f \quad [\text{J kg}^{-1}\text{ K}^{-1}] \tag{2}$$

$$k_{eq} = \theta_s k_s + (1 - \theta_s) k_f \quad [\text{W m}^{-1}\text{ K}^{-1}] \tag{3}$$

Equation (1, COMSOL, 2017) states that change in temperature ($\nabla T$) at one point in the model is caused by conductive and advective processes (left-hand side) or due to a heat source or sink ($Q$, right-hand side). The effective volumetric heat capacity

$(\rho C_p)_{eq}$ (Eq. (2), COMSOL, 2017) and the effective thermal conductivity $k_{eq}$ (Eq. (3), COMSOL, 2017) used in this equation, represents the equalised value between the rock matrix and the fluid (1 – porosity [θ]).

The velocity field, $q$, of the advective term in Eq. (1) was implemented by adding the flow field, as described by Darcy's Law (Eqs. (4) & (5), COMSOL, 2018), which states that the fluid flow direction is controlled by the hydraulic gradient:

$$Q_m = \frac{\partial}{\partial t}(\rho \theta_s) + \nabla \cdot (\rho q) \quad [\text{m}^3\text{ s}] \tag{4}$$

$$q = -\frac{\kappa}{\mu}(\nabla p + \rho g \nabla D) \quad [\text{m s}^{-1}] \tag{5}$$

where $c_p$ is the heat capacity, $k$ is thermal conductivity, $\theta$ is the porosity, $Q_m$ is the mass source, $\rho$ is the density, $\kappa$ is the permeability, $q$ is the fluid velocity field, $\mu$ is the fluid viscosity, $\nabla p$ is the pressure gradient vector, g is acceleration due to gravity, and $\nabla D$ is the unit vector over which gravity acts. The subscripts $s$ and $f$ denote solid and fluid, respectively.

## 2.2 Geometry of the model

The modelled volume measures 4000 x 4000 x 2300 m (length, width, height, respectively) and was placed at a depth from 1600 to 3900 m below the surface (Fig. 1a). The whole model domain can be initially considered a potential reservoir volume, i.e., our study investigates which parameters influence the volume that actually can be utilized as a reservoir. The domain was subdivided in tetrahedral elements with edge lengths that vary from 712 m at the boundaries to 1 m around the injection- and production wells. Since the minimum mesh size was 1 m, this was also the smallest possible well diameter. To adjust for the

unrealistic large diameter of the wells, i.e. to simulate a perforated production zone with a realistic surface area, we assigned the production zone a length of 20 m. Considering a standard well diameter of 6 5/8" (about 17 cm), this corresponds to an production zone of 118 m. Both wells are inclined (to the vertical) by 30° degrees to the west. The reason for this is to place both within the western damage zone of the fault in Scenario 5 (Fig. 1f). To keep the models comparable, this configuration

was used in all scenarios. The injection- and production wells are 1500 m apart, N–S aligned, with the production and injection

wells to the N and S, respectively (Fig. 1a). In one scenario, we investigated the effect of the well spacing by varying the

distance between the wells from 500 m to 2000 m in 250 m steps.

## 2.3 Temperature

We applied a linear geothermal gradient of 0.047°C m$^{-1}$, and a surface temperature of 0°C, which resulted in an initial

temperature of 150°C at the well depth of 3200 m, that allows for geothermal power generation. The initial temperature ranged

from 75°C at the top to 183°C at the bottom of the model domain. A linear geothermal gradient was chosen, because it is a

good first order approximation for temperatures determined by heat conduction.

The top and bottom boundaries were thermally isolated: i.e., heat flux through these boundaries was set to zero. This

approximation ignores the background geothermal heat flux. However, over the comparatively short model time (200 years),

a background heat flux of 65 mW m$^{-2}$ would only add 0.08 J per m$^3$ to the model domain, and therefore would not change

significantly the model results. The side boundaries are modelled as follows**:** In cases where fluid flow is inward directed, heat

can flow into the model, as defined by the temperature gradient. If, however, fluid flow is directed outwards, the heat flux at

the boundary is set to zero. Consequently, the model results are not affected by the size of the modelled volume. The only

restriction caused by the size is that the whole shape of the 100°C isotherm (HDI), i.e., the envelope of the reservoir volume

with temperatures lower than 100°C, can be examined in all cases.

## 2.4 Fluid flow, permeability, and porosity


The upper and lower model boundaries were closed to fluid flow. A BHG was simulated in the model, which was varied in

magnitude and direction in different model runs. The BHG was applied as a pressure gradient on the model boundaries from

different directions and are thus valid for the whole model domain (Fig. 1b).

We applied a specified flow rate of 75 l s$^{-1}$ that was distributed over a cylindrical body that represents the active part of the

injection- and production wells. The BHG and the artificial flow field introduced by injection and production wells can interact.

We decided to use a fixed flow rate in our models, because it warrants, in contrast to the use of a fixed draw-down pressure,

comparability of the models, because the amount of injected cold fluid is constant and thus achieves flow velocities that are

not a function of the bottomhole pressure. Second, a fixed flow rate allows to identify the effect of the tested petrophysical and

structural parameters by providing the necessary fluid flux, i.e. it avoids extremely low flow rates. A further effect is that the

relation between bottomhole pressure and BHG is only controlled by permeability.

The temperature of the reinjected fluid was set to 40°C. The density and viscosity of the fluid were assumed constant (Table

1), which means that fluid flow directly affects temperature, but changes in model temperature do not change fluid density and

cause density-driven fluid flow. This simplification avoids thermal convection and reduces computational time significantly

from about 500 min to about 6 min for each model (PC platform configuration: Intel Xeon E31225 with clock rate:3.1 GHz

and 8 GB RAM). In addition, thermal convection is unlikely to occur in sedimentary settings, because it requires thick

homogeneous and highly permeable formations, whereas the establishment of convection cells is efficiently hindered by thin low-permeability layers that are a common feature in most sedimentary rocks (Bjørlykke et al., 1988; Moeck, 2014). The exception may be thermal convection in large, steep, continuous fault zones (Simms and Garven, 2004), which we do not investigate here. Moreover, thermal convection generates fluid flux that is commonly lower than topography-driven flow (Garven, 1995), and are also lower than the flow regimes induced by the injection and production wells in the model domain. Permeability was implemented using the continuum approach, which is, for sufficiently large volumes, a reasonable approximation (e.g. Berkowitz et al., 1988). In the continuum approach, hydraulic properties are assigned to a replacement media which has the mean hydraulic properties of a given fracture system. In our study, the parameters porosity and permeability are not coupled, i.e., because we vary each parameter separately. Therefore, we do not consider the role of effective porosity. Lithostatic pore pressures affect only fluid flow, but not permeability or porosity. Porosity controls the heat capacity of a given volume. Since fracture porosity is typically not higher than 0.001 % (e.g., Snow, 1968; van Golf-Racht, 1982) its contribution to heat capacity it can be considered neglectable.

## 2.5 Scenarios

In the following, we define the basic model properties. Homogenous models do not include any internal structure; isotropic models do not contain fracture anisotropy. Four basic scenarios were investigated (Fig. 1c–f). Material properties used for all models are listed in Table 1.

In the first scenario (Fig. 1c), the reservoir is homogenous and isotropic. We evaluate the time to thermal breakthrough for all combinations of three porosity values ($\theta$ = 3, 14, and 25%) and three different permeabilities ($\kappa$ = $10^{-15}$, $10^{-13}$, and $10^{-11}$ m$^2$). For the combination of 14% porosity and permeability of $10^{-13}$ m$^2$, we tested the effect of the distance between injection- and production wells.

In the second scenario (Fig. 1d), we introduced five horizontal confining layers, each a 100 m thick at intervals of 300 m, into the model volume. The production- and injection wells were placed in a 300 m-thick reservoir. This scenario comprises three series with different reservoir permeabilities ($\kappa$ = $10^{-15}$, $10^{-13}$, and $10^{-11}$ m$^2$). For each of these series, we set the permeability of the horizontal confining layers to be 1 to 4 orders of magnitude lower than that of the reservoir. All units were assigned porosities of 14%.

In the third scenario (Fig. 1e), the model had a porosity of 14%, and a permeability of $10^{-13}$ m$^2$. We introduced vertical fracture anisotropy that strikes N–S, NE–SW, E–W, and SE–NW and has 1, 2, and 3 orders of magnitude higher fracture permeability compared to the other directions, in an otherwise homogenous media.

To all possible variations of different parameters in these three scenarios, we applied BHGs of 0, 1, 5, and 20 mm m$^{-1}$ and varied the BHG direction from 0° to 315° in 45° steps (Fig. 1b).

In the fourth scenario (Fig. 1f), we tested the effect of a N–S striking, 60° westward-dipping fault, which consists of up to three parts; a 7 m wide fault core and two 40 m wide damage zones. We placed both wells in the western damage zone. We assigned a porosity of 14% to the entire model domain. The permeability of the host model volume, representing the host rock, was set

at $10^{-13}$ m$^2$. In the first sub-scenario, the fault was modelled as a single structure, i.e., only as a damage zone, with a permeability increased by 2 orders of magnitude compared to the host rock. In the second sub-scenario, we simulated a fault that consists of two damage zones and a fault core. The permeability of the damage zones was set to be 2 orders of magnitude higher ($10^{-11}$ m$^2$) than the host rock ($10^{-13}$ m$^2$) and the permeability of the fault core was set to be 5 orders of magnitude lower ($10^{-18}$ m$^2$) than the host rock. Both sub-scenarios were modelled without and with fracture anisotropy within the damage zones. In the latter case, we introduced fracture anisotropy parallel to the fault surface, with permeability 1 order of magnitude higher ($10^{-10}$ m$^2$), compared to all other directions.

In this fourth scenario, the orientations of the BHGs were 0°, 90°, 180°, and 270°, with simulated magnitudes of 0, 1, 5, and 20 mm m$^{-1}$.

In total, we modelled 1027 experiments with increasing geological complexity. Note that since the range of permeabilities analysed was large, we kept other parameters, including the fluid injection rate, constant, to allow different models to be comparable.

The modelled time was 200 years. In several cases, the production temperatures did not reach the threshold in this time, and therefore in these cases the reported lifetimes are underestimated and we give the breakthrough time to be equal or greater than 200 years.

## 2.6 Presentation of results

Since in our sensitivity study, we tested multiple parameters, we decided to present the results of the different (sub) scenarios in multiple figures. In all cases, scatter plots are presented next to each other, which contain the same results. The y-axis always shows the time to thermal breakthrough, i.e., the time until the temperature of the produced fluid falls below the 100°C threshold. The different x-axes are used to portray the results of the different parameters, e.g., permeability, porosity, and direction of the BHG. The magnitude of the BHG is indicated by the colour of the dots. In consequence, the adjacent scatter plots must read as one to identify those parameter(s) that dominantly determine the lifetime of the reservoir.

## 2.7 Simplification of the model

With our approach, we aim to examine how different reservoir parameters and their interaction affect the lifetime of geothermal reservoirs. Thus, our study is not site specific, but rather investigates, using simplified models, which parameters should be known, and to which accuracy they need to be known for realistic site-specific scenarios. These simplified models consequently suppress site specific effects and concentrate on the parameters investigated.

To compare single parameters, other aspects of the model must be kept constant, even if this does not represent a real-world scenario. For instance, bottom-hole pressure, which, due to the fixed flow rate, depend solely on permeability, can exceed lithostatic pressure.

## 3 Results

### 3.1 Models of reservoirs with homogenous and isotropic structure

In Scenario 1, we explore the role of porosity, permeability, orientation and magnitude of the BHG on fluid flow and geothermal lifetime for a homogenous and isotropic reservoir volume (Fig. 1c). The times in which the HDI reaches the production well, range from a few years only, to a span of time that, in many cases, exceeds the modelled time limit (Fig. 2a–c).

Our results show that in the low permeability models ($10^{-15}$ m$^2$), the range of lifetimes observed is less than 40 years. For the model series with intermediate permeability ($10^{-13}$ m$^2$, commonly taken as the threshold for exploitation of a geothermal reservoir) the range is about 80 years, and in the high permeability model ($10^{-11}$ m$^2$), it is almost 200 years (Fig. 2a). This is because higher permeability allows the BHG to outperform the artificially introduced flow field between injection- and production wells (Fig. 2a, b).

The effect of the natural BHG on the modelled temperatures becomes apparent if the shape of the HDI is examined (Fig. 2d–f). The HDI is (sub)spherical in models with low and intermediate permeability and in models without BHG (Fig. 2d). In models in which high permeability is combined with a BHG, the HDI becomes ellipsoidal. In the latter cases, the HDI's ellipsoidal long axis is parallel to the BHG direction and its aspect ratio is controlled by permeability and the magnitude of the BHG (Fig. 2e, f). The consequences of the combination of high permeabilities with BHGs are, first, that the HDI encloses a reduced narrower volume (Fig. 2e–f) and, second, that as a consequence the chances of extreme cases occurring increases (in which the HDI reaches the production well either very quickly or never at all; Fig. 2b, e, f). In other words, the higher the permeability of a reservoir is, the more the development of the shape of the HDI is controlled by the BHG, while the probability of the HDI reaching the production well decreases. For instance, in models with permeabilities of $10^{-11}$ m$^2$, the HDI only reaches the production well, if the BHG is (sub)parallel to the artificial flow field. The expression of the influence of the BHG in terms of the development of the production temperature is shown in more detail in Figure 2g. Without a BHG, the resulting spherical HDI approaches the production well slowly and causes a steady and intermediate temperature drop, compared to the other two examples shown (Fig. 2g). The two contrasting BHGs in Figure 2g show either fast (e), or almost no decrease (f) in production temperature.

Our results show that the role of porosity is subordinate to the other parameters (Fig. 2c). However, porosity still contributes to differences in observed breakthrough times. In case of the high permeability combined with highest southward-directed BHG, the breakthrough times vary by five years. In the same model, with the magnitude of the BHG at only 1 mm m$^{-1}$, they vary by 20 years (Fig. 2a–c). In general, the influence of porosity on expected lifetimes appears to cease with higher permeability and unfavourably directed BHG.

In addition, in this scenario we tested the effect of the distance between production- and injection wells on time to thermal breakthrough. We used the model setup with $\kappa = 10^{-13}$ m² and $\theta = 14\%$, applied BHGs of different magnitudes and orientations, and increased the distance between the wells in increments of 250 m from 500 to 2000 m (Fig. 3). We observe that lifetime

and the range of lifetimes increase with distance. For a well distance of 500 m, the lifetime is approximately 10 years. For a well distance of 1500 m, lifetimes range from 140 to over 200 years. The reason for this is that the BHG gains influence with increasing well distance. According to the previous set of model experiments (Fig. 2), this effect will be stronger for models with higher permeabilities and vice versa. The results also show that the distance between the wells and the modelled lifetimes do not correlate in a linear manner, but that lifetime increases disproportionally quicker with increasing distance, when comparing models with the same BHG configurations. This is, in the first order, because the volume of the HDI grows with the cube, partly defined by the distance between the wells. However, when a BHG is applied, this correlation is further modified, since the shape of HDI tends to become ellipsoidal, i.e., elongated parallel to the BHG. Consequently, the volume of the HDI is affected, but also the chances of the HDI reaching the production well are reduced since the BHG controls the direction in which the HDI propagates and may hinder a thermal breakthrough.

### 3.2 Models of layered reservoirs

In Scenario 2, we investigate the role of permeability contrasts in layered reservoirs by carrying out three series of experiments with different permeabilities (Fig. 1d). Since Scenario 1 showed that porosity is of minor importance, we kept porosity constant and used the medium value for porosity from Scenario 1 (14%) in all Scenario 2 experiments. The permeabilities of the reservoir layers in the three series were assigned values of $10^{-15}$ m$^2$ (series 1; Fig. 4a, b), $10^{-13}$ m$^2$ (series 2; Fig. 4d, e, c, f, i, j), and $10^{-11}$ m$^2$ (series 3; Fig.4 g, h) and the confining layers were assigned permeabilities 1 to 4 orders of magnitude lower than that of the reservoir.

In the models in which the reservoir layers were assigned the lowest permeability ($10^{-15}$ m$^2$, Fig. 4a, b), the lifetimes depended solely on the permeability contrast between reservoir- and confining layers; the BHG is not important. A contrast of 1 order of magnitude has little to no effect on the time to thermal breakthrough, compared with a model experiment without confining layers, which is otherwise identical (Figs. 2, 4a, b). Permeability contrasts higher than 1 order of magnitude, however, reduce the utilizable volume and affect the (anticipated) lifetime. There is a threshold for the permeability contrast that lies between 2 and 3 orders of magnitude (Fig.4c, f, i). Above this threshold, the fluid exchange between the reservoir layers is efficiently suppressed, i.e., the confining layers become effective barriers and the shape of the HDI is flat (Fig. 4i), and the time to thermal breakthrough is reduced significantly to less than 20 years, independent of the configuration of the applied BHG (Fig. 4a, b). In the models with intermediate permeabilities ($10^{-13}$ m$^2$, Fig. 4d, e) assigned to the reservoir layers, the results show a similar pattern to the models with the low permeable reservoir layers (Fig. 4a, b), i.e., for permeability contrast higher than 2 orders of magnitude the utilizable volume is restricted to one reservoir layer (Fig. 4i). However, with increased permeability the BHG's magnitude and direction begin to influence the lifetime. The time to thermal breakthrough is wider spread; the spread increases with the value of the BHG. We observed the largest variation in time to thermal breakthrough, depending on the BHG, of about 70 years, for a permeability contrast of 2 orders of magnitude. For a permeability contrast of 1 order of magnitude, the range is almost 30 years, while for contrasts of 3 and 4 orders of magnitude it is only about 20 years (Fig. 4d). Similar to the low permeability model, the influence of the BHG is diminished and is only minor for permeability contrasts

higher than $10^2$ (Fig. 4a, b, d, e). In this case, even favourably oriented BHGs do not have the potential to improve significantly the reservoir lifetime (Fig. 4d, e).

In models with highly permeable reservoir layers ($10^{-11}$ m$^2$, Fig. 4g, h), we observe the same permeability contrast threshold of $10^2$ that hinders the HDI to expand in and across the confining layers, as for the models with the less permeable reservoir layers. However, this threshold is no longer the dominant control on the time to thermal breakthrough (Fig. 4g, h). This is because the high permeability allows the BHG to shortcut the restrictions caused by the confining layers. The BHG results in variation in lifetime for all permeability contrasts that range from less than 10 years to lifetimes that exceed the model time limit of 200 years. This wide spread of lifetimes is predominantly controlled by the orientation of the BHG with respect to the alignment of the wells. Even a low BHG of 1 mm m$^{-1}$, oriented in opposite direction to the flow induced by the injection- and production wells, is sufficient to outperform the artificial flow field and therefore can hinder thermal breakthrough (Fig. 4g, h). The effect of the BHGs orientation is, however, compensated to a small degree by the fact that the high permeability of the reservoir layers allows for a wider lateral spread of the HDI.

The development of the temperature field over time (Fig. 4j) is shown for three model runs with different permeability contrasts using the model with intermediate reservoir layer permeability (Fig. 4c, f, i). The temperature drop at the production well depends on the permeability contrast between reservoir and confining layers and is quicker with increasing permeability contrast. In the model runs presented in Figure 4j, temperatures stabilize at a final temperature of about 100°C.

**3.3 Models of reservoirs containing vertical fracture anisotropy**

In Scenario 3, we introduce permeability anisotropy. Permeability is increased by 1 to 3 orders of magnitude the in vertical plane, compared to other directions. This model scenario represents a vertically-fractured reservoir (Figs. 1e, 5). These models use the medium porosity ($\theta = 14\%$) and permeability values ($\kappa = 10^{-13}$ m²).

We observe, in this series of models, times to thermal breakthrough that range from less than 10 years to more than 200 years (Fig. 5a–c). This range, however, is restricted to models with N–S striking fracture anisotropy, i.e., when the wells are aligned parallel to the direction of high permeability. In the other cases, with the anisotropy oriented NE–SW, NW–SE or E–W, i.e., at an angle to the well configuration, the HDI does not reach the production well in 200 years (Fig. 5a). This effect occurs for fracture anisotropies of 1 order of magnitude, but at $10^2$ and higher the effect does not increase (Fig. 5b). According to the results shown in Fig. 5a, c the applied BHGs only have an influence on the time to thermal breakthrough when the direction of BHGs is approximately parallel to the well configuration and fracture anisotropy. In such cases, either a high lifetime of the reservoir can be expected if the BHG is directed from the production to the injection well (Fig. 5e, h), or a very short lifetime for the opposite case (Fig. 5g, h). For the latter case, the value of anisotropy has a second-order control on the time to thermal breakthrough because it determines the lateral spread of the HDI and, in consequence, the utilizable reservoir volume.

Comparatively low permeability anisotropies of $10^1$ are sufficient to restrict fluid flux to the direction of highest permeability, which results in the HDI forming a narrow vertical volume parallel to the positive anisotropy (Fig. 5d–g). According to our models, this effect reaches saturation for an anisotropy of $10^2$ (Fig. 5b). This means that the horizontal extent of the HDI is

restricted and a hydraulic connection between the wells becomes more unlikely. The effect of permeability anisotropy is also stronger than that of the BHG. The BHG is only of importance when oriented parallel to permeability anisotropy and well alignment. In this case, its magnitude and whether it is directed away or towards the production well from the injection well controls the reservoir's lifetime.

### 3.4 Models of a faulted reservoir

In Scenario 4, we investigate the thermal development of a faulted geothermal reservoir for two sub-scenarios (Fig. 1f). In the first sub-scenario, the fault zone consists of a highly permeable damage zone ($10^{-11}$ m$^2$), while in the second sub-scenario, it consists of two symmetrical damage zones ($10^{-11}$ m$^2$) with a low permeable fault core ($10^{-18}$ m$^2$) at the centre. Both variations were modelled with and without fault parallel fracture anisotropy that increases the permeability by 1 order of magnitude in the damage zone(s) and for BHGs that are either parallel or normal to the strike of the fault. The permeability of the host rock

was in all cases set to $10^{-13}$ m$^2$.

For the first sub-scenario, the shape of the HDI is partly defined by the damage zone and expands predominantly in the surroundings of the injection well, taking an overall prolate shape (Fig. 6a–c). The temperature at the production well stabilises after about 10-20 years (Fig. 6d). The 100°C threshold is reached, for the strongest southward-oriented BHG, after about five years, while it is not reached in the modelled time for the highest northwards-oriented BHG. In case of the BHG normal to the

strike of the fault, the HDI's shape is comparable with that for the fault-parallel BHG (Fig. 6e, f), but the temperature development in this setup shows almost no difference with the different magnitudes of the BHG or with BHGs oriented east or west (Fig. 6g). The temperature falls within 15 to 20 years below the 100°C threshold and stabilizes at this point. In general, we observe the fault causes a channelling effect.

When modelling cases with a fault-parallel fracture anisotropy in the damage zone (Fig. 7), for north, east, south and west-

directed BHGs, the channelling effect increases and the volume of the HDI, independent of the orientation of the BHG, is restricted to the fault zone (Fig. 7a–c, e–f). This channelling effect leads, in the case of fault-parallel BHGs, to a wide spread of time-dependant temperature behaviour (Fig. 7d). For example, in case of the highest northwards-directed BHG, almost no temperature reduction at the production well is observed. For the highest southward-directed BHG, the production temperature falls below the 100°C threshold after about 2 years (Fig. 7d). All temperatures stabilize after about 15 years.

For the east- and westwards-directed BHGs, differences in the temperature development exist (Fig. 7g), in contrast to similar models without anisotropy (Fig. 6g). In these cases, BHGs oriented in the dip direction of the fault show a faster temperature drop compared to BHGs opposed to the dip direction.

In the second sub-scenario, we increased the permeability of the damage zone compared to the host by 2 orders of magnitude to $10^{-11}$ m$^2$ and introduced a fault core with a permeability of $10^{-18}$ m$^2$.

In the case without fault-parallel fracture anisotropy, the shapes of the HDIs are restricted on the eastern side by the impermeable fault core and extrude on the western side into the host rock (Fig. 8a–c, e–f). This bulge is concentrated around the injection well; otherwise, the channelling effect leads to a HDI that is largely defined by the high permeable part of the

western damage zone, in which both wells are placed. These observations are independent of the orientation of the BHG, i.e., parallel or normal to the strike of the fault.

The temperature development follows the same pattern as in the previous model without fault core (Figs. 6, 8), and it is independent of the orientation and magnitude of the BHG. It is, however, slightly quicker; the range between the different north- and southward-directed BHGs is smaller and does not exist between the east- and westwards-directed BHGs. Only for the highest northwards-oriented BHG, does the temperature stays above the 100°C threshold over the modelled time, while for the strongest southward-oriented BHG, it is reached after about 8 years. After the sharp initial temperature drop at the production well, a very slow further temperature reduction is observed for the rest of the modelled time (Fig.8 d, g).

Introducing fault-parallel permeability anisotropy into the damage zone (Fig. 9), has the effect that the HDI becomes restricted almost entirely to the western damage zone, i.e., the part in which the wells are situated. In the case in which a BHG is not applied (Fig. 9a), the HDI utilizes a large part of the damage zone, restricted to the south by the production well. Northwards-directed BHGs (Fig. 9b) produce a "fin"-like pattern within the modelled domain, i.e., the HDI extends from the production towards the injection well along the damage zone. High southward-directed BHGs (Fig. 9c), in contrast, result in a small oblate HDI confined between production and injection wells. In terms of the temperature development (Fig. 9d), these patterns reflect different behaviours. The highest northwards-directed BHG almost entirely hinders a temperature drop at the production well. On the contrary, the highest southwards-oriented BHG causes the production temperature to fall below the 100°C threshold within 5 years. Stable temperatures at the production well are reached after about 10 years, independent of the BHG properties (Fig. 9).

When testing this setup for west- and eastward-directed BHGs, the production temperatures reached the 100°C limit very quickly and fell below 100°C, with the exception of the highest eastward-oriented BHG, within about 10 years (Fig. 9g). The shape of the HDI is restricted to the width of the western damage zone. However, the N–S extent of the HDI varies in the vertical, along the fault plane. In case of westwards-oriented BHG, i.e., with the BHG direction in line with the dip direction of the fault, the N–S extension is wider at deeper levels and vice versa for eastwards-directed BHGs (Fig. 9e–f). This observation reflects also in the potential lifetimes of the reservoirs (Fig. 9g), which show a wider spread with respect to the temperature development with comparable models without increased fault-parallel fracture anisotropy (Fig. 8). Here the eastwards directed BHGs result in an improved lifetime that, for the highest eastward BHG, allow the production temperature to stay above the critical 100°C level.

**4 Discussion**

Petrophysical properties, e.g., porosity and permeability, control the quality of a geothermal reservoir. However, these properties may vary significantly within a given volume. For instance, permeability is frequently observed to vary over several orders of magnitude (Heap et al., 2017; Kushnir, 2018; Manning and Ingebritsen, 1999); porosity may vary by 10 percent or more (e.g., Farrell et al., 2014; Heap et al., 2017; Kushnir, 2018; Zhang, 2013). The variability of these (Freudenthal, 1968;

Krumbholz et al., 2014a) and other petrophysical parameters (Alava et al., 2009; Lobo-Guerrero and Vallejo, 2006) increases with scale. In addition, their heterogeneous distribution and property values are often anisotropic in terms of orientation, e.g., permeability caused by fractures often has a preferred orientation (e.g., Laubach et al., 2004; Marrett et al., 2007; Nelson, 1985; Watkins et al., 2018). To make a reliable prediction of reservoir quality, it must first be determined to which accuracy these parameters must be known (Bauer et al., 2017; Bauer, 2018). The ranges of the parameter values we use in our modelling

experiment for porosity, permeability, and fracture anisotropy are, for example, typical for sandstones in the Upper Rhine Graben (Germany; Fig. 10). Even in this comparatively small area, the porosity is reported to cover a range from close to zero to more than 25% (Bär, 2012; Bauer et al., 2017; Jodocy and Stober, 2011; Fig. 10a). The permeabilities determined for the same reservoir rocks range from about $10^{-18}$ m$^2$ to $10^{-11}$m$^2$, more than 7 orders of magnitude (Bär, 2012; Bauer et al., 2017; Jodocy and Stober, 2009, 2011; Stober and Bucher, 2014; Fig. 10b). Furthermore, permeability anisotropy in fracture networks

can reach several orders of magnitude (e.g., Bense and Person, 2006; Caine and Forster, 1999; Jourde et al., 2002; Watkins et al., 2018).

With our simplified models, we systematically investigate the effects and the interplay of these important parameters on reservoir performance. In addition, we have taken into account the effects of the BHG. Our experiments do not aim to describe a specific reservoir model, but rather to identify prime parameters in terms of geothermal reservoir performance and the

400 accuracy to which they should be known. It was therefore important to keep some model parameters constant. Thus, the use of model parameters was a trade-off between values that lie in a realistic range, and values that allow comparison of the model results in the modelled time span. For this reason, we decided to assign a relatively low temperature of 40°C to the injected fluid and to use a fixed flow rate of 75 l s$^{-1}$. This fixed flow rate used for the three different permeability scenarios leads to bottomhole pressures that vary from approximately 3800 MPa at $k = 10^{-15}$ m$^2$, over approximately 180 MPa at $k = 10^{-13}$ m$^2$,

to 30 MPa at $k = 10^{-11}$ m$^2$. For low and intermediate permeabilities, these values exceed the lithostatic pressure at injection depth, which is about 80 MPa. In site-specific models, fluid pressures could be kept at sub-lithostatic values by adjusting the length of the well over which injection takes place, by reducing the flow rate, or by injecting the fluid at a greater depth.

## 4.1 Porosity

Rock volumes constitute in many cases, especially in sedimentary rocks, dual porosity systems (Gringarten, 1984; Warren and

410 Root, 1963) in which pore space and fractures are fluid-filled. Since pore fluid has a higher heat capacity than rock (rock ~700–1100 J kg$^{-1}$ K$^{-1}$ (Schärli and Rybach, 2001; Stober et al., 2017); water = 4184 J kg$^{-1}$ K$^{-1}$), high porosity has a positive effect on the heat capacity of the reservoir. Our Scenario 1 concurs with this; high porosities have a positive effect on the lifetime of a geothermal reservoir. The size of this positive effect, however, varies largely according to the other parameters. For instance, in Scenario 1, the time that passes before the HDI reaches the production well ranges from decades to only a few years, in

models in which porosity is the only varied parameter (Fig. 2c). Especially in the case of high permeability, i.e., exceeding a value of $10^{-13}$ m$^2$, often taken as the threshold for economical exploitation of a geothermal reservoir (Agemar et al., 2014; Stober et al., 2017), the effect of porosity on the lifetime of the reservoir is dramatically reduced.

## 4.2 Permeability and hydraulic background gradient

The reason for these variations, and the predominantly subordinate role of porosity, is according to Scenario 1, provided by the interplay between permeability and the BHG (Fig. 2a–c). If permeability is high enough, the BHG controls shape, volume and propagation direction of the HDI during heat extraction. For permeabilities as low as $10^{-15}$ m$^2$, the BHG is outperformed by the artificial flow field caused by the very high bottomhole pressure. The result of this is that the HDI takes on a spherical shape and thus maximizes the exploited volume. This in consequence allows to maximize the "gain" from high porosity, i.e., high heat capacities, in a large rock volume. If, however, the permeabilities lie in a range suitable for geothermal energy exploitation, i.e., $10^{-13}$ m² or higher, the necessary bottomhole pressure is decreased. Consequently, the artificial flow field is weaker and can be outperformed by the BHG. The higher the permeability, the larger the effect of the BHG, which ultimately causes an ellipsoidal HDI in an isotropic volume. In this case, direction and magnitude of the BHG thus become the controlling factors whether or not the HDI reaches the production well. Thus, we show that the BHG, which is rarely taken into account when it comes to predict the potential of geothermal reservoirs, is essential in any exploration strategy.

The importance of these findings is shown by cases, such as the geothermal Hatchōbaru field in Japan (Bödvarsson and Tsang, 1982; DiPippo, 2005; Horne, 1982b). There, the injection- and the production wells were placed within a fault zone, resulting in good hydraulic connection. Moreover, the location of the wells was chosen in a way that the natural hydraulic background was oriented from the production to the injection well, to avoid early thermal breakthrough. Nevertheless, the artificially-introduced flow field was strong enough to outperform the BHG and the early drop in production temperature shows how fragile such high permeable systems can be.

## 4.3 Permeability contrast

(Sub)horizontal permeability contrasts can be caused by layering in sedimentary rocks and can span several orders of magnitude (Zhang, 2013), even though these sealing properties are altered or reduced by barren fractures. One example are clay layers that typically have permeabilities in the range of $10^{-17}$ to $10^{-23}$ m$^2$ (Neuzil, 1994) and therefore restrict fluid flow across them, independent of their thickness (e.g., Bjørlykke et al., 1988; Moeck, 2014). However, permeability contrast can also be caused by diagenesis, i.e., some sedimentary layers are more cemented and thus have a lower matrix permeability. According to our models, in a rock volume with an overall permeability of $10^{-13}$ m$^2$, permeability contrasts between two layers, as low as 1 order of magnitude, will start to affect significantly the shape of the HDI (Fig. 4c). Confining layers that have a permeability between 2 and 3 orders of magnitude lower than the reservoir layer do not allow the HDI to expand across them, leading to a exploitable reservoir volume that is layer-parallel, and consequently, has a drastically reduced volume (Fig. 4f, i). Notably, in the low and intermediate permeable models, where permeability contrasts are higher than 2 order of magnitude, high bottomhole pressures do not allow the BHG to affect the system, i.e. none of the tested BHG configurations could compensate for the small volume (Fig. 4a, d). Only in the high permeable case, i.e., where the reservoir layers have a

permeability of $10^{-11}$ m$^2$, could a favourably-oriented BHG push the cold reinjected fluid away from the production well, providing therefore a potentially longer reservoir lifetime.

## 4.4 Fracture anisotropy

The greater part of permeability in a sedimentary rock volume is typically provided by fractures and their networks (e.g., Bear, 1993; De Marsily, 1986; Hestir and Long; 1990; Nelson, 1985). Thus, a thorough understanding and prediction of this reservoir property constitutes a major requirement when planning a geothermal reservoir. While microcracks are commonly predictable, (e.g., Kranz, 1983; Krumbholz et al., 2014b; Vollbrecht et al., 1994), making assumptions about fractures at decimetre- and metre-scale, i.e., the fractures that principally control fluid flow, is difficult (e.g., Bauer et al., 2017; Bauer, 2018; Laubach et al., 2004). When considering the Cubic Law, a single fracture, if it is wide enough, can control the fluid flow in a reservoir (e.g., De Marsily, 1986; Nelson, 1985; Odling et al., 1999). In consequence, the prediction of fracture anisotropy at depth with the necessary accuracy is, at least, challenging and, at the most, impossible (e.g., Laubach et al., 2004; Ortega and Marrett, 2000; Watkins et al., 2018). Our models show that a fracture anisotropy of 1 order of magnitude (Fig. 5b) can channel the fluid flow so efficiently that the HDI´s propagation, in all other directions, is suppressed. If the anisotropy reaches just 2 orders of magnitude, which according to the literature (e.g., Bense and Person, 2006; Caine and Forster, 1999; Jourde et al., 2002; Watkins et al., 2018) must be considered to be a typical value, this suppression effect becomes saturated. In our models, even high BHGs cannot counteract this behaviour. According to our results, in this case, the BHG can only effects the reservoir's lifetime when the BHG and fracture anisotropy are in line with the wells (Fig. 5a). The consequences are twofold: First, even comparatively low fracture anisotropy can hinder the establishment of a closed hydrothermal system. Second, fracture anisotropy in the range of 1 order of magnitude, with respect to the bulk permeability, leads to either very short- or long-lived geothermal reservoirs, depending on the BHG properties and the orientation of fracture anisotropy (Fig. 5a, b, c).

## 4.5 Distance between wells

The distance between production and injection wells is the only parameter known to any accuracy. It is inherent that increasing distance between production and injection wells has a positive effect on the lifetime of a geothermal reservoir. However, precise site-specific estimation of this effect requires in-depth knowledge of highly heterogeneous parameters, such as permeability and porosity. Given that both wells need to be hydraulically connected, the distance has a disproportionally high impact (Fig. 3), since the volume, between the wells, grows cubically. Our models suggest that the achievable lifetime does not necessarily scale directly with the volume, because the HDI is the result of the complex interaction between permeability contrast, fracture anisotropy, and the BHG. Moreover, increasing the distance between production and injection well also reduces the chance of establishing a closed system.

## 4.6 Faulted reservoirs

Faults have been recently the focus of many studies of the economical exploitation of geothermal energy. The main reason for this is that fault damage zones promise significantly increased permeabilities (e.g., Bense et al., 2013; Caine et al., 1996; Caine and Forster, 1999; Sibson, 1977), and can provide positive temperature anomalies (Sanjuan et al., 2014; Vidal and Genter, 2018). The typical characteristics of fault zones thus increase the chance of high production rates of hot fluid, and this is potentially further improved by fracture anisotropy within the damage zone, which is often (sub)parallel to the fault (e.g., Bense et al., 2013; Caine et al., 1996; Faulkner et al., 2010; Shipton and Cowie, 2003). However, a number of critical studies exists (e.g., Bakhsh et al., 2016; Bauer et al., 2017; Bauer, 2018; Biemans, 2014; Diaz et al., 2016; Loveless et al., 2014) that discuss the risk and difficulties of exploring and exploiting fault zones as geothermal reservoirs. The two main concerns reported are that first a fault's architecture at reservoir depth is, due to the heterogeneous nature of rocks, and, in particular, that of the faults, difficult to predict, i.e., exploration risk increases with complexity and heterogeneity of the envisaged reservoir (e.g., Bauer 2018; Bauer et al., 2018; Loveless et al., 2014). The second concern is directly correlated to the expected high permeability that makes a fault a prime target in geothermics. This is because localized high permeabilities lead to channelling effects, i.e., the geothermal reservoir potentially becomes restricted to the fault zone (e.g., Biemans, 2014; Bakhsh et al., 2016; Moeck, 2014). Thus, the exploitation of fault zones constitutes a trade-off between high permeability and reduced reservoir volumes. Our simplified models support these findings and show that faults, with damage zones that constitute positive permeability contrasts of just 2 orders of magnitude, already exhibit these channelling effects (Fig. 6). In these cases, the shape of the HDI is almost entirely described by the extent of the damage zone. In most investigated cases, this limitation of reservoir volume quickly leads to a sharp drop in production temperature, i.e., in most configurations, the temperature falls below the 100°C threshold within in a few years (Figs. 6, 8). This fast depletion of such a fault-related reservoir is further accelerated if the hydraulic connection between production and injection wells is improved by fracture anisotropy, which is often parallel to the fault (e.g., Bense et al., 2013; Caine et al., 1996; Faulkner et al., 2010; Shipton and Cowie, 2003; Figs. 7 and 9).

However, the most promising configurations that allowed for longevity of fault-related reservoirs are those with a high, fault-parallel, BHG that leads directly from the production well to the injection well (e.g., Figs. 6b, d, 7b, d). Interestingly, for a BHG normal to the strike of the fault, we also observe different reservoir lifetimes as a consequence of differently shaped HDIs. We observed that, when the BHG is directed against the dip direction of a fault, the fault can be considered a more sustainable target for geothermal exploitation than a BHG oriented in the opposite direction (Figs. 7e, f, 8e, f). We argue that in the first case, the BHG works against gravity in the fault zone, which slows the propagation of the HDI down, while in the opposing case, the BHG is supported by gravity and the progression of the HDI, and, in consequence, the depletion of the reservoir, is accelerated.

However, our models are still simplistic and the next steps of investigation should study how the channelling effect is altered as the permeability contrast between the fault zone and host rock increases, i.e., whether there is a transition zone of significant width between the low and high permeable zones.

Another question that needs to be addressed when exploring fault-hosted or fractured reservoirs - is replenishment due to convective heat transport along the fault zone, as reported for numerous thermal anomalies in the upper Rhine Graben (e.g., Sanjuan et al., 2014; Vidal and Genter, 2018). This point, often taken as an argument in favour of fault-related geothermal reservoirs, is not part of our study. Nevertheless, our results can help to constrain the observed channelling effects and therefore how high the replenishment rates in such reservoirs must be in order to counteract this.

## 5 Conclusions

We use over one thousand numerical experiments to systematically investigate the effect of a number of parameters, e.g., permeability, porosity, on the potential lifetime of a geothermal reservoir. We varied the reservoir parameters within realistic ranges and applied BHGs of different orientations and magnitudes. From the results of our numerical sensitivity study, we conclude:

1.  That permeability, permeability heterogeneity, and fracture anisotropy together with the BHG are the critical parameters that affect the lifetime of a geothermal reservoir.

2.  While high permeability is an asset for the exploitation of geothermal energy, our experiments demonstrate that it also comes with a risk. One the one hand, high permeabilities are needed to generate sufficient flow rates. On the other hand, high permeabilities in general, and in particular localized high permeability, such as in layered sedimentary systems, in fractures, and especially in fault damage zones, channel fluid flow and strongly restrict the size of the geothermal reservoir that can be utilized.

3.  Typically, geothermal energy production aims to establish a closed system. This is, according to our models, not trivial and depends strongly on permeability, permeability heterogeneity, the internal structure of the reservoir, and the BHG. This is especially true if the permeability is high ($>= 10^{-13}$ m$^2$), i.e., values that are desired usually for geothermal reservoirs.

4.  Our models imply that, in many cases, the positive effect of porosity on heat capacity and thus on the reservoir lifetime, is subsidiary, compared to the effects of permeability and BHG; the impact of porosity decreases as permeability increases.

5.  Due to the heterogeneous nature of rocks and fracture systems, it is, in general, difficult to predict the lifetime of a geothermal system. This holds especially true even if the required conditions for permeability are met or exceeded and a BHG exists, because the BHG determines the shape and propagation direction of the HDI. Therefore, the uncertainty in estimating the lifetime inevitably increase.

Our results show that parameters, such as permeability and the BHG, can have unforeseeable large effect on the lifetime of geothermal system. Thus, our findings provide an important step forward in judging which parameters must be known and to which degree they must be known to make site-specific models as reliable and accurate as possible in the future.

## Data availability

All data analysed in this study are presented in the manuscript.

## Author contribution

JFB and MK contributed equally to the manuscript. The model experiments were designed by JFB, MK, and EL. Results were discussed by all authors. MK, JFB, EL, and DCT wrote the manuscript. JFB, MK, and EL prepared figures. All authors commented, read, and approved the final manuscript.

## Competing interests

The authors of this manuscript declare that they do not have any conflict of interest.

## Acknowledgments

We thank our colleagues, Hans Ruppert, Klaus Wemmer, Jonas Kley, and Nicole Nolte for their support. We also thank Federico Rossetti, two anonymous reviewers, and Owen Callahan for their comments that helped improve our manuscript.

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

**Table 1: Material properties used for all models. The rock and fluid properties are oriented on Triassic Buntsandstein sandstone and a geothermal brine, respectively. Rock properties are adopted from Bär (2012), the fluid properties are typical for geothermal brine at 100°C (e.g., Birner et al., 2013; Stober et al., 2017).**

| parameter | symbol | value | unit |
|---|---|---|---|
| fluid properties | | | |
| density | $\rho$ | 1100 | kg m$^{-3}$ |
| heat capacity | $c_p$ | 4200 | J kg$^{1}$ K$^{-1}$ |
| thermal conductivity | $\kappa$ | 0.6 | W m$^{-1}$ K$^{-1}$ |
| viscosity | $\mu$ | 0.0002 8 | Pa s |
| rock properties | | | |
| density | $\rho$ | 2400 | kg m$^{-3}$ |
| heat capacity | $c_p$ | 670 | J kg$^{1}$ K$^{-1}$ |
| thermal conductivity | $k$ | 2.6 | W m$^{-1}$ K$^{-1}$ |

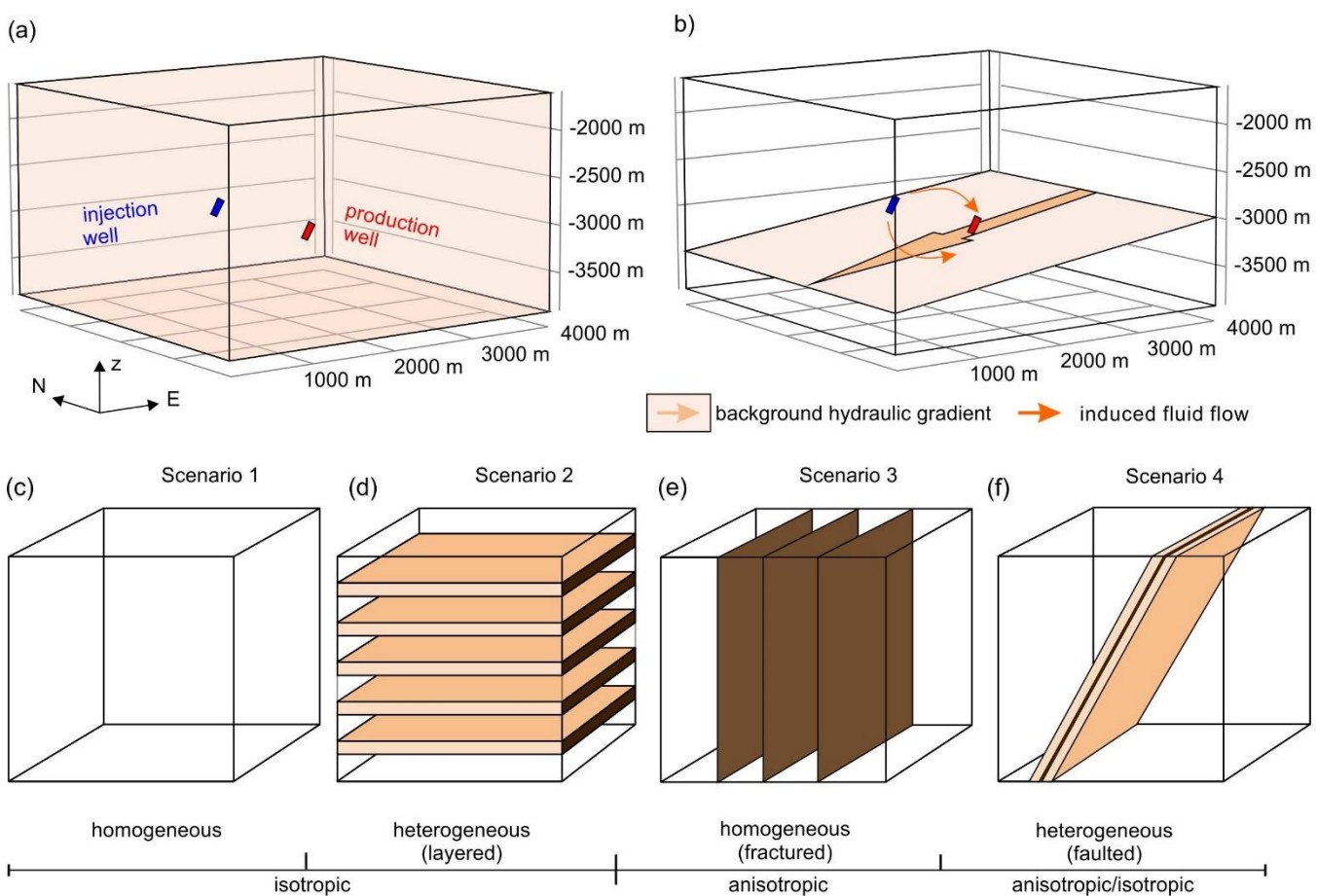

**Fig. 1. Model setup used in our study. (a) the rectangular cuboid model lies at a depth between 1600 and 3900 m and has side lengths of 4000 m. Injection- and production wells are 1500 m apart. (b) shows one example of an westward directed BHG. (c–f) show sketches of the different scenarios investigated, comprising of reservoirs that are (c) homogenous, (d) horizontally layered, (e) include vertical fracture anisotropy, and (f) a 60° west-dipping fault zone that consists of a damage zone, with and without a fault core, and with and without fracture anisotropy in the damage zone. Modified after Bauer (2018).**

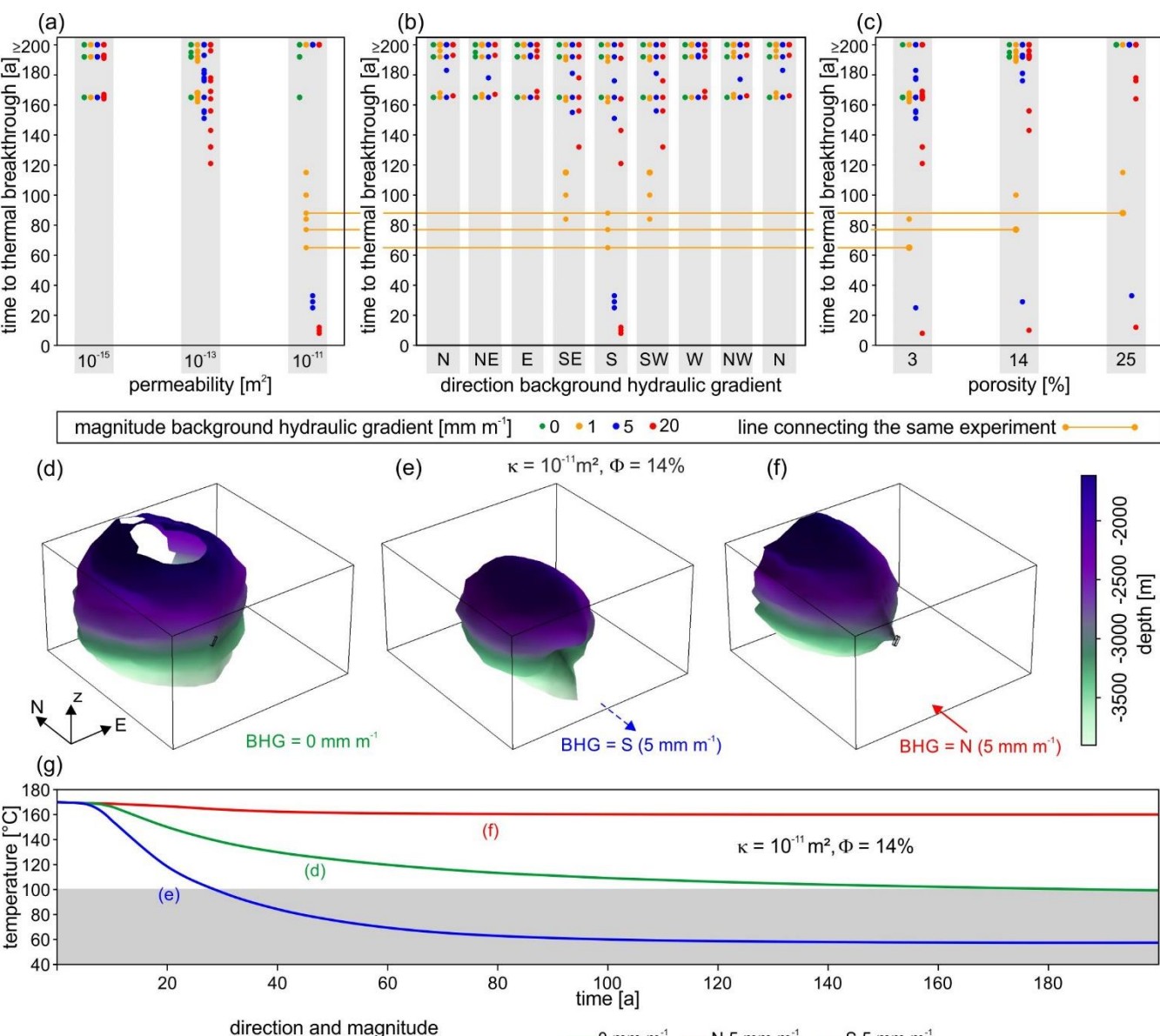

**Fig. 2: Parameter values against time to thermal breakthrough for model experiments with homogenous and isotropic structure (Scenario 1). Plots (a), (b), and (c) each contain the same results of 225 simulations, but are arranged according different x-axes. The results show that with increasing permeability, the BHG becomes the prime factor that determines time to thermal breakthrough, while the effect of porosity is minor. Plots (d), (e), and (f) show the shape of the HDI after 100 years for a model with a permeability of $10^{-11}$ m² and porosity of 14%, with no BHG (d), with a BHG of 5 mm m⁻¹ to the S (e), and with a BHG of 5 mm m⁻¹ to the N (f). At high permeabilities and magnitudes of the BHG, the shape of the HDI becomes ellipsoidal. Panel (g) shows the development of the production temperature over time for the models presented in (d–f).**

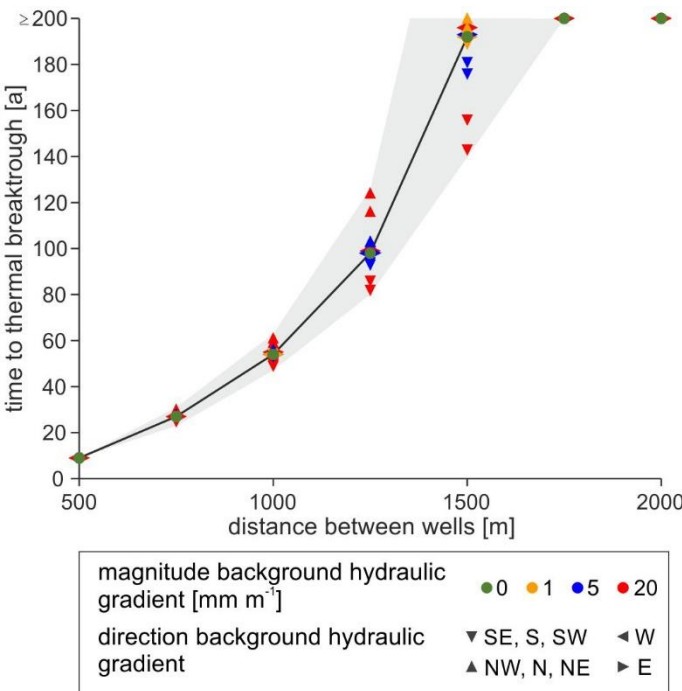

Fig. 3: Time to thermal breakthrough vs. distance between injection- and production well. The plot contains the results of 150 simulations. For models without BHG, the relationship of time to thermal breakthrough to distance is not linear. The increasing spread observed at the greater distance is due to the influence of the BHG, which increases with distance. The basic model setup is identical to the medium porosity and permeability model ($\kappa = 10^{-13}$ m$^2$ and $\theta = 14\%$). The black line connects the models with zero BHG, the grey-shaded area shows expected lifetimes for possible configurations of BHG orientation.

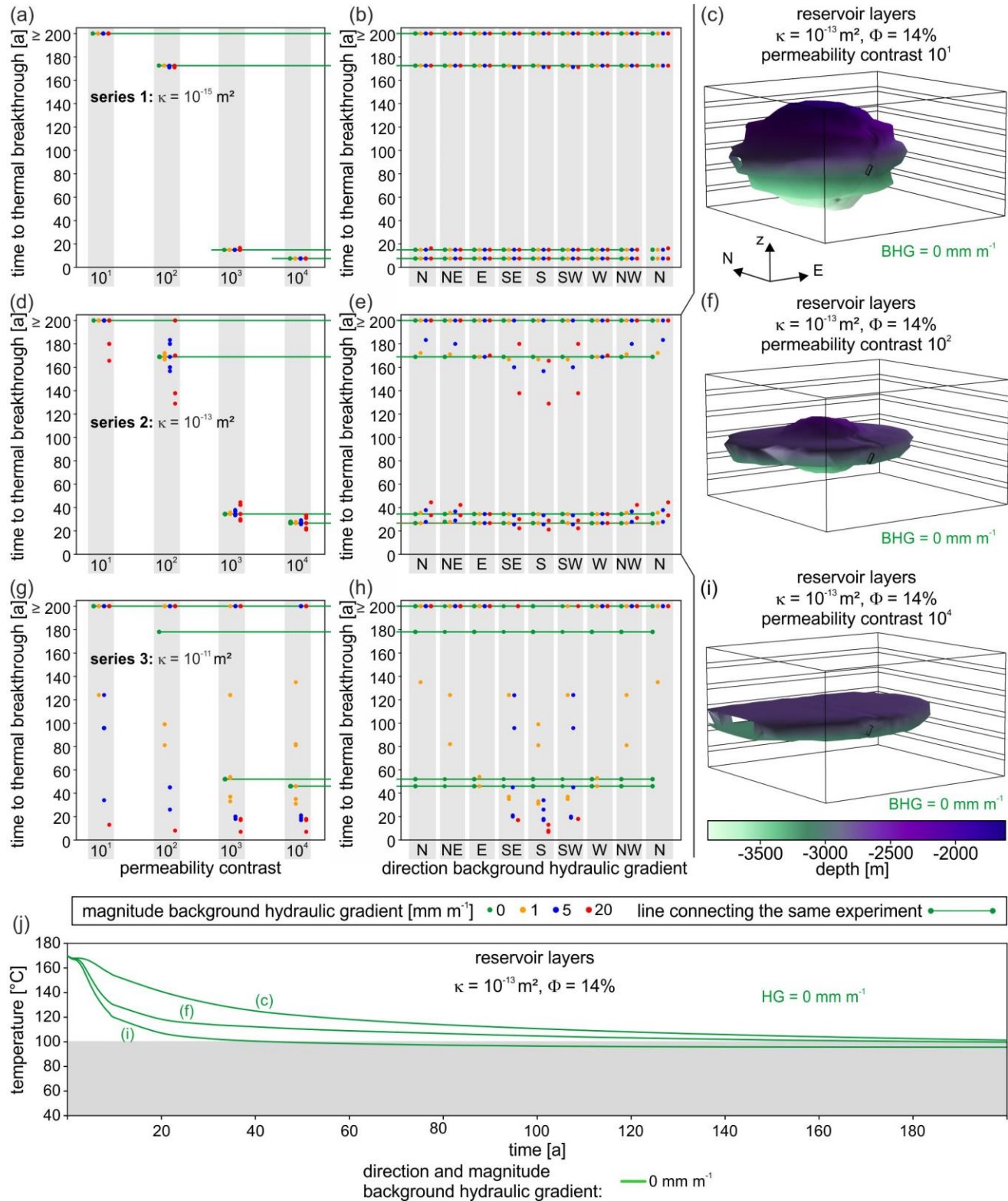

**Fig. 4: Parameter values against thermal breakthrough for model experiments with layered permeability contrasts (Scenario 2). Plots (a) to (b), (d) to (e), and (g) to (h) show the results of three permeability series. Each series contains the results of the same 100 simulations, but are arranged according different x-axes. The results show that a threshold exists for permeability contrasts between 2 and 3 orders of magnitude between reservoir- and confining layers above which fluid exchange is suppressed. The result is an increasingly flatter shape of the HDI, i.e., a reduction of utilizable reservoir volume, which only at high permeabilities can be compensated for by a favourably oriented BHG. In (c), (f), and (i), the shape of the HDI after 100 years is shown for the case of intermediate permeable reservoir layers (series 2), permeability contrasts of 2, 3, and 4 orders of magnitude and without applied BHG. Panel (j) shows the development of the production temperature over time for the models presented in (c), (f), and (i).**

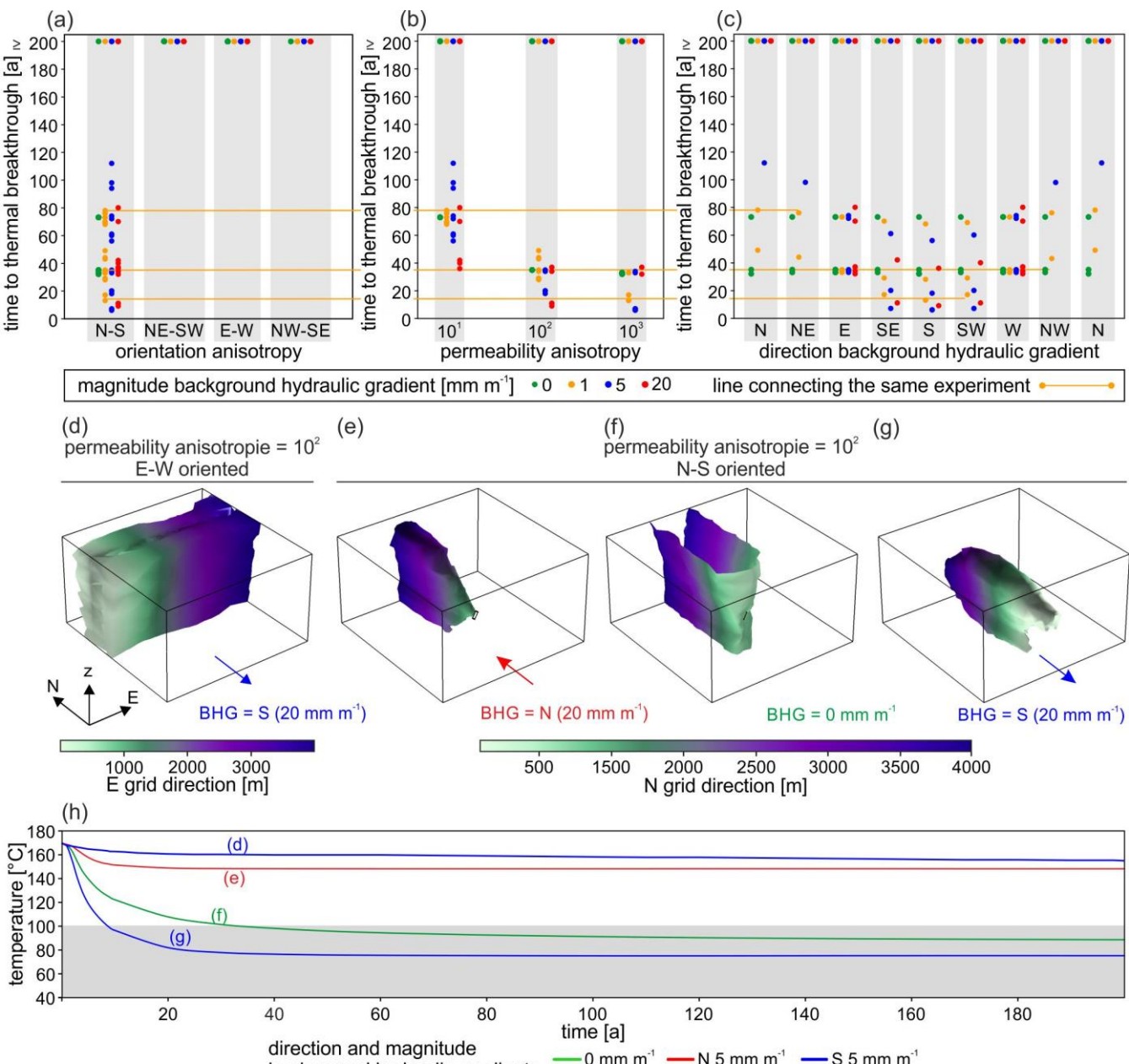

**Fig. 5: Parameter values against time to thermal breakthrough for model experiments with vertically oriented fracture anisotropy (Scenario 3). Plots (a), (b), and (c) each contain the same results of 300 simulations, but are arranged according different x-axes. The results show that permeability anisotropy largely controls time to thermal breakthrough and that the effect of anisotropy reaches its maximum at 10². (a) for cases with the anisotropy (sub)parallel to well configuration the BHGs direction is of importance and causes either a very short or long reservoir lifetime. (d), (e), (f), and (g) HDI after 100 years. The HDI forms a vertical volume parallel to the orientation of the anisotropy, independent of the direction of the BHG. Panel (h) shows the development of the production temperature over time for the models presented in (d–g).**

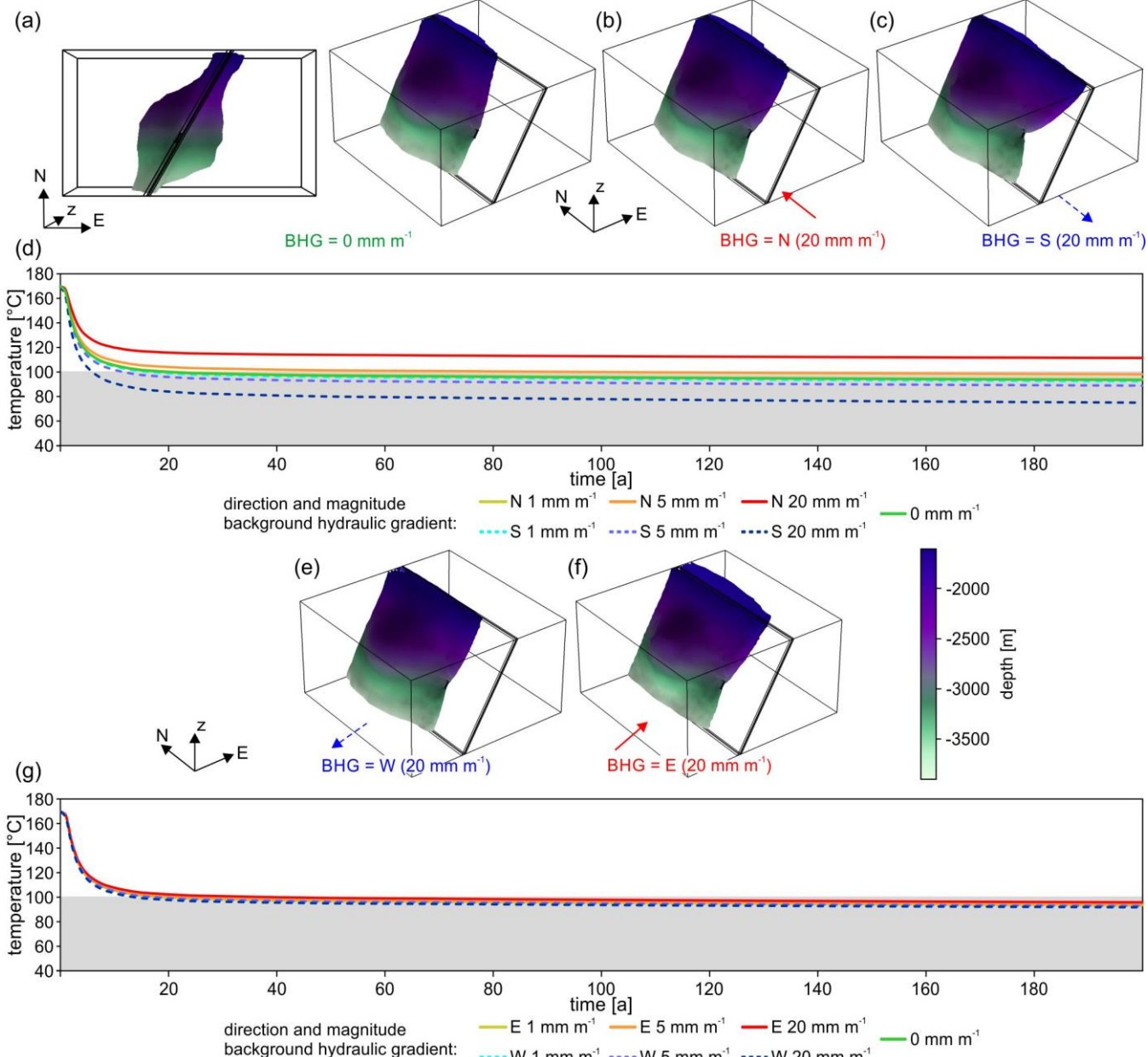

Fig. 6: Temperature development and shapes of the HDI for faults that consist of a damage zone. In (a), (b), and (c), the shapes of the HDI after 100 years for fault-parallel BHGs. The HDI is in large parts restricted to the damage zone, but expands around the injection well into the host rock. Plots (d) and (g), show the temperature developments over time of the produced fluid and contain the results of 13 individual simulations. In (d) it shows, that the production temperatures, independent of BHG, drop sharply in the first 10 years and become, independent of BHG, almost stable after about 20 years. Notably, the differences between the different BHGs are comparatively small. In (e) and (f), the shapes of the HDI for BHGs normal to the strike of the fault after 100 years. (g) the temperature developments for east- and westwards-oriented BHGs are almost identical.

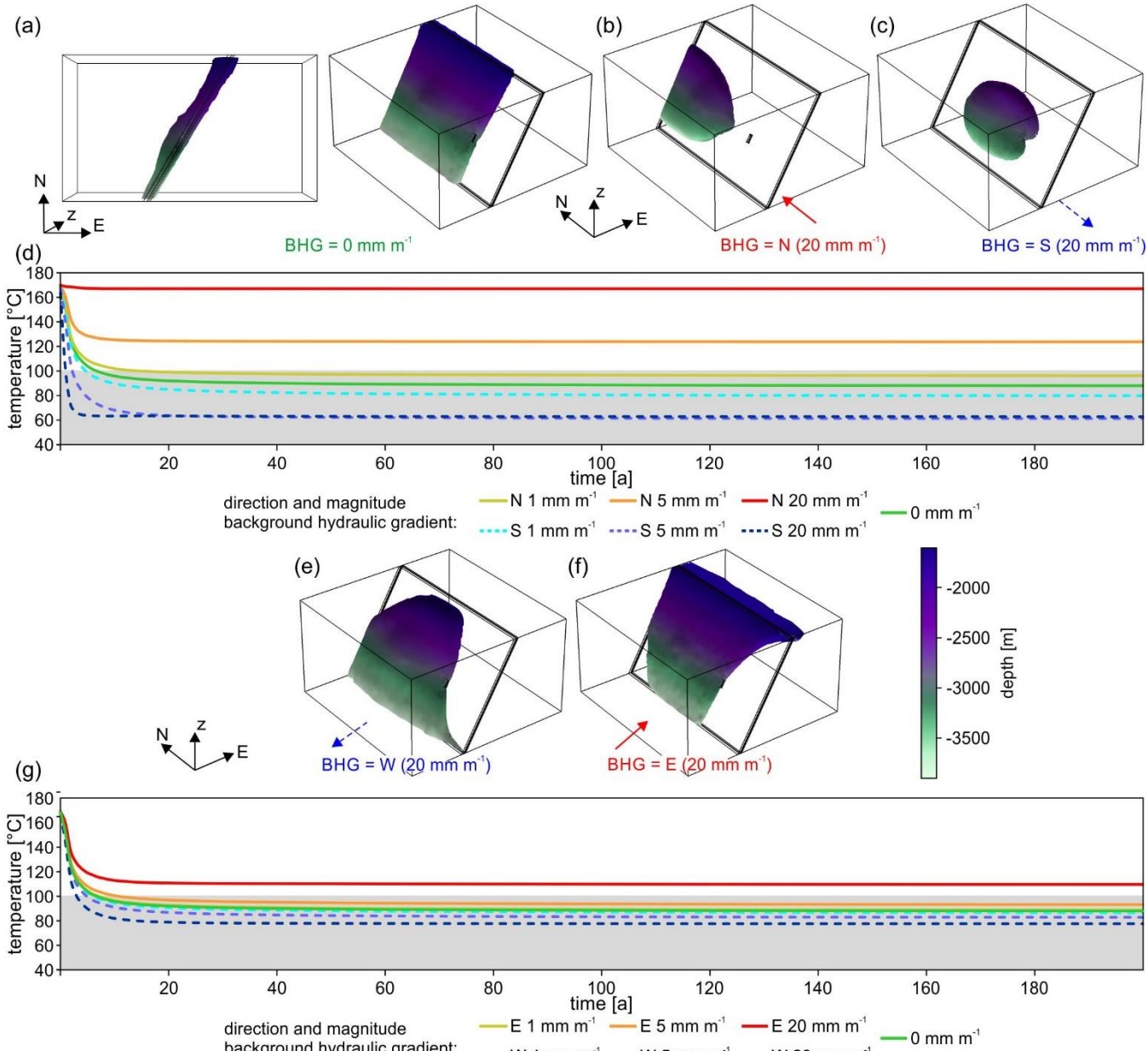

Fig. 7: Temperature development and shapes of the HDI for faults that consist of a damage zone with a positive fault-parallel fracture anisotropy. In (a), (b), and (c), the shapes of the HDI after 100 years for fault-parallel BHGs. The HDI is restricted to the damage zone. Fault-parallel BHG has a strong effect on the shape of the HDI. Plots (d) and (g) show the temperature developments over time of the produced fluid and contain the results of 13 individual simulations. (d) high northward BHGs hinder thermal depletion efficiently, while southward BHGs cause almost immediate depletion. In (e) and (f), shapes of the HDI for BHGs normal to the strike of the fault after 100 years. Together with (g) they show that a BHG in dip direction of the fault has a negative effect on reservoir lifetime, compared with a BHG opposed to the dip direction, which leads to a very short thermal lifetime, i.e., less than 10 years.

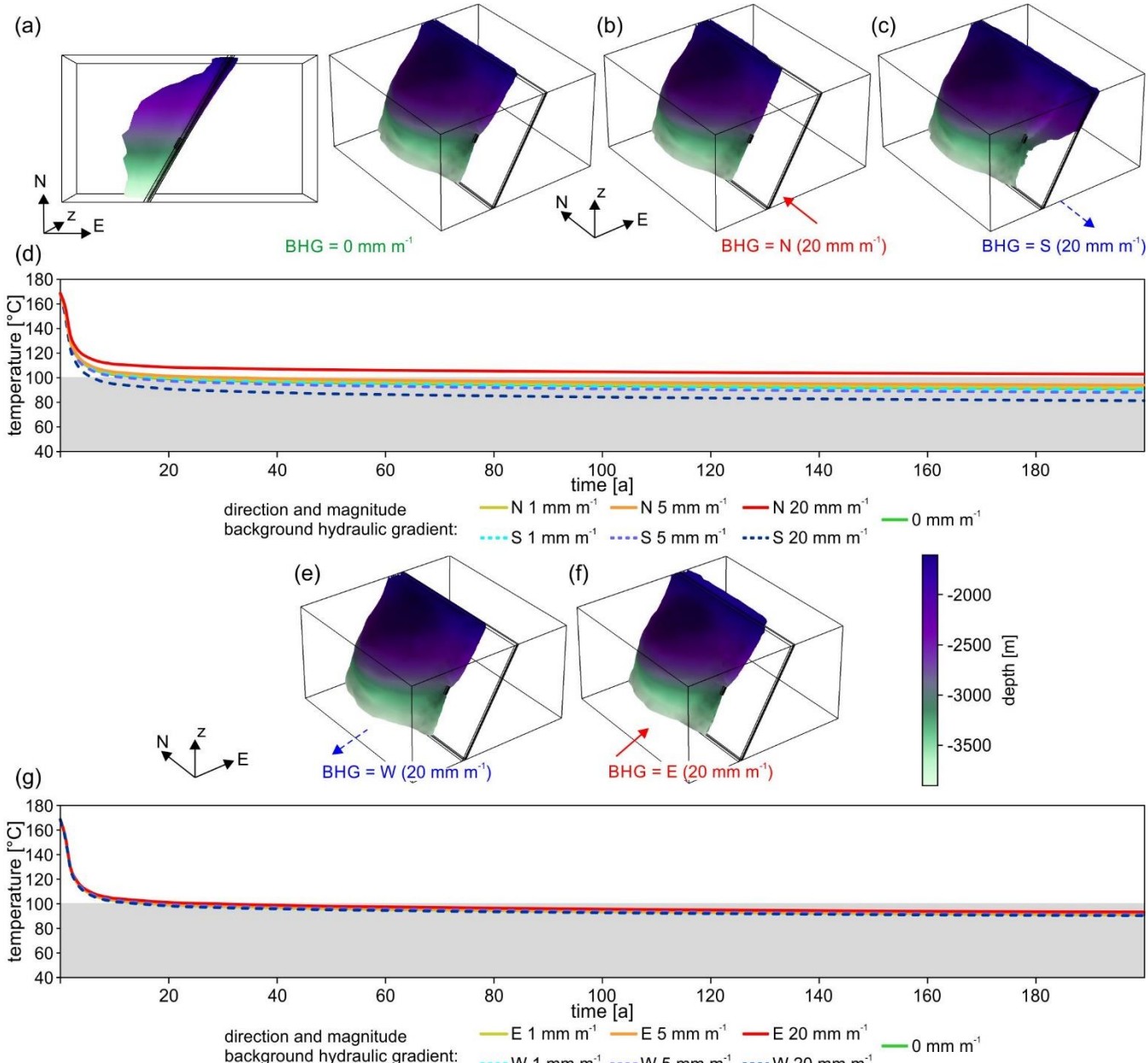

**Fig: 8: Temperature development and shapes of the HDI for faults that consist of an impermeable fault core surrounded by two damage zones. In (a), (b), and (c), the shapes of the HDI after 100 years for fault-parallel BHGs. The HDIs are restricted to the east by the impermeable fault core, but extend to the west into the host rock at the injection well. Plots (d) and (g) show the temperature development over time of the produced fluid and contain the results of 13 individual simulations. (d) orientation and magnitude of the BHG have comparable small impact on the temperature development. After a sharp initial drop, the temperatures stabilize after about 10 years. Only in the case of the highest northward-directed gradients does the temperature stays above 100°C. In (e) and (f), shapes of the HDI for BHGs normal to the strike of the fault after 100 years. The HDI shapes are almost identical to those for BHGs**

parallel to the fault. (g) the temperature developments for BHGs normal to the fault are similar for all tested magnitudes, i.e., the 100°C threshold is reached after 10–20 years and the temperatures stabilize just below the threshold.

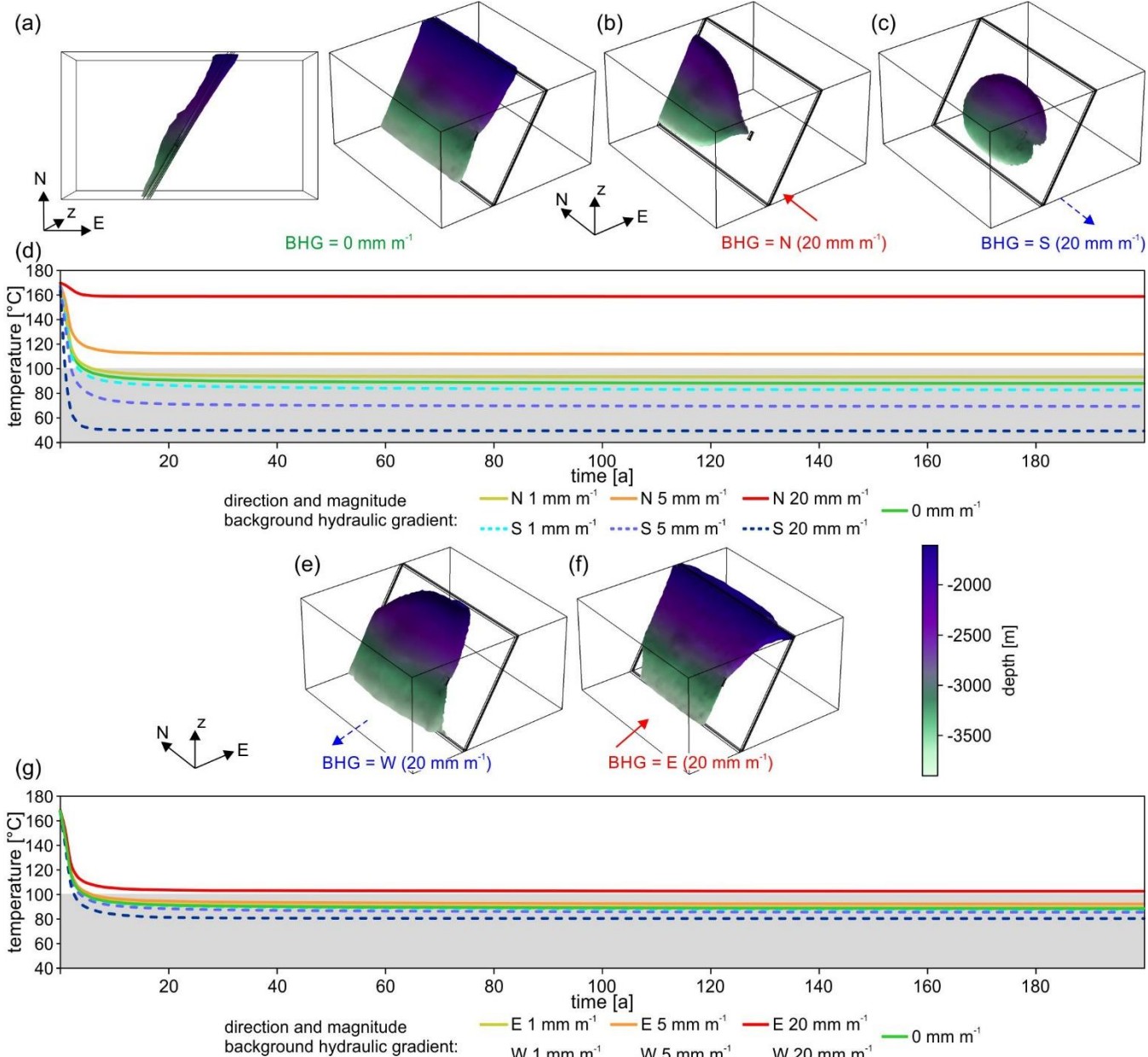

Fig: 9: Temperature development and shape of the HDI for faults consisting of a damage zone with fault parallel fracture anisotropy and an impermeable fault core. In (a), (b), and (c), the shapes of the HDI after 100 years for fault-parallel BHGs. The HDIs are entirely restricted to the western damage zone. Their shape within the fault zone varies significantly, depending on the BHG. Plots (d) and (g), show the temperature development over time of the produced fluid and contain the results of 13 individual simulations.

**(d) northward BHGs completely suppress temperature drops at the production well, southward BHGs lead to immediate depletion of the reservoir. In (e) and (f), shapes of the HDI for BHGs normal to the strike of the fault after 100 years.**

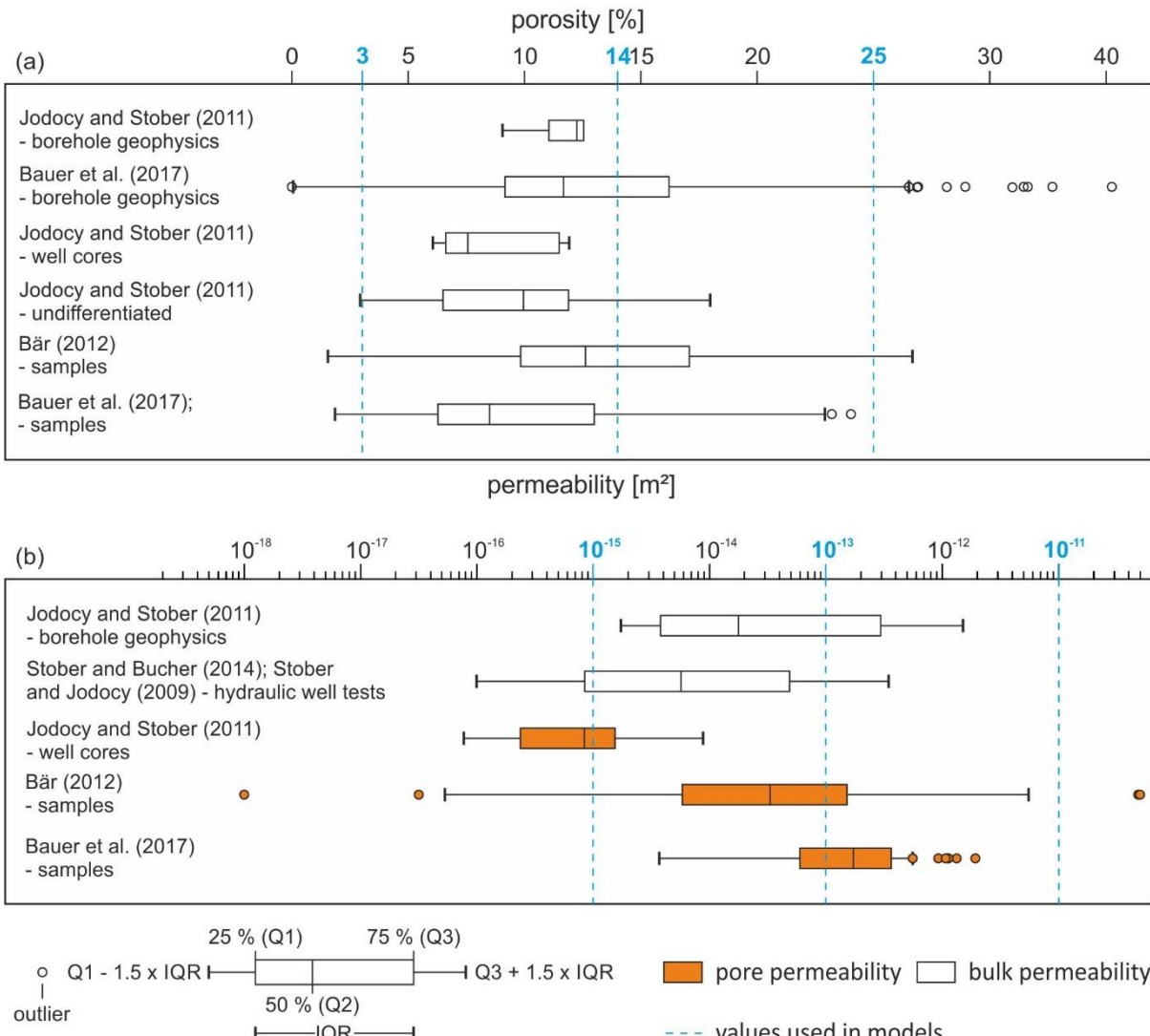

Fig. 10: Box-and-whisker plot of values reported for permeability and porosity of Buntsandstein rocks from the Upper Rhine Graben. (a) The values for porosity reach from almost zero to more than 25%, with the majority between about 7 and 17%. (b) The values for permeability cover a range larger than 7 orders of magnitude. Parameters used in this study are indicated by the blue lines. Modified after Bauer (2018).