# Peer review of "A numerical sensitivity study of how permeability, porosity, geological structure, and hydraulic gradient control the lifetime of a geothermal reservoir"

_Solid Earth, 2019_

## Referee Comment (RC1) · Anonymous Referee #1 · 1 Jul 2019

**General comments**

The manuscript deals with the impact of petrophysical properties, background hydraulic gradient, fractures and fault zones on the performance/lifetime of a deep geothermal reservoir. Thereby, a systematic numerical modelling approach has been applied. The manuscript is well written. The English writing does not need any improvement, except for some very rare typos. Figures are of excellent quality. The topic is also of interest to both the deep geothermal and wider geoscientific community (e.g. basin hydraulics) and publication is recommended. However, the manuscript would benefit from some minor revisions related to the structure and in particular some technical points:

- Manuscript structure: For any generic numerical study, appropriate input parameters and real-world analogs are important. I would therefore recommend to merge the first part of the "Discussion" (lines 312-327) with the "Introduction" and to move or even repeat some parts in the "Methods" section, in particular the "Scenarios" section. The reader of the manuscript would greatly benefit from a direct real world example for the chosen permeabilities, porosities and in particular background hydraulic gradients (BHG) right in the "Methods" section. Especially, the various BHGs require some geological scenarios (what can cause a directed BHG? Topography, overpressure, ...?). Also, the authors might consider merging the entire discussion with the results section for better readability.

- Convection: Convection is not considered in the numerical modelling to save computational cost. As the authors state correctly, convection is likely to be neglected in sediment layers. However, in fault zone-controlled reservoirs, convection is known to have a big impact on the initial temperature field (e.g. Soultz-sous-Forets). Please at least discuss the possible impact of convection on this study's results related to fault zones or consider running a few models that account for convective flow to highlight the impact.

- Bottomhole pressure (BHP) and flow rate: The authors work with a fixed flow rate, which for the low and medium permeability scenarios results in impossible bottomhole pressures well above the lithostatic stress. Nevertheless, this is only mentioned briefly at the end of the manuscript. Here the authors also state that in these cases "*the BHG is outperformed by the artificial flow field caused by the very high bottomhole pressure*". This has to be mentioned directly in the "Methods" section. The actual value of the low and medium permeability models has to be questioned. The BHG appears to be one of the main drivers, but it is completely overruled by the impossible BHPs in the low and possibly also medium perm-scenarios. In that way, only the low and medium perm model without BHG (0 mm/m) might have some value since the shape of the HDI should not be impacted in that scenario (or is it?). In addition, wouldn't the induced BHPs also impact the flow velocity in the reservoir and therefore also thermal breakthrough (I am not certain here, but at least mention and discuss)? As a consequence, I would recommend to exclude all other low and medium perm scenarios with a BHG > 0 mm/m. Otherwise please discuss accordingly and inform the reader in the "Methods" section about a) the unrealistic BHPs, b) their impact and c) why the models might still have some value. Alternatively, the models could be rerun for different flow rates (e.g. with a fixed draw-down pressure, which is a much better technical parameter to be controlled and more or less independent of the geology/petrophysics).

**Please see detailed line-by-line comments below:**

**Abstract**
Well written, please consider to avoid usage of acronyms (BHG and HDI)

**Introduction**
Line 33: Maybe better say hydrothermal than deep geothermal (petrothermal/HDR is also deep geothermal, but only produces from fractures)

**Methods**
Very minor, but almost all sentences start with "We…"

*Numerical model:*
This section is very well written and nicely explains the governing equations!

*Geometry of the model:*
The horizontal extent of the model seems to be rather small (only 4 km), while the vertical extent is very high (2.3 km). It is not clear if this extent only represents the reservoir or also overburden and footwall sediments. Please specify.

Line 91: The rescaling of the well diameter and "length" is confusing. Please explain in more detail, how and why the rescaling has been done and what is meant by "length" and "active part" (perforated production zone?).

*Temperature:*
Line 97: The gradient's unit is wrong (should be 0.047 degC/m not per km). Also, please briefly explain why the respective gradient and surface temperature have been chosen. Especially, since the gradient is very high and the surface temperature is very low.

Line 105: This explanation of the model size should be move to the geometry section (2.2). The explanation itself is not really convincing: the model probably could have been extended to 10x10 km without significantly more cells, since no high resolution is required at the boundaries and far away from the wells. Please at least mention/discuss possible effects here and in the discussion section.

*Fluid flow:*
Please explain the setups of the various background hydraulic gradients here or later (see next comment). Also please explain how the variation is implemented. Figure 1b is not doing a good job explaining the variation. Is the BHG varying from the center towards a certain direction? Or from one "edge" of the model domain to the opposite one? Is the BHG a differential gradient in the reservoir or the entire cube? Since this seems to be such an important parameter, please try to be as precise as possible. Also, please provide some geological scenarios that justify the chosen variations in hydraulic gradient.

*Scenarios:*
Line 127: At 2-3 km burial depth, a matrix permeability of 10-11 m2 (10 Darcy) seems a bit high and probably impossible, when combined with 3% or 14% porosity. Please discuss or at least think about removing the high-perm-low-poro scenarios (or give an adequate geological scenario). In general, please consider giving some real world analogs/examples for the chosen poro-perm scenarios. The sandstone reservoir literature should be full of good examples.

Line 145-146: It would be nice to have some real-world justification for the chosen fault permeabilities. There is a lot of literature available.

Lines 149/150: Please provide some geological scenarios that justify the chosen variations in hydraulic gradient.

**Results**
Line 165/166: According to figures 2e & 2f, this is only true if the BHG is applied in the direction of the injection well (fig. 2f).

Line 180: This makes sense, but how realistic is it to have a rock/sediment with a permeability of $10^{-11}$ m² and a porosity of only 5% or 14%?

Line 236: Why is the stabilization at 100°C?

Line 237: Wouldn't you expect a significant effect of convective flow in a vertical fracture?

Line 253-254: Please rephrase or put more detail. What do you mean by: "a closed geothermal loop may not be feasible"?

Line 258: Not sure what we can really learn from this part, since many real-world projects have shown the significant impact of convection on the temperature field of fault-controlled reservoirs (e.g. Soultz-sous-Forets).

Line 258f: What is the permeability of the matrix (host rock)?

Line 291: "…BHG,  the temperature stays…"

**Discussion**
Line 313-328: Maybe this part would be much better placed in the introduction and in some parts in the "Scenarios"-part (see previous comments on mentioning analogs etc).

Line 335: How does the bottomhole pressure impact the influence of the BHG? In particular in the low-permeability case? Please mention earlier (e.g. in the Methods or Scenarios section(s)).

Line 335f: Here is the answer of the last comment: "the BHG is outperformed by the artificial flow field caused by the very high bottomhole pressure". Actually, the bottomhole pressures in the medium and low permeability cases are impossible in nature. The question is then, what is the meaning of the modelling results? An elegant way to avoid this problem would be to work with a constant draw-down instead.

Line 361: Please consider providing some geological scenarios for variations in BHG.

Line 379f: "Notably, in the low and intermediate permeable models, where permeability contrasts are higher than 1 order of magnitude, none of the tested BHG configurations could compensate for the small volume". Or is this again related to the unnaturally high BHPs in the low and medium permeability scenarios? Please discuss.

Line 387: instead of "borecore": core from boreholes

---

## Referee Comment (RC2) · Anonymous Referee #2 · 24 Jul 2019

The manuscript is well written, original and of interest. It is worthwhile to be published in Solid Earth Journal.

In the following, my (hope useful) comments and suggestions to improve the manuscript. General comments:

Figures in general have a small scale for (small) colored dots and a (uselessly) large scale for the vertical. Furthermore, the colors used are the same. This is misleading the reader. My suggestion is to use different color codes for the two parameters (depth and BHG) and change the relative dimensions of the two scales, since the focus of the manuscript is on the BHG (color-coded dots).

[Figure]

Figs.2a-c (as well as other corresponding plots) either have inverted y-axis scale (sic!), or I did not understand the figure and/or the text (cfr. lines 161-165). This produced some initial misunderstanding of the work (the text is not properly describing what is presented in the figure). In many experiments the temperature stabilizes at around 100°C (Figs. 2j, 6g,7g, 8d, 8g, 9g). The reason for this coincidence with the HDI not clear or explained. The author should justify this "convergence" in the various models.

In Fig.1a the projection of the wells provides the impression that their trajectory is oblique. The Author should either correct the figure or describe the reason for oblique wells as well as quantify it. In the Figures the Authors should include the number of experiments represented (i.e. the number of dots in the single figure). The author should discuss the case of a strong variation in the results (e.g. Fig. 1a, red dots for BHG=20 mm/m, at permeability 10-11). Numbers are very small to pretend some statistics (mean, sd), but they could mean unreliable results and should be discussed.

The "line connecting the same experiment" is not clear. In Fig. 2 the yellow line connect one yellow dot per figure, and it is easily understood. On the other hand, in other figures (e.g. Fig. 4a-b, green lines) the do connect multiple dots in the same figure. This is confusing: how many numerical experiments were responsible for each dot in each figure (I assumed one)? Maybe they partially overlap.

Specific comments

In lines 34-39 the Authors discuss the poor improvement in porosity due to the presence of fractures. This is true, but the author are not considering the main role provided by fractures in improving the effective porosity by connecting isolated pores, as is normally achieved in tight-gas reservoirs (gas-shale). In general, the manuscript is not discussing on the difference between total porosity and effective porosity. I guess that the porosity they consider in the numerical experiment is merely the effective one, and this should be clearly mentioned. On the other hand, a brief note on the role of influence of fractures on effective porosity is required to complete the introduction and the

discussion paragraphs.

Equations in Line 72,81,82 seem correct, yet references for the general audience (as Solid Earth also has) are required.

Line 91-92. The limit to 20m is not easy to be understood (e.g. where these 20m were located along the well). Maybe a better way to express this correction would be to express it as a percentage of the well hole surface (the cylinder), or by presenting the equivalent reduction in permeability between the well cell and the surrounding ones in the mesh.

Line 96: I guess that the geothermal gradient is in reality expressed by m and not by km... Line 105 and 113. I guess that "computational costs" really intends the more appropriate expression "computational time". It would be of interest to the readers to quantitatively justify this sentence: add in lines 69 71 information on the used computer platform and the approximated run-time for a single numerical experiment.

Line 126 and through all the experiments. My opinion is that a permeability of 10exponential-11 m2 is unfair to be reached in a reservoir at the used resolution of the model, with the exception of karst cavities. The Author might include here a descriptive correspondence to the reservoir permeability (e.g. tight reservoir for 10 exponential -15 m2, medium-high permeable reservoir 10 exponential -13 m2, karst structures 10exponential-15 m2).

Lines 143-147 more references on measure of permeability in fault core are important here (e.g. works by R.J Knipe and/or Q.J Fisher)

Line 161-165. As mentioned, the only way I found to correlate text and Fig2a-c is to invert the Y-axis scale. Anyhow the description, even with this correction, does not correlate for 10exponential-11 permeability experiments, that scatter results all along the entire span 0->200 a apparently without any rule (e.g. red dots). Did I understand properly the figure? If not, a more careful introduction to the figure and description

might be necessary.

In the Figs. the meaning of the represented surface is not completely described. The Authors refer to "HDI shape". I am not sure but Iguess that, considering the experiments, these surfaces represent the envelope of the volume where the temperatures become lower than the HDI due to the successful heath extraction. An explanation on the meaning of the HDI shape is required in the text (and maybe in the caption for the fast readers...).

Line 173-174 the probability concept should be better introduced.

Line 178: I guess the Authors intend Fig.2g and not 2e.

Line 198 "three series". This is not clear: I see in the figure3 different permeability (these are the three series), 4 permeability contrasts and 8 different orientation for BHG with 4 possible gradients, for total of 3x4x8x4=384 combinations. Then just three BHG shapes, but for the same permeability (same series). This might be confusing. A more complete description of the model procedure might help to understand the results.

Fig4b is not clear, and in general figs 2,4,5 are not easy figures. Same color dots appear both on high and very low times to breakthrough. This could mean the excessive scattering of results, or that results are from experiments with different, not specified, parameters. I think to have properly understood the relations between the dots in Fig. 4a,b and the reason for the limited connection presented in Figs. At the present stage , the figure is very difficult to be understood (also due to the high number of combinations in the experiments – i-e- the number of parameters used - and the limiting 2D of the journal pages...). The Author could try to improve the correlations by either using different symbols for each experiment (good luck, it would be a big effort with questionable results) or by adding a reference number to each dot. The diagrams have a relative small number of dots and al lot of empty space. A simpler alternative might be to add in the text the clear description of a correlation among dots as an example. There are also some evident overlap of dots (just comparing among figures) and this should be described (or slightly move one of the dots within the resolution of the results).

Line 215-220 again: the cited 70 years seems to correspond to 130 years in Fig.4d, second column. Is there again reversed the Y-axis scale?

Line 235. As previously mentioned. Why at 100°C? This should be justified by the Authors Fig.5 the origin of dots on top of the plots a-c (i.e. at >200a) is not clear.

Line 264-265 Fig. 5g shows that temperatures stabilize at 100°C. How this happens at exactly the critical temperature chosen for the HDI? Is this input in the model? Some explanation is needed. Lines 340-349 Here is perhaps the proper space to discuss the total porosity and the effective one I discussed above. As I understand, the chosen porosity is intended to be 100% effective. A sentence explaining this should be anyhow added to the article.

Line 371. This assumption may be too forced, and I am sorry for the referenced articles. Secondary fractures and faulting allow permeability to take over thinner clay layers that lose their sealing property. This is more difficult in thicker clay layers. I understand that in the useful proposed model are necessary simplifications, but it is not the case for the complexity of real geothermal reservoirs. Line 399. I do not see evidence in Fig. 5b to justify this sentence. At my sight, the resulting timings are fully independent from the BHG values (colored dots). May be the Authors are referring here on the BHG orientation of Fig.5c.

Line 423-414 Fractures and secondary faulting associated to faults have generally various angles to the faults and only a minority lies parallel to it (cfr. Riedel). This results in: fracture intersections, fracture opening by the stress induced from the kinematics along the fault (friction). These factors guarantee the higher permeability of fault damage zone to a certain extent, as described in the literature. To be explicit: "often-observed" of "fault-parallel fracture anisotropy" does not correspond to either field outcrops and

cores across fault zones, apart from S-C structures, where in any case C planes are generally subordered in number to S ones, My suggestion is simply to eliminate the "often-observed" attribute.

Line 429 The previous concept is repeated here: useless redundancy and same comment.

Line434-435 the use of the terms "opposed/opposite" to indicate opposite (!) dipping is misleading. A rephrase would solve it.

Lines 62, 442: they were 1027 (from line 150). This is an interesting and serious number of runs and it would be effective to remark this number both in the introduction (say, "over one thousand numerical experiments") as well in the Conclusions "(1027)". My impression is that "large series" or "a series of" would be –alas – interpreted as much smaller number in present-day publish-or-perish scientific environment.

Line 457: This is not so simple. This sentence does not take in consideration the improvement of the effective porosity that is induced by fracturing that in turn may be enhanced by the oriented stress that develops in presence of strong BHG. Since the point about effective porosity changes is not taken into consideration in the presented models, my suggestion is to specify this in the sentence (referring to "in many cases" might be not sufficient).

Line 459-462. On the contrary, results from this work well represent the first step to model real, complex geothermal reservoirs with their Stochastic modelling by adding in the mesh the proper random values! And I am sure that the "computational costs" at that stage will be an insignificant obstacle. This might be a further point and a better conclusion to your article (follow the Hollywood-movie style: end always your articles with a true, positive sentence on your results...)

---

## Author Comment (AC1) · 3 Aug 2019

Reply to the review by RC1:

We thank the reviewer for the positive evaluation, for the critical questions, and the insightful comments, which help us to improve the manuscript. The final corrections of the manuscript will be provided and marked up in the final response to all reviews. Please find our answers below:

General comments:

1. Manuscript structure: Reviewer comment 1.1: For any generic numerical study,

appropriate input parameters and real-world analogs are important. I would therefore recommend to merge the first part of the "Discussion" (lines 312-327) with the "Introduction" and to move or even repeat some parts in the "Methods" section, in particular the "Scenarios" section. The reader of the manuscript would greatly benefit from a direct real world example for the chosen permeabilities, porosities and in particular background hydraulic gradients (BHG) right in the "Methods" section.

Authors reply: We agree with the reviewer and will in agreement with the comments of reviewer 2 improve the Introduction regarding this matter. We will now introduce the values we used for permeability, porosity and BHG in the Introduction and discuss how these compare to typical values values for sedimentary basins worldwide. With our manuscript, we, however, present a non-site specific, numerical sensitivity study that investigates the influence of various reservoir parameters on geothermal reservoir lifetimes and how exactly they have to be known to provide reliable estimates on the lifetime of a geothermal reservoir. For this reason, we did not only chose parameter values for permeability and porosity that are desired in geothermics, but also values that lie above and below them. Since our sensitivity study is not site specific, we only present real world scenarios in the discussion (see also point 8.1). This is why we prefer to keep the current structure of our manuscript.

Reviewer comment 1.2: Especially, the various BHGs require some geological scenarios (what can cause a directed BHG? Topography, overpressure, ...?). Also, the authors might consider merging the entire discussion with the results section for better readability.

Authors reply: We agree with the reviewer and see the necessity to explain in more detail in the introduction why we choose to investigate the influence of the BHG and provide examples of settings in which BHGs are to expect or in which they have been observed. We will also justify the values we used for the BHG's magnitude, and refer to the according literature e.g. Fan et al. (2013, Science), Gleeson et al. (2016, NGS) and Grauls (1999). We, however, prefer, regarding the second point addressed here,

to keep the results and the discussion sections separate.

2. Convection: Reviewer comment 2.1: Convection is not considered in the numerical modelling to save computational cost. As the authors state correctly, convection is likely to be neglected in sediment layers. However, in fault zone-controlled reservoirs, convection is known to have a big impact on the initial temperature field (e.g. Soultz-sous-Forets). Please at least discuss the possible impact of convection on this study's results related to fault zones or consider running a few models that account for convective flow to highlight the impact.

Authors reply: We agree that faults/fault zones can have significant heat flow by density-driven convection. We are also aware, as the reviewer states, that there are several real world examples of faults in which free convection has been observed and we will include this fact in the discussion. However, in many scenarios it is also likely that, due to the heterogeneous nature of faults, convection is not present. However, there a very few published examples in the literature. Our models likely underestimate the lifetime of fault-related reservoirs, because they do not include density-driven convection and thus heat supply from deeper levels. The effect of density-driven convection, however, at least to a certain degree, would be to counteract the negative influence of the channeling effect of a fault (see also points 7.4 and 7.6). Regarding the reviewer's suggestion to rerun these models with density-driven convection, would mean that these scenarios are not comparable anymore with the other parts of our study. We will follow the reviewer's suggestion and will now address the possible effects of density driven convection in the discussion section.

3. Bottomhole pressure (BHP) and flow rate: Reviewer comment 3.1: The authors work with a fixed flow rate, which for the low and medium permeability scenarios results in impossible bottomhole pressures well above the lithostatic stress.

Authors reply: The reviewer is correct that the pressures for the low permeability scenarios are extremely high or even impossible. However, since we chose for our numerical sensitivity study to investigate the impact of a range of parameters and parameter values (point 1.1), it is inevitable that some of the combinations represent unrealistic scenarios. These results are nethertheless part of our study and as such help to draw the picture and to understand the effect of the investigated parameters within geothermal reservoirs. Without them, some of the effects would not have been identified by us. In consequence we are convinced that they constitute an integral part and should not be rejected. We hope the reviewer can agree with this and is also referred to our answer to point 3.6. We also wish to note that in case of the medium permeability model, the high pressure could be easily corrected in the model by changing i.e. the depth of the well or the reservoir, or the borehole diameter (see line 334-339 of our manuscript).

Reviewer comment 3.2: Nevertheless, this is only mentioned briefly at the end of the manuscript. Here the authors also state that in these cases "the BHG is outperformed by the artificial flow field caused by the very high bottomhole pressure". This has to be mentioned directly in the "Methods" section. The actual value of the low and medium permeability models has to be questioned.

Authors reply: We agree with the reviewer: we will add in the method section that artificial flow field and BHG interact. That some of the models return unrealistic BHP, i.e. represent unrealistic scenarios, will be mentioned in the introduction and the method sections. Please see also our reply to your point 3.1. Regarding the medium permeability values, please see line 334 - 339 (discussion paper).

Reviewer comment 3.3: The BHG appears to be one of the main drivers, but it is completely overruled by the impossible BHPs in the low and possibly also medium perm-scenarios. In that way, only the low and medium perm model without BHG (0 mm/m) might have some value since the shape of the HDI should not be impacted in that scenario (or is it?).

Authors reply: The value of these models is that they show that if the artificial flow-field

introduced by the bottomhole pressure is stronger than the BHG, the importance of the BHG ceases (Fig. 2a, b). Even though, these models represent unrealistic cases in terms of the bottomhole pressure. To show the same effect in a model suite with higher permeability we would need a BHG far smaller than used in our study. Another example is that the impact of layering on thermal breakthrough times, is less well observable in the high permeability models (because the bottom hole pressure is too low in these cases to investigate the effect; please compare Figure 4a with Figure 4g) and can only clearly seen in the unrealistic low permeability models. We think therefore that these (unrealistic) parameter combinations are an integral part of the study and should not be omitted.

Reviewer comment 3.4 In addition, wouldn't the induced BHPs also impact the flow velocity in the reservoir and therefore also thermal breakthrough (I am not certain here, but at least mention and discuss)?

Authors reply: Flow velocities are limited by how much water is injected and produced from the system, and are therefore not a function of the BHP's. The main effect of the BHP can be seen in its interaction with the background hydraulic gradient. We agree with the reviewer and will now mention this point in the revised manuscript.

Reviewer comment 3.5 As a consequence, I would recommend to exclude all other low and medium perm scenarios with a BHG > 0 mm/m. Otherwise please discuss accordingly and inform the reader in the "Methods" section about a) the unrealistic BHPs, b) their impact and c) why the models might still have some value.

Authors reply: These models are an integral part of our study. Please see our answer to points 1.1, 3.1, and 3.3.

Reviewer comment 3.6 Alternatively, the models could be rerun for different flow rates (e.g. with a fixed draw-down pressure, which is a much better technical parameter to be controlled and more or less independent of the geology/petrophysics).

Authors reply: Firstly, if we had chosen a fixed draw-down-pressure, we would have had to deal, at least in part, with extremely low or high flow rates, i.e., the amount of injected cold water would change. Consequently, we would not be able to analyze the interaction and impact of the tested petrophysical and structural parameters, which is the main focus of our manuscript. Secondly, for making a large series of models that can be compared to each other and in which the effects of individual parameters can be isolated, the option would be to either to use a fixed bottom hole pressure, which will induce unrealistic flow rates in some models, or fixed flow rate, which will result in unrealistic pressures. The choice for fixed flow rate is because this has the least disturbing effect on the model results because the amount of injected cold fluid stays the same. With other words, to rerun some of the models with fixed draw-down pressure would not correspond to the setup of our study, rather it would alter the results of these particular models and therefore destroy comparability. Please see also our answer to your comment 3.1 and 3.3 above, where we answer similar questions.

Detailed line-by-line comments

4. Abstract Reviewer comment 4.1: Well written, please consider to avoid usage of acronyms (BHG and HDI).

Authors reply: Here we follow the standards of the Journal that require the introduction of acronyms in the abstract.

5. Introduction: Reviewer comment 5.1 Line 33: Maybe better say hydrothermal than deep geothermal (petrothermal/HDR is also deep geothermal, but only produces from fractures).

Authors reply: We follow the advice of the reviewer and replace "deep geothermal" with "hydrothermal".

6. Methods

Reviewer comment 6.1: Very minor, but almost all sentences start with "We…"
Authors reply: We agree and we will reformulate this part.

Geometry of the model: Reviewer comment 6.2: The horizontal extent of the model seems to be rather small (only 4 km), while the vertical extent is very high (2.3 km). It is not clear if this extent only represents the reservoir or also overburden and footwall sediments. Please specify.

Authors reply: We think that the best way to approach this issue is: that in our sensitivity study, the whole model domain should be seen as a potential reservoir volume, i.e. our study investigates which parameters control and or influence the volume that actually can be utilized as a reservoir. We will improve the text accordingly to avoid potential misunderstandings. For your comment on the lateral extent, please see our answer to your comment on line 105 below (see also point 6.5).

Reviewer comment 6.3: Line 91: The rescaling of the well diameter and "length" is confusing. Please explain in more detail, how and why the rescaling has been done and what is meant by "length" and "active part" (perforated production zone?).

Authors reply: We will follow the suggestion of the reviewer and rewrite this part accordingly. Standard well diameters are a few decimeters. This in turn would need a very fine mesh. To avoid this issue we used a larger diameter for the wells. To account for the unrealistic high diameter and thus the area of the "perforated production and injection zone" we choose to adjust the area via its length to a size that is in a realistic range.

Temperature:

Reviewer comment 6.4: Line 97: The gradient's unit is wrong (should be 0.047 degC/m not per km). Also, please briefly explain why the respective gradient and surface temperature have been chosen. Especially, since the gradient is very high and the surface temperature is very low.

Authors reply: Thanks for identifying this mistake. We corrected the typo 0.047°C/m.

Since we carried out a non-site specific numerical sensitivity study, we chose a realistic gradient that allows for electricity production at this depth. The surface temperature was chosen arbitrarily to be 0°C. This is in our opinion neither particularly high nor low, especially when considering that our numerical sensitivity study is not site specific. Nethertheless, the effect on the model results can be neglected, since a slightly increased surface temperature would alter not the temperature at target depth or the model results significantly.

Reviewer comment 6.5: Line 105: This explanation of the model size should be move to the geometry section (2.2). The explanation itself is not really convincing: the model probably could have been extended to 10x10 km without significantly more cells, since no high resolution is required at the boundaries and far away from the wells. Please at least mention/discuss possible effects here and in the discussion section.

Authors reply: In Line 103-104 of our manuscript, we describe that the temperature boundary conditions do not affect the model results, i.e. the size of the model domain does not affect the model results. The sentence in line 105-106 is thus obsolete. We will delete the last sentence. This solution also makes merging the description of the model geometry unnecessary. The only limitation by the comparatively small model domain is that we cannot examine in all cases the complete geometry of the HDI (hundred degree isotherm).

Fluid flow: Reviewer comment 6.6: Please explain the setups of the various background hydraulic gradients here or later (see next comment).

Authors reply: Please see our answer to point 6.7.

Reviewer comment 6.7: Also please explain how the variation is implemented. Figure 1b is not doing a good job explaining the variation. Is the BHG varying from the center towards a certain direction? Or from one "edge" of the model domain to the opposite one? Is the BHG a differential gradient in the reservoir or the entire cube? Since this seems to be such an important parameter, please try to be as precise as possible.

Also, please provide some geological scenarios that justify the chosen variations in hydraulic gradient.

Authors reply: The BHGs are valid for the whole model domain, i.e. the BHG is not varied in the individual models, but interacts with the artificially introduced flow field. The BHG is applied as a pressure gradient on the model boundaries. We will explore this in more detail and improve Figure 1b. Please see also our answer to point 1.2.

Scenarios: Reviewer comment 6.8: Line 127: At 2-3 km burial depth, a matrix permeability of 10-11 m2 (10 Darcy) seems a bit high and probably impossible, when combined with 3% or 14% porosity. Please discuss or at least think about removing the high-perm-low-poro scenarios (or give an adequate geological scenario). In general, please consider giving some real world analogs/examples for the chosen poro-perm scenarios. The sandstone reservoir literature should be full of good examples.

Authors reply: We improve the method section to clarify this misunderstanding and add that the permeability values are not linked, respectively provided/controlled by the matrix porosity. We used instead a continuum approach (Berkowitz et al., 1988; Lege et al., 1996; Kolditz, 1997), that uses a replacement media for the fractures and which provides mean hydraulic properties of a given fracture system. This is in our opinion a justified assumption, since permeability is in consolidated sediments often to large parts provided by fractures (Bear, 1993; De Marsily, 1986; Hestir and Long, 1990; Nelson, 1985). Please see also our answer to point 1.1.

Reviewer comment 6.9: Line 145-146: It would be nice to have some real-world justification for the chosen fault permeabilities. There is a lot of literature available.

Authors reply: We accept the suggestions of the reviewer and justify the chosen parameter values in the introduction. Please see also our answer to point 1.1.

Reviewer comment 6.10: Lines 149/150: Please provide some geological scenarios that justify the chosen variations in hydraulic gradient.

Authors reply: We follow the reviewer's suggestion and explore the topic in more depth in the introduction. Please, see our comment above to point 1.2.

7. Results:

Reviewer comment 7.1: Line 165/166: According to figures 2e & 2f, this is only true if the BHG is applied in the direction of the injection well (fig. 2f).

Authors reply: The reviewer is correct. Here we provide/describe the ranges of reservoir lifetimes observed in scenario 1, for different reservoir permeabilities. These ranges depend naturally also on the other parameters varied in our multi parameter sensitivity study. This is why we choose to present our results in different plots, e.g. lifetime vs, permeability, and lifetime vs direction of the hydraulic gradient.

Reviewer comment 7.2: Line 180: This makes sense, but how realistic is it to have a rock/sediment with a permeability of 10-11 $m^2$ and a porosity of only 5% or 14%?

Authors reply: Please see our answer to your comments to Line 127 (point 6.8).

Reviewer comment 7.3: Line 236: Why is the stabilization at 100°C?

Authors reply: In our study, we investigate the effect of multiple parameters; there are certain combination that can produce similar results, in this case the convergence to 100°C in Figure 2j, 6g, 7g, 8d, 8g, 9g. To analyse this in more depth, would require a different sensitivity study with a different setup. We will also modify the sentence to: "In the presented model runs as shown in Figure 4, temperatures stabilize at a final temperature of about 100°C."

Reviewer comment 7.4: Line 237: Wouldn't you expect a significant effect of convective flow in a vertical fracture?

Authors reply: We assume that this question is likely caused by the fact that we were not clear enough about how permeability is implemented. See also our answer to point 7.6 and 2.1. We will also rephrase the sentences in line 238-239 (discussion paper)

[Figure]

to: "..., compared to the other directions, as common in fractured reservoirs (Figs. 1e, 5)." We did not introduce additional vertical fractures in this scenario, but increase the fracture anisotropy in the given plane. This question would be necessary to answer if we would have used a discrete fracture model. We are convinced that this question will be answered after we improved the method section regarding the implementation of permeability and porosity. Also in natural fracture systems the vertical extent of fractures is commonly restricted, i.e. many fractures stop at sedimentary contacts/layers and thus density-driven heat flow would be hindered in the vertical direction, as the reviewers agrees in point 2.1.

Reviewer comment 7.5: Line 253-254: Please rephrase or put more detail. What do you mean by: "a closed geothermal loop may not be feasible"?

Authors reply: We agree with the reviewer and rephrase the sentence to "…...the establishment of a closed geothermal system becomes unlikely."

Reviewer comment 7.6 Line 258: Not sure what we can really learn from this part, since many real-world projects have shown the significant impact of convection on the temperature field of fault-controlled reservoirs (e.g. Soultz-sous-Forets).

Authors reply: We discuss this limitation now. The main point will be: Whereas our models likely underestimate the lifetime of fault-related reservoirs that allow for convection, they allow for improved estimate of how strong the effect of convection should be to counteract the negative influence of the channelling effect. Thus, it shows the importance to know the budgets of both the channeling effect and the effect of density driven convection to make assumptions about their effect on the potential lifetime of a geothermal reservoir. Please see also our answer to point 2.1 and to the lines 363-368 of our manuscript, where we refer to a real world example of this observation.

Reviewer comment 7.7: Line 258f: What is the permeability of the matrix (host rock)?

Authors reply: See line 141f (Discussion paper). We agree with the reviewer and now

repeat the value of the bulk permeability of the host rock in this section to improve readability. Please note our answer to point 6.8 in which we clarify how permeability is implemented.

Reviewer comment 7.8: Line 291: "…BHG, does the temperature stays…"

Authors reply: We thank the reviewer and will correct the sentence.

8. Discussion

Reviewer comment 8.1: Line 313-328: Maybe this part would be much better placed in the introduction and in some parts in the "Scenarios"-part (see previous comments on mentioning analogs etc).

Authors reply: Our study is a non-site-specific sensitivity study (with simplified models). We use this part as an introduction in the discussion section to show how parameters such as porosity and permeability can be highly variable. We discuss that even comparatively small variations of these parameters have a strong effect on a reservoir's performance. Thus, we think that the structure of the manuscript, as it is, is justified. We are aware, however, that it would be possible to tell the story in a different way. However, now we will discuss in more detail the implication of figure 10 for our study and geothermal energy in general. Further, we will better explain in the Introduction the aim of our study. This will also include a point regards the variability of geological systems. See also our answer to point 1.1.

Reviewer comment 8.2: Line 335: How does the bottomhole pressure impact the influence of the BHG? In particular in the low-permeability case? Please mention earlier (e.g. in the Methods or Scenarios section(s)).

Authors reply: The influence of the bottomhole pressure on the BHG depends on the ratio between both. If the bottomhole pressure is higher than the BHG it dominates and vice versa. The low permeable cases are due to high bottomhole pressures unrealistic, but allow to investigate the effect of other parameters like permeability contrasts. In our

opinion, these points are preferably placed in the discussion section. We, however, will state that some of the scenarios are unrealistic and that both fluid systems interact and we will specify that point in the method and discussion section. Please see also our answer to point 3.3.

Reviewer comment 8.3: Line 335f: Here is the answer of the last comment: "the BHG is outperformed by the artificial flow field caused by the very high bottomhole pressure". Actually, the bottomhole pressures in the medium and low permeability cases are impossible in nature. The question is then, what is the meaning of the modelling results? An elegant way to avoid this problem would be to work with a constant draw-down instead.

Authors reply: We agree with the reviewer please see above. The low permeability/ high fluid pressure models underestimate the effects of the background hydraulic gradient. The importance of the findings of the low and medium permeability models as well as the use of the constant production and injection rates are justified above in points 3.1, 3.3, 3.6. We hope that answers the questions raised by the reviewer.

Reviewer comment 8.4: Line 361: Please consider providing some geological scenarios for variations in BHG.

Authors reply: We assume that this is a misunderstanding. We have not introduced variations of the BGH within one individual model, but we assigned different BHG to individual models. We improved the method section to clarify this issue. Please see our comments to points 1.2 and 6.7. We also, as requested, provide improved introduction regarding the BHG.

Reviewer comment 8.5: Line 379f: "Notably, in the low and intermediate permeable models, where permeability contrasts are higher than 1 order of magnitude, none of the tested BHG configurations could compensate for the small volume". Or is this again related to the unnaturally high BHPs in the low and medium permeability scenarios? Please discuss.

Authors reply: We agree with the reviewer, and add that the unrealistic high bottomhole pressures do not allow the BHG to affect the system. We also correct the typo "higher than 1 order of magnitude" to "higher than 2 orders of magnitude".

Reviewer comment 8.6: Line 387: instead of "borecore": core from boreholes.

Authors reply: We thank the reviewer and we will correct the sentence accordingly.

---

## Referee Comment (RC3) · Owen Callahan (Referee) · 12 Aug 2019

Bauer et al. (2019) present results of a sensitivity analysis exploring the impact of various geologic parameters (porosity, permeability, sedimentary layering, faults, and fracture anisotropy), well separation, and background hydraulic gradient (BHG) on breakthrough time of and shape of a $100°C$ isotherm in a generic sedimentary geothermal system. The results lay out a convincing argument that BHG is a significant and perhaps under appreciated aspect of geothermal field longevity, although the manuscript in its current form does not establish the importance of BHG early enough, nor does it stay focused on BHG as a central part of the hypothesis, and therefore loses some

impact.

General comments (see specific comments for details)

Model parameters, including selection of porosity-permeability combinations, length of model duration, selection of 100°C isotherm, are not sufficiently justified, and may not be relevant to operating geothermal fields.

Use of references and citations is inconsistent. In some cases, statements with long lists of references are too vague to be useful (i.e. not clearly tied to particular geothermal fields or a specific type of inquiry (numerical, field, experimental...)) and in other cases the listed references do not seem appropriate for citing in their current context.

The structure of the paper fails to emphasize the role of BHG nor does it discuss enough real world scenarios were the impact of BHG, or even suspected impact of BHG, can be shown. As it stands, almost all of the conclusions are about BHG, but BHG only gets 3 lines in the introduction.

Specific comments

Line 11: This sentence neglects economic factors. Rather than "can be exploited" maybe describe geologic factors influencing economic viability, as you do in the introduction.

Line 17: 100° C isotherm is not well justified. See additional comments below.

Line 29: The first few lines of this paragraph make it seem like these references pertain to hydrothermal settings specifically. In this current configuration, Laubach et al. (2009) does not seem like an appropriate reference as they do not describe fracture patterns in hydrothermal systems, nor do they explicitly describe the impact of fractures on permeability or volume (other than tangentially) but rather compare fracture and mechanical stratigraphy.

Line 32: Manning and Ingebrigtsen (1999) concerns theoretical permeability at the

crustal scale and in metamorphic rocks in particular. The link between this reference and the statement are again tenuous unless more clearly explained.

Line 37-39: The logic here is odd. You describe high porosity in sedimentary geothermal systems, then say fracture porosity in sedimentary rocks is low (are dam sites really the best analog, i.e. Snow, 1968?), but that fractures dominate geothermal systems. Separately these statements may all be true, but fractures commonly dominate in geothermal systems because geothermal systems are commonly not hosted in sedimentary rocks. Also, you may want to specify "clastic" sedimentary reservoirs, as fractures can be very significant contributors in carbonate rocks.

Line 42-43: The statement about specific failures needs referencing.

Line 45: Beall (1994) does not appear to be about declines in production fluids nor fault damage zones, but rather to be about tracer tests and what can be learned about fluid saturations.

Line 48-50: BHG is a huge part of your overall paper but has a tiny role in the introduction. This should be much larger, with specific examples of where it has impacted production. It could be your primary hypothesis and seems like the major contribution, but it is not firmly established in the introduction. As it stands, the introduction does not lay the necessary foundation for the paper, not establish a clear hypothesis, but it could be reworded to emphasis BHG (see comments about 363-368).

Line 58: The lifespan of 200 years in not well justified. This is longer than the nominal lifespan of geothermal powerplants (which may be closer to 30-50 years). Furthermore, most of your graphs show major deviations between scenarios early in the life of the model. I'd change the approach and the figures (graphs) to emphasize time frames that are more relevant to plant economics.

Line 61: Regarding 100°C as a threshold. On cursory examination, I did not find reference to this number (which seems very low and rarely economic unless the system is

particularly shallow, productive, or in a great market) in the DiPippo volume. Instead, look into Bertani (2005) for some examples of typical producing (and presumably economic) values. Furthermore, I would expect major economic and efficiency loss well before your production temperature declined from 150 to 100°C.

Bertani, R. (2005). World geothermal power generation in the period 2001–2005. Geothermics, 34(6), 651-690. 10.1016/j.geothermics.2005.09.005

Line 66-68: Consider emphasizing BHG instead of all the others.

Line 94-95: The issue with well spacing seems to distract from BHG, until you specifically related the impact of BHG on effective well spacing. The introduction of parameters overall could take more care.

Line 97: Change to 0.047°C/m-1

Line 97: Is a linear gradient throughout justified? In higher permeability systems you may expect isothermal reservoirs.

Line 117-119: You have a high geothermal gradient given limited vertical advection. Perhaps this study really is best described as analogous to hot sedimentary aquifers, rather than more conventional fault-fracture hydrothermal systems? I don't recall seeing this distinction.

Line 127-129: Is the combination of porosity of 14% and a permeability of 10-15m2 realistic?

Line 140: A 7 m wide fault core is quite large. Can you include references to justify this model parameter?

Line 154: Again, the model time of 200 years, while perhaps arbitrary, is not particularly relevant to producing geothermal fields.

Line 193-197: This is an interesting finding, but it is lost in the paper because the structure is not set up as a test of the influence of BHG compared to other parameters

(see lines 66-68). Couching this section in terms of BHG would bring more coherence to the results and discussion.

Line 202: 10-15 m2 seems very low for a sandstone with 14% porosity. Better geologic constraints on parameter space would make the results more defensible (see notes Line 720).

Line 317: There seems to be a disconnect between statement and reference here. I don't think Alava et al. (2009) discuss porosity or permeability, and if it is a different parameter they describe it should perhaps be clearly specified separately instead of grouped with other references.

Line 337: Although bottom hole pressures exceeding lithostatic may not be unreasonable, it is not clear that your model responds to these conditions by fracturing, nor would this condition be favorable (or even permissible) in a permitted injection well. Constraining your model space to geologically reasonable conditions would make the results more useful.

Line 342: Aren't pores and fractures always filled with fluid?

Line 342. "Since pore space often exceeds..." is not needed in this argument, as you say "high porosity" later in the sentence. The "since" statement is distracting, as there are many counter examples.

Line 348. Again, regarding parameter space, if 10-13 m2 is the threshold, why bother with the very low permeability cases?

Line 363-368: This passage makes the point that your models considering BHG are important, but it needs to be expanded, and more rigorously explored and cited (there should be many examples of fields that target outflow zones for reinjection and upflow zones for production). I'd also consider moving a version of this into the introduction when you describe the importance of BHG.

Line 387: Suggest changing "borecore" to "core" or similar.

[Figure]

Line 388: Check "metre" for journal style.

Line 406: "scales" to "scale"

Line 411: I would either cite or change this first statement.

Line 411-424: Another and significant reason there is an interest in fault zones is that fault zones are fundamental parts in many producing geothermal fields because they provide the necessary vertical permeability and advection of heat and fluid so that high temperatures are shallow enough to be economically exploited. I think your passage misses this by focusing on the complexities of faults instead of the constraint that many fields and models will by necessity involve faults.

Line 439-441. This passage is probably not necessary.

Line 445. Although the ranges may be real, the combination of ranges seem less plausible.

Line 472: There is an extra space resulting in a broken link.

Line 648 (Figure 2 g). Please consider a shorter time span and temperature range. The timespan of 200 years and wide range in T (40-180°C) masks the more relevant changes early in the lifespan of a well or geothermal field. Furthermore, smaller drops in temperature would nonetheless have major impacts on plant efficiency. This comment applies even more to your fault-controlled models that show major changes in the first few years.

Line 720 (Figure 10). It would be nice to see these plotted together as x-y, so you could support your use of 14% porosity and low permeability. Because this is described as a more generic model, might it also make sense to show values from other geothermal fields producing in sedimentary basins?

---

## Author Comment (AC2) · 13 Aug 2019

We appreciate the positive evaluation and thank the second reviewer for their comments and suggestions. We will use them to improve our manuscript. Please see our answers to the question raised below:

General comments/ Figures:

Reviewer comment 1.1: Figures in general have a small scale for (small) colored dots and a (uselessly) large scale for the vertical. Furthermore, the colors used are the same. This is misleading the reader. My suggestion is to use different color codes for

the two parameters (depth and BHG) and change the relative dimensions of the two scales, since the focus of the manuscript is on the BHG (color-coded dots).

Authors reply: We follow the reviewer's suggestions and will adjust the size of the legend for the HDI plots. Further, we'll try to find a different colour scale for the HDI.

Reviewer comment 1.2: Figs.2a-c (as well as other corresponding plots) either have inverted y-axis scale (sic!), or I did not understand the figure and/or the text (cfr. lines 161-165). This produced some initial misunderstanding of the work (the text is not properly describing what is presented in the figure).

Authors reply: The y-scales of each scatter plot show the time to thermal breakthrough, i.e., the time at which the production temperature reaches 100°C. There are no depth scales in the scatter plots. There are no inverted scales in any of the figures of our manuscript. We assume, as the reviewer pointed out in 1.1, that this misunderstanding is caused by the fact that the colours for the scatter plots overlap in parts with the colour code used for the figures showing the HDIs. We will try to find a different colour scale for the figures that present the shape of the HDIs. We are aware that the presentation of the results, owed to the multiple parameters we analysed, is somewhat unconventional. We will follow the suggestion to provide a short introduction/explanation on how to read the figures in the Method section (see also point 1.5, 1.6, and 2.8).

Reviewer comment 1.3: In many experiments the temperature stabilizes at around 100_C (Figs. 2j, 6g, 7g, 8d, 8g, 9g). The reason for this coincidence with the HDI not clear or explained. The author should justify this "convergence" in the various models.

Authors reply: The question why in some of our models the temperature converges to 100°C was also raised by reviewer 1 (RC1s point 7.3). The answer is that in these cases it is a coincidence and the result of a complex interplay between the chosen parameters, i.e. thermal gradient, surface temperature, porosity, permeability.

Reviewer comment 1.4: In Fig.1a the projection of the wells provides the impression
that their trajectory is oblique. The Author should either correct the figure or describe the reason for oblique wells as well as quantify it.

Authors reply: The wells are indeed oblique. We will include the parameters in the method section. The reason for the inclined wells is that it allows us to keep the whole well within the damage zone of the fault in Scenario 5, which is also oblique, i.e. it dips. For comparability, we used the inclined wells consequently in all the other models.

Reviewer comment 1.5: In the Figures the Authors should include the number of experiments represented (i.e. the number of dots in the single figure).

Authors reply: The reviewer is correct: The number provided in the figure captions could be misinterpreted. Indeed each of the corresponding plots contains the same results, i.e. the same number of dots. The difference is that in the different plots the same results are presented with different x-axis to explore the impact of the multiple parameters. We will clarify this issue as promised above (point 1.2) using a short introduction in the figure setup in the method section. We will also correct the figure captions, from e.g.: "Plots (a), (b), and (c) contain the results of 225 simulations." to "Plots (a), (b), and (c) each contain the results of the same 225 simulations." We think this is less confusing than presenting in each sub figure the same number of experiments. We hope the reviewer agrees with this option.

Reviewer comment 1.6: The author should discuss the case of a strong variation in the results (e.g. Fig. 1a, red dots for BHG=20 mm/m, at permeability 10-11).

Authors reply: We are convinced that the influence of the BHG is sufficiently discussed in the lines 351 to 368, where we describe that, at high permeabilities, the BHG can outperform the artificially-introduced flow field. We are convinced that the reviewer's question becomes obsolete with the new improved method section that introduces the setup of the scatter plots, i.e. that the scatter plots presented next to each other must be seen in combination and not as stand-alone results. In this particular example cited by the reviewer, the corresponding Figures 2a, b, and c should read as follows. The
thermal breakthroughs of less than 20 years, is according Figure 2a, at permeabilities of 10-11 $m^2$. The same data points in Figure 2b (red dots with lowest thermal breakthrough times) show that this short time to thermal breakthrough is observed for hydraulic gradients that are directed southwards and have a magnitude (colour code red, as shown in the legend) of 20 mm m-1. In Figure 2c, the same data points are plotted according to the porosities used in the models. Here, it shows that, in this case, porosity plays a minor role in determining the expected lifetime of the reservoir. We additionally visualized this connection using horizontal lines that connect results of the same experiments.

Reviewer comment 1.7: Numbers are very small to pretend some statistics (mean, sd), but they could mean unreliable results and should be discussed.

Authors reply: We agree that a further statistical evaluation is futile. We also think that providing parameters like mean and standard deviation might not be the way to further investigate our results, because the present results are from different experiments with varied model parameters. This means, our experiments do not allow such statistics.

Reviewer comment 1.8: The "line connecting the same experiment" is not clear. In Fig. 2 the yellow line connect one yellow dot per figure, and it is easily understood. On the other hand, in other figures (e.g. Fig. 4a-b, green lines) the do connect multiple dots in the same figure. This is confusing: how many numerical experiments were responsible for each dot in each figure (I assumed one)? Maybe they partially overlap.

Authors reply: The reviewer is correct. Each dot in each panel is the result of one numerical experiment, and each of the according panels contains the same model results. Due to the large number of experiments, we unfortunately cannot avoid that, in some cases, the line connecting the dots, which belong to same experiment, also cross other points. In this case (Figure 4a, b), it shows that the varied parameters (direction and magnitude of the BHG) does not influence the time to thermal breakthrough in the modelled time span. Consequently, the dominating parameter that determines the

reservoir's lifetime in the low permeability series is the permeability contrast. Please see also lines 370f and 204f of the discussion paper. Please see also our answer to point 1.6.and 2.8.

Specific comments

Reviewer comment 2.1: In lines 34-39 the Authors discuss the poor improvement in porosity due to the presence of fractures. This is true, but the author are not considering the main role provided by fractures in improving the effective porosity by connecting isolated pores, as is normally achieved in tight-gas reservoirs (gas-shale). In general, the manuscript is not discussing on the difference between total porosity and effective porosity. I guess that the porosity they consider in the numerical experiment is merely the effective one, and this should be clearly mentioned. On the other hand, a brief note on the role of influence of fractures on effective porosity is required to complete the introduction and the discussion paragraphs.

Authors reply: We will now mention the effect of fractures on matrix porosity. The second part of this question address how permeability is implemented in the models. The same issue was raised by reviewer 1 (e.g. point 6.8), it will be answered in the method section and we will clarify that the permeability in the models is not linked to matrix porosity. Instead the continuum approach (Berkowitz et al., 1988; Lege et al., 1996; Kolditz, 1997), which uses replacement media for the fractures and provides mean hydraulic properties of a given fracture system, was utilized.

Reviewer comment 2.2: Equations in Line 72,81,82 seem correct, yet references for the general audience (as Solid Earth also has) are required.

Authors reply: We will follow the reviewer's request and will provide additional literature.

Reviewer comment 2.3: Line 91-92. The limit to 20m is not easy to be understood (e.g. where these 20m were located along the well). Maybe a better way to express this correction would be to express it as a percentage of the well hole surface (the

cylinder), or by presenting the equivalent reduction in permeability between the well cell and the surrounding ones in the mesh.

Authors reply: We thank the reviewer. A similar question was raised by reviewer 1 (point 6.3). We will follow the suggestion of the reviewers and rewrite this part accordingly. Standard well diameters are a few decimetres. This in turn would need a very fine mesh. To avoid this issue we use a larger diameter for the wells. To account for the unrealistic high diameter and thus the area of the "perforated production and injection zone" we choose to adjust the size of the area via its length to a size that is in a realistic range.

Reviewer comment 2.4: Line 96: I guess that the geothermal gradient is in reality expressed by m and not by km...

Authors reply: We thank the reviewer and correct the typo.

Reviewer comment 2.5: Line 105 and 113. I guess that "computational costs" really intends the more appropriate expression "computational time". It would be of interest to the readers to quantitatively justify this sentence: add in lines 69 71 information on the used computer platform and the approximated run-time for a single numerical experiment. Line 126 and through all the experiments.

Authors reply: We will now provide information on the computer platform and the differences in the runtime for the fully- and unidirectional coupled experiments.

Reviewer comment 2.6: My opinion is that a permeability of 10exponential-11 m2 is unfair to be reached in a reservoir at the used resolution of the model, with the exception of karst cavities. The Author might include here a descriptive correspondence to the reservoir permeability (e.g. tight reservoir for 10 exponential -15 m2, medium-high permeable reservoir 10 exponential -13 m2, karst structures 10exponential-15 m2).

Authors reply: Both of the here addressed issues are also made by reviewer 1. The first point regarding the high permeabilities is answered by how permeability and porosity

are implemented in the models. We are now aware that we were not clear enough about this and will now address this issue in the method section to avoid potential misunderstanding. Please see our answer to your point 2.1 and to e.g. point 6.8 by reviewer 1. This second question, which also ties directly in with a request of reviewer 1, is to provide some real world examples. We will improve the Introduction and try to find the best compromise between both suggestions.

Reviewer comment 2.7: Lines 143-147 more references on measure of permeability in fault core are important here (e.g. works by R.J Knipe and/or Q.J Fisher).

Authors reply: We will provide more literature on fault permeability, as also requested by reviewer 1 (point 6.9). We, however, prefer to present them in the introduction instead of placing them in the results. We hope the reviewer finds this compromise acceptable.

Reviewer comment 2.8: Line 161-165. As mentioned, the only way I found to correlate text and Fig2a-c is to invert the Y-axis scale. Anyhow the description, even with this correction, does not correlate for 10exponential-11 permeability experiments, that scatter results all along the entire span 0->200 a apparently without any rule (e.g. red dots). Did I understand properly the figure? If not, a more careful introduction to the figure and description might be necessary.

Authors reply: Please see our answer to your points 1.2 and 1.6. We will provide an introduction on how the plots are to read. In detail, on the example of figure 2a-c. Each of the plots a), b), and c) contains the same model results for all combinations of the three porosities, the three permeabilities, the eight directions of the hydraulic gradient and of the 4 different magnitudes of the BHGs. As the reviewer agrees, these are a lot of parameters to visualize. Whereas this is a typical approach to plot results in multi parameter studies, it is not very common in the geosciences. We decided to keep the y-axis constant, which shows the time to thermal breakthrough. The x-axis was used to plot the same data points in different ways, to produce the patterns that

show the different effects of the different values (permeability, direction of the hydraulic background gradient, porosity). In consequence, it is important to see these plots as a whole, i.e. the value they provide only becomes apparent by looking at them in combination. The connecting line is used to show the above issue and connects the same result of one experiment and allows to identify the parameter values for each individual model run.

Reviewer comment 2.9: In the Figs. the meaning of the represented surface is not completely described. The Authors refer to "HDI shape". I am not sure but I guess that, considering the experiments, these surfaces represent the envelope of the volume where the temperatures become lower than the HDI due to the successful heath extraction. An explanation on the meaning of the HDI shape is required in the text (and maybe in the caption for the fast readers...).

Authors reply: We accept this point and will improve the description of what the HDI actually is. The reviewer is correct with the description. The HDI encloses the volume with temperatures lower than 100°C.

Reviewer comment 2.10: Line 173-174 the probability concept should be better introduced.

Authors reply: We accept the suggestion and will describe in more detail that elongated ellipsoidal HDIs, where their direction is controlled by the HBG, may result in a reduced probability/chance that the injected cold fluid reaches the production well.

Reviewer comment 2.11: Line 178: I guess the Authors intend Fig.2g and not 2e.

Authors reply: We thank the reviewer to point out this mistake. We will also rephrase the according lines for better readability: ...."The two contrasting BHGs in Fig. 2g show, either fast (e), or almost no decrease (f) in production temperature".....

Reviewer comment 2.12: Line 198 "three series". This is not clear: I see in the figure3 different permeability (these are the three series), 4 permeability contrasts and 8

different orientation for BHG with 4 possible gradients, for total of 3x4x8x4=384 combinations. Then just three BHG shapes, but for the same permeability (same series). This might be confusing. A more complete description of the model procedure might help to understand the results.

Authors reply: We will improve the text. In detail in line 203 (discussion paper) we will directly refer to the according figures and model series when introducing the permeabilities. We will also mark in the figures the series to which they belong. With regard to figures 4c, f, and i, we selected exemplarily three HDI shapes from the medium permeability models. Given the 300 individual model runs presented in this scenario, we decided to show these because they constitute a good compromise, i.e. they (1) are in a realistic permeability range and (2) clearly show the effect of the permeability contrasts.

Reviewer comment 2.13: Fig4b is not clear, and in general figs 2, 4, 5 are not easy figures. Same color dots appear both on high and very low times to breakthrough. This could mean the excessive scattering of results, or that results are from experiments with different, not specified, parameters.

Authors reply: Regarding the Figures in general, please see our answer to your points 1.6, and 2.8. Regarding Figure 4b in particular please see our comment on your point 1.8. Figure 4b shows, in combination with Figure 4a, that the only controlling factor in this low permeability case is the permeability contrast in the reservoir, i.e. the orientation and magnitude of the BHG is of no importance. This correlation however is altered (Figure 4d, e and Figure 4 g, h), if the permeability of the reservoir layers increases, i.e. the BHG can, in these cases, compensate the limitations introduced by the permeability contrast. Please see lines 376f and results 3.2 Models of layered reservoirs in the discussion paper. We assure the reviewer that all parameters are specified and can be picked in the figures as described in e.g. point 2.8.

Reviewer comment 2.14: I think to have properly understood the relations between the

dots in Fig. 4a, b and the reason for the limited connection presented in Figs. At the present stage, the figure is very difficult to be understood (also due to the high number of combinations in the experiments – i.e- the number of parameters used - and the limiting 2D of the journal pages. . .). The Author could try to improve the correlations by either using different symbols for each experiment (good luck, it would be a big effort with questionable results) or by adding a reference number to each dot. The diagrams have a relative small number of dots and al lot of empty space. A simpler alternative might be to add in the text the clear description of a correlation among dots as an example. There are also some evident overlap of dots (just comparing among figures) and this should be described (or slightly move one of the dots within the resolution of the results).

Authors reply: We assure the reviewer that we have discussed and tested many options to present the data, including the ones made by the reviewer, i.e., using different symbols and/or adding references. It, however, did not improve the figures, as expected by the reviewer. We found that the way we finally chose, as is common in multi-parameter studies, is probably the best. However, we will provide an additional section in the methods that helps to understand the concept of the figures. See also our answer to your points 1.6, 1.8, 2.8, 2.13 above.

Reviewer comment 2.15: Line 215-220 again: the cited 70 years seems to correspond to 130 years in Fig.4d, second column. Is there again reversed the Y-axis scale?

Authors reply: Here we are writing about the range of observed lifetimes. The range of observed lifetimes, in this case, is indeed about 70 years, i.e. between 130 and 200 years.

Reviewer comment 2.16: Line 235. As previously mentioned. Why at 100_C? This should be justified by the Authors.

Authors reply: Please see our answer to your point 1.3.

Reviewer comment 2.17: Fig.5 the origin of dots on top of the plots a-c (i.e. at >200a) is not clear.

Authors reply: In this, as in many other model runs, the production temperature did not fall below the 100°C threshold. In consequence, we decided to assign the results of expected lifetimes to be at least 200 years, i.e. the time modelled. On the example of Figure 5. This the only model configuration that allowed a hydraulic connection between the wells because the fracture anisotropy is parallel to the well alignment. In the other cases, the fracture anisotropy, even the lowest, hinders the reinjected cold fluid reaching the production well, and consequently, under the applied model setup, the temperature stays above the threshold.

Reviewer comment 2.18: Line 264-265 Fig. 5g shows that temperatures stabilize at 100_C. How this happens at exactly the critical temperature chosen for the HDI? Is this input in the model? Some explanation is needed.

Authors reply: Please see our answer to your point 1.3.

Reviewer comment 2.19: Lines 340-349 Here is perhaps the proper space to discuss the total porosity and the effective one I discussed above. As I understand, the chosen porosity is intended to be 100% effective. A sentence explaining this should be anyhow added to the article.

Authors reply: Please see also our answer to point 6.8 from reviewer 1 and your point 2.1.

Reviewer comment 2.20: Line 371. This assumption may be too forced, and I am sorry for the referenced articles. Secondary fractures and faulting allow permeability to take over thinner clay layers that lose their sealing property. This is more difficult in thicker clay layers. I understand that in the useful proposed model are necessary simplifications, but it is not the case for the complexity of real geothermal reservoirs.

Authors reply: We acknowledge that the reviewer agrees that this is an unfortunate but

necessary restriction of the models. This is however, the concept of our study, i.e. to use simplified models and to show that, even with these simplifications, predictability of the modelled systems is extremely complex. This agrees also with the reviewer's point 2.27 that our manuscript presents a first step in pointing out these difficulties. We agree, of course, with the reviewer that fractures also propagate through "sealing" layers. Even though such softer layers hinder fracture propagation. For this reason we wrote,"restrict fluid flow across them" to avoid a too strong statement. We will additionally rephrase the sentence in line 371 (discussion paper) to: "(Sub)horizontal permeability contrasts can be caused by layering in sedimentary rocks and can span several orders of magnitude (Zhang, 2013), even though these sealing properties are altered or reduced by barren fractures." We hope this is an acceptable compromise.

Reviewer comment 2.21: Line 399. I do not see evidence in Fig. 5b to justify this sentence. At my sight, the resulting timings are fully independent from the BHG values (colored dots). May be the Authors are referring here on the BHG orientation of Fig.5c. .

Authors reply: The reviewer is correct. Indeed, we refer here to Figure 5a, b, and c. With this correction, the statement in line 399 is justified. The sentence in Line 399-400 will be rephrased to: "Second, fracture anisotropy in the range of 1 order of magnitude, with respect to the bulk permeability, leads to either very short- or long-lived geothermal reservoirs, depending on the BHG properties and the orientation of fracture anisotropy (Fig. 5a, b, c)."

Reviewer comment 2.22: Line 423-414 Fractures and secondary faulting associated to faults have generally various angles to the faults and only a minority lies parallel to it (cfr. Riedel). This results in: fracture intersections, fracture opening by the stress induced from the kinematics along the fault (friction). These factors guarantee the higher permeability of fault damage zone to a certain extent, as described in the literature. To be explicit: "often-observed" of "fault-parallel fracture anisotropy" does not correspond to either field outcrops and cores across fault zones, apart from S-C structures, where

in any case C planes are generally subordered in number to S ones, My suggestion is simply to eliminate the "often-observed" attribute.

Authors reply: We accept the reviewer's comment that our statement is eventually too strong. However, following the suggestion to delete "often-observed" makes the statement much stronger. We will modify the sentence to: "This typical characteristic of fault zones thus increases the chance of good hydraulic connection between injection- and production wells and is potentially further improved by fracture anisotropy in the damage zone, which is often (sub)parallel to the fault."

Reviewer comment 2.23: Line 429 The previous concept is repeated here: useless redundancy and same comment.

Authors reply: We do not fully agree with the reviewer. In the lines 411 to 425, we discuss why faults have become recently prime targets in geothermics and the difficulties that have been reported. In lines 425 to 440, we discuss the results of our models and how they agree or disagree with common knowledge. We will slightly modify the sentence in line 425-426 to: "Our simplified models support these findings and show that faults, with damage zones that constitute positive permeability contrasts of just 2 orders of magnitude, exhibit these channelling effects (Fig. 6)." to make the structure of this section clearer.

Reviewer comment 2.24: Line434-435 the use of the terms "opposed/opposite" to indicate opposite (!) dipping is misleading. A rephrase would solve it.

Authors reply: We accept the reviewer's suggestion and will rephrase the sentence accordingly to: "We observed that, when the BHG is oriented against the dip direction of a fault, the fault can be considered a more sustainable target for geothermal exploitation than a fault with a BHG oriented in dip direction (Figs. 7e, f, 8e, f)."

Reviewer comment 2.25: Lines 62, 442: they were 1027 (from line 150). This is an interesting and serious number of runs and it would be effective to remark this number

both in the introduction (say, "over one thousand numerical experiments") as well in the Conclusions "(1027)". My impression is that "large series" or "a series of" would be –alas – interpreted as much smaller number in present-day publish-or-perish scientific environment.

Authors reply: We very much appreciate that the reviewer values our work and we will stronger pronounce the number of experiments carried out by us.

Reviewer comment 2.26: Line 457: This is not so simple. This sentence does not take in consideration the improvement of the effective porosity that is induced by fracturing that in turn may be enhanced by the oriented stress that develops in presence of strong BHG. Since the point about effective porosity changes is not taken into consideration in the presented models, my suggestion is to specify this in the sentence (referring to "in many cases" might be not sufficient).

Authors reply: The reviewer is correct that effective porosity is, in many cases, improved by fracturing. However, the reviewer accepts also that an investigation of this effect is not part of our experiments. We, here, refer to our model results that show that the positive effect of porosity has on heat capacity and thus on the reservoir lifetime is minor, compared to that of permeability and BHG. We will rephrase the sentence accordingly to ", in many cases, the positive effect of porosity has on heat capacity and thus on the reservoir lifetime, is minor, compared to that of permeability and BHG....". We hope that the reviewer can accept this solution. Please see also our answer to point 2.1, in which we explain how porosity and permeability are implemented in the models.

Reviewer comment 2.27: Line 459-462. On the contrary, results from this work well represent the first step to model real, complex geothermal reservoirs with their Stochastic modelling by adding in the mesh the proper random values! And I am sure that the "computational costs" at that stage will be an insignificant obstacle. This might be a further point and a better conclusion to your article (follow the Hollywood-movie style:

end always your articles with a true, positive sentence on your results. . .).

Authors reply: We thank the reviewer for the positive evaluation and welcome the suggestion to extend the conclusions and end the manuscript as proposed with a positive outlook, i.e., how our findings help to improve geothermal exploration in the future. We will include at the end of our manuscript: "Our results show that realistic site-specific models are difficult to achieve, because parameters, such as permeability structure and BHG, are often poorly constrained but can have unforeseeable large effects on the lifetime of geothermal systems. Thus our findings provide an important step forward to judge which parameters must be known to which degree to make site specific models as reliable and accurate as possible in the future, by implementing the controlling parameters in advanced stochastic models."

———————————————————

---

## Author Comment (AC3) · 8 Sep 2019

**Referee #3 Owen Callahan**

Dear Owen Callahan,

We thank you for your review. In the following response, we will answer the questions and concerns raised by you.

| | General comments: | |
|---|---|---|
| | | |
| 1.1 | Model parameters, including selection of porosity-permeability combinations, length of model duration, selection of 100 C isotherm, are not sufficiently justified, and may not be relevant to operating geothermal fields. | We consider feasible ranges of all parameters, including values below and above typical benchmark parameters. This is prerequisite for a sensitivity study. From the combination of all the results, we determine particular parameters and their values that exert control on the geothermal reservoir. The necessity of this approach was provided and explained by us in our response to R1 point 3.3.

 Porosity and Permeability
 In accordance to R1 and R2, we have now improved the introduction and introduce the values for permeability, porosity earlier in the text. We discuss how these compare to typical values in different geothermal settings. We now explain better the modelling approach that we use.

 100°C and threshold
 See also points 2.2, 2.10, 2.25:
 A universally applicable (economic) threshold cannot exist, because of the different site-specific demands of geothermal power plants e.g., district heating, electricity generation, output, depth of the reservoir. The 100°C isotherm or the 100°C threshold must be arbitrary, at least to some degree, even not taking into account that it must somehow balance with the model duration. We chose 100°C because it is sometimes referred to as minimum temperature that allows electricity production with |

| | | |
|---|---|---|
| | | binary cycles (e.g., Bhatia, 2014; Buness et al., 2010; Erec, 2004; Huenges, 2010; Mergner et al., 2012). We have rephrased the sentence and improved references.

Model duration
points 2.9 and 2.18:
There are two points to be made here. Firstly, we do not investigate the lifespan of hydrothermal power plants, but rather the role of individual reservoir parameters on the thermal development of geothermal reservoirs (see line 53-54 SED). Secondly, there is a balance between threshold temperature and duration of the model. If, as requested by the reviewer, we had chosen 150°C, then the effect of the parameters on the thermal lifetime, would be less clearly shown. If we, in addition, had only run the models for 40-50 years, then the majority of the model runs would not reach the important/higher threshold, i.e. there would be no data to show. For instance in scenario 1 (Fig. 2a, b, c) and 2 (Fig.4a, b, d, e, g, h) we could not identify the impact of the different parameters.

We strongly disagree that this study is not relevant to operating fields. The model results give an indication of the importance of different hydrogeological parameters on the lifetime of geothermal reservoirs, and even if the modelled lifetime of the reservoir exceeds the lifetime of a geothermal power plant, the relative importance of different parameters remains the same. |
| 1.2 | Use of references and citations is inconsistent. In some cases, statements with long lists of references are too vague to be useful (i.e. not clearly tied to particular geothermal fields or a specific type of inquiry (numerical, field, experimental: : :)) and in other cases the listed references do not seem appropriate for citing in their current context. | We feel that there is some room for improvement. However, we disagree that the references are too vague.
See our comments to your detailed criticism below. |
| 1.3 | The structure of the paper fails to emphasize the role of BHG nor does it discuss enough real world scenarios were the impact of BHG, or even suspected impact of BHG, can be shown. As it stands, | We strongly disagree. The results, the discussion section, and the conclusions contain a sufficient information about the BHG; in our opinion, balanced together with the other parameters. The only part that can be improved with regards to the BHG is the introduction. This was our answer to the reviews by R1 and R2. |

| | almost all of the conclusions are about BHG, but BHG only gets 3 lines in the introduction. | That we cannot discuss the effect of BHG for a large amount of real world scenarios is because of, to our best knowledge, the lack of data and case studies in the literature. BHG has not been considered in equivalent studies in literature before our paper.
 We appreciate that the reviewer likes our findings regarding the BHG. We will improve the introduction concerning the BHG. We, however, disagree that almost all of the conclusions are about the BHG, all of the conclusion points emphasize the important role of permeability and permeability heterogeneity as well. The BHG - even though you agree that this is an important parameter- is an underestimated parameter. It is, however, still just one of the parameters that we investigated in our manuscript, and its ranking in the modelling, as a whole, needs to be understood. |

**Specific comments**

| | | |
|---|---|---|
| 2.1 | Line 11: This sentence neglects economic factors. Rather than "can be exploited" maybe describe geologic factors influencing economic viability, as you do in the introduction. | Our manuscript considers geological reasons for geothermal lifetime. Our manuscript does not focus on economic factors and we have therefore removed any reference to this subject in the introduction. |
| 2.2 | Line 17: 100°C isotherm is not well justified. See additional comments below. | This is the abstract of our manuscript, we here report solely the threshold we use and are convinced that any justification of the 100°C isotherm at this place would be misplaced.
See also point 1.1. |
| 2.3 | Line 29: The first few lines of this paragraph make it seem like these references pertain to hydrothermal settings specifically. In this current configuration, Laubach et al. (2009) does not seem like an appropriate reference as they do not describe fracture patterns in hydrothermal systems, nor do they explicitly describe the impact of fractures on permeability or volume (other than tangentially) but rather compare fracture and mechanical stratigraphy. | This line introduces the difficulties to predict reservoir properties. This is the case of permeability. Permeability is commonly provided by fractures. In Laubach et al. (2009), they point out that fracture patterns are difficult to predict (and therefore also permeability). Thus, we are convinced that citing Laubach et. al., 2009 here is justified.

For instance, Laubach et al. (2009) wrote:
*"In subsurface studies, current mechanical stratigraphy is generally measurable, but because of inherent limitations of sampling, fracture stratigraphy is commonly incompletely known. To accurately predict fractures in diagenetically and structurally complex settings, we need to use evidence of loading and mechanical property history as well as current mechanical states."* |
| 2.4 | Line 32: Manning and Ingebrigtsen (1999) concerns theoretical permeability at the crustal scale and in metamorphic rocks in particular. The link between this reference and the statement are again tenuous unless more clearly explained. | Here we write that permeability and porosity in general are rock properties that are highly heterogeneous, independent of rock type.

Manning and Ingebritsen (1999) wrote:
*"Near the Earth's surface, permeability exhibits extreme spatial variability (heterogeneity) and anisotropy, both among geologic units and within particular units".*
Thus, in our opinion, the reference is justified.

However, we will rephrase the sentence accordingly: |

| | | They are, independent of rock type, often highly heterogeneous because of layering, localized fracturing, and diagenesis (e.g., Aragón-Aguilar et al., 2017; De Marsily, 1986; Lee and Farmer, 1993; Manning and Ingebritsen, 1999; Zhang, 2013). |
|---|---|---|
| **2.5** | 2.5.1 Line 37-39: The logic here is odd. You describe high porosity in sedimentary geothermal systems, then say fracture porosity in sedimentary rocks is low (are dam sites really the best analog, i.e. Snow, 1968?), but that fractures dominate geothermal systems.

2.5.2. Separately these statements may all be true, but fractures commonly dominate in geothermal systems because geothermal systems are commonly not hosted in sedimentary rocks.

2.5.3. Also, you may want to specify "clastic" sedimentary reservoirs, as fractures can be very significant contributors in carbonate rocks. | 2.5.1: We do not understand the logic of the reviewer's comment. We wrote that fractures have a dominant control on rock permeability in geothermal reservoirs, even though their contribution to bulk porosity is negligible compared to matrix porosity. We are convinced that we have communicated this correctly in our discussion paper.

*Snow (1968) is highly appropriate in this case, because*
    *(1) Snow (1968) analysed fracture porosity in different rock types,*
    *(2) The fact Snow (1968) used outcrops at dam sites is highly relevant here, because the known permeability at the dam sites were an asset to calculate the fracture porosity.*

2.5.2: We strongly disagree that geothermal systems are commonly not hosted in sedimentary rocks. There is a large number of examples worldwide for geothermal systems in sedimentary rocks (see Moeck, 2014). We also strongly disagree with the reviewer's point that fractures only play a minor role in geothermal systems hosted in sediments. For instance in the Upper Rhine Graben, the permeability is controlled by fractures in lithified sedimentary rocks (Meixner et al., 2014; Egert et al., 2018).

2.5.3: We disagree. This study could be used for both clastic and carbonate reservoirs. The range of parameters used in this study covers both cases. Fractures dominate most deep geothermal systems. |
| **2.6** | Line 42-43: The statement about specific failures needs referencing. | We are aware of this issue. However, this is tricky, since such negative examples are commonly not published in scientific literature. In our experience links to webpages on failed projects disappeared over time. Nevertheless, we will provide links to websites, if possible. |
| **2.7** | Line 45: Beall (1994) does not appear to be about declines in production fluids nor fault damage zones, but rather to be about tracer tests and what can be learned about fluid saturations. | We have deleted the reference to Beall (1994). |

| 2.8 | Line 48-50: BHG is a huge part of your overall paper but has a tiny role in the introduction. This should be much larger, with specific examples of where it has impacted production. It could be your primary hypothesis and seems like the major contribution, but it is not firmly established in the introduction. As it stands, the introduction does not lay the necessary foundation for the paper, not establish a clear hypothesis, but it could be reworded to emphasis BHG (see comments about 363-368). | Please see our answer to your general comment (point 1.3) above.

In accordance to R1 and R2, we have improved the introduction regarding the BHG. However, we are convinced that, even though the BGH is important, that the introduction as well as the other parts of the manuscript should remain balanced regarding the investigated parameters.

We strongly disagree that the aim of our manuscript was not sufficiently communicated.
In lines 51-56 (discussion paper), we did provide the objective of our manuscript.

We, as requested by R1 and R2, have modified this part and included more details. |
|---|---|---|
| 2.9 | Line 58: The lifespan of 200 years in not well justified. This is longer than the nominal lifespan of geothermal powerplants (which may be closer to 30-50 years). Furthermore, most of your graphs show major deviations between scenarios early in the life of the model. I'd change the approach and the figures (graphs) to emphasize time frames that are more relevant to plant economics. | See our answer to point 1.1. |
| 2.10 | Line 61: Regarding 100°C as a threshold. On cursory examination, I did not find reference to this number (which seems very low and rarely economic unless the system is particularly shallow, productive, or in a great market) in the DiPippo volume. Instead, look into Bertani (2005) for some examples of typical producing (and presumably economic) values. Furthermore, I would expect major economic and | We do not consider efficiency loss. We also do not carry out an economic feasibility study; see our objective. In our introduction (discussion paper), we communicate that we carried out a sensitivity study in which we investigate the influence of petrophysical and other parameters on the thermal development of geothermal reservoirs. Thus, the points addressed by the reviewer are not the focus of our manuscript.
See our answer to point 1.1. |

| | | |
|---|---|---|
| | efficiency loss well before your production temperature declined from 150 to 100°C. Bertani, R. (2005). World geothermal power generation in the period 2001–2005. Geothermics, 34(6), 651-690. 10.1016/j.geothermics.2005.09.00_ | |
| **2.11** | Line 66-68: Consider emphasizing BHG instead of all the others. | We have, according to the comments by R1 and R2, rewritten the last section of the introduction, with focus on the objective of our study. In our opinion, the presentation and discussion of the results is well balanced concerning the different investigated parameters. |
| **2.12** | Line 94-95: The issue with well spacing seems to distract from BHG, until you specifically related the impact of BHG on effective well spacing. The introduction of parameters overall could take more care. | With our manuscript, we do not concentrate solely on the BHG. We present a sensitivity study in which we examine different parameters for their importance. One is well spacing. No changes needed.
Again, the reviewer draws all the attention to the BHG. In addition, the effect of the BHG on well distance is made. We described it in short but appropriately and with the possible details in lines 187 – 197 (discussion paper). |
| **2.13** | Line 97: Change to 0.047C/m-1 | We thank the reviewer and corrected the typo. |
| **2.14** | Line 97: Is a linear gradient throughout justified? In higher permeability systems you may expect isothermal reservoirs. | Numerous studies have shown that a linear geothermal gradient is a good first order approximation for temperatures that are determined by heat conduction only. The initial temperatures in the model represent temperatures that are undisturbed by fluid flow, and therefore can be represented by a linear geothermal gradient. In some high permeability systems, thermal convection or topography-driven flow could affect background temperatures to an unknown degree. However, the focus of the paper was to explore the effect of induced fluid flow between the injection and production well on subsurface temperatures. Using a different initial geothermal gradient for different parameter sets would make it difficult to compare the different model runs. |
| **2.15** | Line 117-119: You have a high geothermal gradient given limited vertical advection. Perhaps this study really is best described as analogous to hot sedimentary aquifers, rather than more conventional fault-fracture hydrothermal systems? I don't recall seeing this distinction. | Our model scenarios describe both situations, i.e., we have model runs for geothermal reservoirs with fracture anisotropy, faults and for layered sedimentary aquifers. The geothermal gradient that we use is relatively high, but not unusual. The reason for not varying the geothermal gradient for the different model scenarios is discussed in the reply to the previous point. |

| | | |
|---|---|---|
| **2.16** | Line 127-129: Is the combination of porosity of 14% and a permeability of 10-15m2 realistic? | To carry out a sensitivity study, we also need to combine different parameter values, even if they are sometimes unrealistic. This is inevitable in a one at a time sensitivity study. The base case value of a permeability of 10-13 m2 and a porosity of 14% is certainly realistic. We did not consider co-varying porosity and permeability in our sensitivity study.
We understood also from the comments from R1 and R2 that we had to improve our methods section regarding this matter. We have now modified it and describe how permeability and porosity is implemented. |
| **2.17** | Line 140: A 7 m wide fault core is quite large. Can you include references to justify this model parameter? | 7m is wide, but not unusual; see for instance Childs et al. (2009). Furthermore, we chose to model the fault core with this thickness to avoid the high computational cost of very fine meshes. At any rate, the thickness of the fault core is somewhat irrelevant, because the fault core was modeled as an impermeable unit. |
| **2.18** | Line 154: Again, the model time of 200 years, while perhaps arbitrary, is not particularly relevant to producing geothermal fields. | Please see our answer to point 1.1 |
| **2.19** | Line 193-197: This is an interesting finding, but it is lost in the paper because the structure is not set up as a test of the influence of BHG compared to other parameters (see lines 66-68). Couching this section in terms of BHG would bring more coherence to the results and discussion. | We thank the reviewer.
We investigated many more parameters and we do not agree that this point is lost. Instead, we feel that our manuscript is well balanced when discussing the contributions of BHG but also the other parameters that were included in the sensitivity study.
see also point 2.11 |
| **2.20** | Line 202: 10-15 m2 seems very low for a sandstone with 14% porosity. Better geologic constraints on parameter space would make the results more defensible (see notes Line 720). | Please see our answers to points 1.1 and 2.16. |
| **2.21** | Line 317: There seems to be a disconnect between statement and reference here. I don't think Alava et al. (2009) discuss porosity or permeability, and if it is a different parameter they describe it should | We have rephrased this part accordingly.
Instead of:
The variability of these and other petrophysical parameters increases with scale (Alava et al., 2009; Freudenthal, 1968; Krumbholz et al., 2014a).
We now write: |

| | | |
|---|---|---|
| | perhaps be clearly specified separately instead of grouped with other references. | The variability of these (Freudenthal, 1968; Krumbholz et al., 2014a) and other (petro)physical parameters (Alava et al., 2009; Lobo-Guerrero and Vallejo, 2006) increases with scale. |
| **2.22** | Line 337: Although bottom hole pressures exceeding lithostatic may not be unreasonable, it is not clear that your model responds to these conditions by fracturing, nor would this condition be favorable (or even permissible) in a permitted injection well. Constraining your model space to geologically reasonable conditions would make the results more useful. | The model does not include any fracturing, i.e., lithostatic pore pressures only affect fluid flow, and not permeability or porosity.
Constraining the parameter space to sub-lithostatic pore pressures would result in a loss of information, because either parameters would have to be varied together (i.e., adjusting injection rate along with permeability) which would make it much more difficult to compare models and to isolate the effect of a single parameter |
| **2.23** | Line 342: Aren't pores and fractures always filled with fluid? | We agree with the reviewer and will delete "commonly" in Line 342. |
| **2.24** | Line 342. "Since pore space often exceeds: : :" is not needed in this argument, as you say "high porosity" later in the sentence. The "since" statement is distracting, as there are many counter examples. | We will rephrase the sentence. |
| **2.25** | Line 348. Again, regarding parameter space, if 10-13 m2 is the threshold, why bother with the very low permeability cases? | See our answer to point 1.1. |
| **2.26** | Line 363-368: This passage makes the point that your models considering BHG are important, but it needs to be expanded, and more rigorously explored and cited (there should be many examples of fields that target outflow zones for reinjection and upflow zones for production). I'd also consider moving a version of this into the introduction when you describe the importance of BHG. | Regarding the many examples: we are not aware of many published examples of geothermal fields that discuss or report BHG. See also our answer to point 1.3. However, we will improve the introduction regarding the BHG. The BHG, as our study shows, cannot be analysed or ranked as a standalone parameter, it must be seen in combination with other parameters. |
| **2.27** | Line 388: Check "metre" for journal style. | We used British style English throughout the manuscript, as allowed by the Journal. |
| **2.28** | Line 406: "scales" to "scale" | Done |

| | | |
|---|---|---|
| **2.29** | Line 411: I would either cite or change this first statement. | The statement is, in our opinion, sufficiently referenced after the following sentence. See Line 412-413 in discussion paper. |
| **2.30** | Line 411-424: Another and significant reason there is an interest in fault zones is that fault zones are fundamental parts in many producing geothermal fields because they provide the necessary vertical permeability and advection of heat and fluid so that high temperatures are shallow enough to be economically exploited. I think your passage misses this by focusing on the complexities of faults instead of the constraint that many fields and models will by necessity involve faults. | We thank the reviewer for the suggestion and will add a statement about faults as thermal anomalies. However, we consider the effects of the fault on the reservoir itself and do not consider the possible thermal anomaly that allows for a shallower exploitation (see line 117-118 SED). |
| **2.31** | Line 439-441. This passage is probably not necessary. | We disagree, we think it is important to discuss or least mention the restriction of our study. |
| **2.32** | Line 445. Although the ranges may be real, the combination of ranges seem less plausible. | The combination of ranges may seem less plausible, but this combination was necessary to see the effects of individual parameters, and is a standard approach in one-at-a-time sensitivity analysis. See point 1.1. |
| **2.33** | Line 472: There is an extra space resulting in a broken link. | Corrected |
| **2.34** | Line 648 (Figure 2 g). Please consider a shorter time span and temperature range. The timespan of 200 years and wide range in T (40-180°C) masks the more relevant changes early in the lifespan of a well or geothermal field. Furthermore, smaller drops in temperature would nonetheless have major impacts on plant efficiency. This comment applies even more to your fault-controlled models that show major changes in the first few years. | See our answer to point 1.1. |

| 2.35 | Line 720 (Figure 10). It would be nice to see these plotted together as x-y, so you could support your use of 14% porosity and low permeability. Because this is described as a more generic model, might it also make sense to show values from other geothermal fields producing in sedimentary basins? | This is not possible, because the data are derived from several publications. Same region, but different places. In addition, most of the data are not linked (with the exception of Bauer et. al. (2017)). The purpose of this figure is to show just how variable rock properties are. See point 1.1. |
|---|---|---|

**References:**

Alava, M. J., Nukala, P. K. V. V., and Zapperi, S.: Size effects in statistical fracture, J. Phys. D: Appl. Phys., 42, 21, https://doi.org/10.1088/0022-3727/42/21/214012, 2009.

Aragón-Aguilar, A., Izquierdo-Montalvo, G., López-Blanco, S., and Arellano-Gómez, V.: Analysis of heterogeneous characteristics in a geothermal area with low permeability and high temperature, Geosci. Front., 8, 1039–1050, https://doi.org/10.1016/j.gsf.2016.10.007, 2017.

Bauer, J. F., Krumbholz, M., Meier, S., and Tanner, D. C.: Predictability of properties of a fractured geothermal reservoir: The opportunities and limitations of an outcrop analogue study, Geothermal Energy, 5, 24, https://doi.org/10.1186/s40517-017-485 0081-0, 2017.

Beall, J. J., Adams, M. C., and Hirtz, P. N.: R-13 tracing of injection in The Geysers, Geothermal Resources Council Transactions, 18, 151–159, 1994.

Bhatia, S. C.: Geothermal power generation, in: Advanced Renewable Energy Systems, edited by: Bathia, S. C., Woodhead Publishing India, 334-388, https://doi.org/10.1016/B978-1-78242-269-3.50014-0, 2014.

Buness, H., Hartmann, H., Rumpel, H. M., Beilecke, T., Musmann, P., Schulz, R.: Seismic Exploration of Deep Hydrogeothermal Reservoirs in Germany. Expanded Abstracts, World Geothermal Congress, 2010.

Childs, C., Manzocchi, T., Walsh, J. J., Bonson, C. G., Nicol, A., Schöpfer, M-P.J.: A geometric model of fault zone and fault rock thickness variations. Journal of Structural Geology, 31, 117-127, https://doi.org/10.1016/j.jsg.2008.08.009, 2009.

De Marsily, G.: Quantitative hydrogeology: Groundwater hydrology for engineers, 1st ed., Acad. Press, New York, 434 pp., 1986.

Egert, R., Seithel, R., Kohl, T., Stober, I.: Triaxial testing and hydraulic–mechanical modeling of sandstone reservoir rock in the Upper Rhine Graben, Geothermal Energy, 6, 23, https://doi.org/10.1186/s40517-018-0109-0, 2018.

EREC (European Renewable Energy Council): Renewable Energy in Europe: Building Markets and Capacity, 1st ed., James and James, London, UK, 202 pp., 2004.

Freudenthal, A. M.: Statistical approach to brittle fracture, in: Fracture, an advanced treatise, 2nd ed., edited by: Liebowitz, H., Academic Press, 591–619, 1968.

Huenges, E.: Geothermal energy systems: Exploration, development, and utilization, 1st ed., Wiley-VCH Verlag GmbH & Co. KGaA, Weinheim, 463 pp., 2010.

Krumbholz, M., Hieronymus, C. F., Burchardt, S., Troll, V. R., Tanner, D. C., and Friese, N.: Weibull-distributed dyke thickness reflects probabilistic character of host-rock strength, Nat. Commun., 5, 3272, https://doi.org/10.1038/ncomms4272, 2014.

Laubach, S. E., Olson, J. E., and Gross, M. R.: Mechanical and fracture stratigraphy, AAPG Bull., 93, 1413–1426, https://doi.org/10.1306/07270909094, 2009.

Lee, C.H., Farmer, I.: Fluid flow in discontinuous rocks. Chapman and Hall, London, 1993.

Lobo-Guerrero, S. and Vallejo, L.E.: Application of weibull statistics to the tensile strength of rock aggregates, Journal of Geothechnical and Geoenvironmental Engineering, 132, 6, https://doi.org/10.1061/(ASCE)1090-0241(2006)132:6(786), 2006

Manning, C. E., and Ingebritsen, S. E.: Permeability of the continental crust: Implications of geothermal data and metamorphic systems, Rev. Geophys., 37, 127–150, https://doi.org/10.1029/1998RG900002, 1999.

Meixner, J., Schill, E., Gaucher, E., Kohl. T.: Inferring the in situ stress regime in deep sediments: an example from the Bruchsal geothermal site. Geothermal Energy, 2, 7, https://doi.org/10.1186/s40517-014-0007-z, 2014.

Mergner, H., Eggeling, L., Kölbel, T., Münch, W., Genter, A.: Geothermische Stromerzeugung: Bruchsal und Soultz-sous-Forêts, mining+geo, 4, 2012.

Moeck, I. S.: Catalog of geothermal play types based on geologic controls, Renewable Sustainable Energy Rev., 37, 867–882, https://doi.org/10.1016/j.rser.2014.05.032, 2014.

Snow, D. T.: Rock fracture spacings, openings, and porosities, Journal of the Soil Mechanics and Foundations Division, 94, 73–92, 1968.Zhang, L.: Aspects of rock permeability, Frontiers of Structural and Civil Engineering, 7, 102–116, https://doi.org/10.1007/s11709-013-0201-2, 2013.

Zhang, L.: Aspects of rock permeability, Frontiers of Structural and Civil Engineering, 7, 102–116, https://doi.org/10.1007/s11709-013-0201-2, 2013.

---

## Author Response (AR1)

**General Comments RC1:**

| 1. | Manuscript structure: | | Line numbers refer to the annotated version of the revised manuscript. |
|---|---|---|---|
| 1.1: | For any generic numerical study, appropriate input parameters and real-world analogs are important. I would therefore recommend to merge the first part of the "Discussion" (lines 312-327) with the "Introduction" and to move or even repeat some parts in the "Methods" section, in particular the "Scenarios" section. The reader of the manuscript would greatly benefit from a direct real world example for the chosen permeabilities, porosities and in particular background hydraulic gradients (BHG) right in the "Methods" section. | We agree with the reviewer and will in agreement with the comments of reviewer 2 improve the Introduction regarding this matter. We will now introduce the values we used for permeability, porosity and BHG in the Introduction and discuss how these compare to typical values for sedimentary basins worldwide. With our manuscript, we, however, present a non-site specific, numerical sensitivity study that investigates the influence of various reservoir parameters on geothermal reservoir lifetimes and how exactly they have to be known to provide reliable estimates on the lifetime of a geothermal reservoir. For this reason, we did not only chose parameter values for permeability and porosity that are desired in geothermics, but also values that lie above and below them. Since our sensitivity study is not site specific, we only present real world scenarios in the discussion (see also point 8.1). This is why we prefer to keep the current structure of our manuscript. | Changes made in lines 58-86. To accommodate the changes suggested all the reviewers, we now decided to present permeabilities and porosities of geothermal reservoir rocks in general. |
| 1.2 | Especially, the various BHGs require some geological scenarios (what can cause a directed BHG? Topography, overpressure, ...?). Also, the authors might consider merging the entire discussion with the results section for better readability. | We agree with the reviewer and see the necessity to explain in more detail in the introduction why we choose to investigate the influence of the BHG and provide examples of settings in which BHGs are to expect or in which they have been observed. We will also justify the values we used for the BHG's magnitude, and refer to the according literature e.g. Fan et al. (2013, Science), Gleeson et al. (2016, NGS) and Grauls (1999). We, however, prefer, regarding the second point addressed here, to keep the results and the discussion sections separate. | Changes made in lines 58-66. |
| 2. | Convection: | | |

| 2.1 | Convection is not considered in the numerical modelling to save computational cost. As the authors state correctly, convection is likely to be neglected in sediment layers. However, in fault zone-controlled reservoirs, convection is known to have a big impact on the initial temperature field (e.g. Soultz-sous-Forets). Please at least discuss the possible impact of convection on this study's results related to fault zones or consider running a few models that account for convective flow to highlight the impact. | We agree that faults/fault zones can have significant heat flow by density-driven convection. We are also aware, as the reviewer states, that there are several real world examples of faults in which free convection has been observed and we will include this fact in the discussion. However, in many scenarios it is also likely that, due to the heterogeneous nature of faults, convection is not present. However, there a very few published examples in the literature. Our models likely underestimate the lifetime of fault-related reservoirs, because they do not include density-driven convection and thus heat supply from deeper levels. The effect of density-driven convection, however, at least to a certain degree, would be to counteract the negative influence of the channeling effect of a fault (**see also points 7.4 and 7.6).** Regarding the reviewer's suggestion to rerun these models with density-driven convection, would mean that these scenarios are not comparable anymore with the other parts of our study. We will follow the reviewer's suggestion and will now address the possible effects of density driven convection in the discussion section. | Changes made in lines 142-144 and 534-538. |
| :--- | :--- | :--- | :--- |
| **3.** | **Bottomhole pressure (BHP) and flow rate:** | | |
| 3.1 | The authors work with a fixed flow rate, which for the low and medium permeability scenarios results in impossible bottomhole pressures well above the lithostatic stress. | The reviewer is correct that the pressures for the low permeability scenarios are extremely high or even impossible. However, since we chose for our numerical sensitivity study to investigate the impact of a range of parameters and parameter values (**point 1.1**), it is inevitable that some of the combinations represent unrealistic scenarios. These results are netherdeless part of our study and as such help to draw the picture and to understand the effect of the investigated parameters within geothermal reservoirs. Without them, some of the effects would not have been identified by us. In consequence we are convinced that they constitute an integral part and should not be rejected. We hope the reviewer can agree with this and is also referred to our answer to **point 3.6.** We also wish to note that in case of the medium permeability model, the high pressure could be easily corrected in the model by changing i.e. the depth of the well or the reservoir, or the borehole diameter (see line 334-339 of our manuscript). | Changes made in lines 159-164 and 229-235. |
| 3.2 | Nevertheless, this is only mentioned briefly at the end of the manuscript. Here the authors also state that in these cases "the BHG is outperformed by the artificial flow field | We agree with the reviewer: we will add in the method section that artificial flow field and BHG interact. That some of the models return unrealistic BHP, i.e. represent unrealistic scenarios, will be mentioned in the introduction and the method sections. | Changes made in lines 159-164 and 229-235. |

| | | | |
|---|---|---|---|
| | caused by the very high bottomhole pressure". This has to be mentioned directly in the "Methods" section. The actual value of the low and medium permeability models has to be questioned. | Please see also our reply to your **point 3.1**. Regarding the medium permeability values, please see line 334 - 339 (discussion paper). | |
| **3.3** | The BHG appears to be one of the main drivers, but it is completely overruled by the impossible BHPs in the low and possibly also medium perm-scenarios. In that way, only the low and medium perm model without BHG (0 mm/m) might have some value since the shape of the HDI should not be impacted in that scenario (or is it?). | The value of these models is that they show that if the artificial flow-field introduced by the bottomhole pressure is stronger than the BHG, the importance of the BHG ceases (Fig. 2a, b). Even though, these models represent unrealistic cases in terms of the bottomhole pressure. To show the same effect in a model suite with higher permeability we would need a BHG far smaller than used in our study. Another example is that the impact of layering on thermal breakthrough times, is less well observable in the high permeability models (because the bottom hole pressure is too low in these cases to investigate the effect; please compare Figure 4a with Figure 4g) and can only clearly seen in the unrealistic low permeability models. We think therefore that these (unrealistic) parameter combinations are an integral part of the study and should not be omitted. | Changes made in lines 82-84. |
| **3.4** | In addition, wouldn't the induced BHPs also impact the flow velocity in the reservoir and therefore also thermal breakthrough (I am not certain here, but at least mention and discuss)? | Flow velocities are limited by how much water is injected and produced from the system, and are therefore not a function of the BHP's. The main effect of the BHP can be seen in its interaction with the background hydraulic gradient. We agree with the reviewer and will now mention this point in the revised manuscript. | Changes made in line 159-164.. |
| **3.5** | As a consequence, I would recommend to exclude all other low and medium perm scenarios with a BHG > 0 mm/m. Otherwise please discuss accordingly and inform the reader in the "Methods" section about a) the unrealistic BHPs, b) their impact and c) why the models might still have some value. | These models are an integral part of our study. Please see our answer to **points 1.1, 3.1, and 3.3.** | Changes made in lines 78-86, 159-164, and 233-235. |
| **3.6** | Alternatively, the models could be rerun for different flow rates (e.g. with a fixed draw-down pressure, which is a much better technical parameter to be controlled and | Firstly, if we had chosen a fixed draw-down-pressure, we would have had to deal, at least in part, with extremely low or high flow rates, i.e., the amount of injected cold water would change. Consequently, we would not be able to analyse the interaction and impact of the tested petrophysical and structural parameters, which is the main focus of our manuscript. | Changes made in lines 159-164. |

| | more or less independent of the geology/petrophysics). | Secondly, for making a large series of models that can be compared to each other and in which the effects of individual parameters can be isolated, the option would be to either to use a fixed bottom hole pressure, which will induce unrealistic flow rates in some models, or fixed flow rate, which will result in unrealistic pressures. The choice for fixed flow rate is because this has the least disturbing effect on the model results because the amount of injected cold fluid stays the same. With other words, to rerun some of the models with fixed draw-down pressure would not correspond to the setup of our study, rather it would alter the results of these particular models and therefore destroy comparability. Please see also our answer to your comment **3.1 and 3.3** above, where we answer similar questions. | |

**Specific comments R1:**

| 4. | **Abstract** | | |
|---|---|---|---|
| 4.1 | Well written, please consider to avoid usage of acronyms (BHG and HDI). | Here we follow the standards of the Journal that require the introduction of acronyms in the abstract. | No changes made. |
| 5. | **Introduction:** | | |
| 5.1 | Line 33: Maybe better say hydrothermal than deep geothermal (petrothermal/HDR is also deep geothermal, but only produces from fractures). | We follow the advice of the reviewer and replace "deep geothermal" with "hydrothermal". | Changed in line 36. |
| 6. | **Methods** | | |
| 6.1 | Very minor, but almost all sentences start with "We…" | We agree and we will reformulate this part. | Changed in line 88-101. |
| | **Geometry of the model:** | | . |
| 6.2 | The horizontal extent of the model seems to be rather small (only 4 km), while the vertical extent is very high (2.3 km). It is not clear if this extent only represents the reservoir or also overburden and footwall sediments. Please specify. | We think that the best way to approach this issue is: that in our sensitivity study, the whole model domain should be seen as a potential reservoir volume, i.e. our study investigates which parameters control and or influence the volume that actually can be utilized as a reservoir. We will improve the text accordingly to avoid potential misunderstandings.
For your comment on the lateral extent, please see our answer to your comment on line 105 below (**see also point 6.5**). | Changes made in lines 126-127 and 150-153.. |

| 6.3 | Line 91: The rescaling of the well diameter and "length" is confusing. Please explain in more detail, how and why the rescaling has been done and what is meant by "length" and "active part" (perforated production zone?). | We will follow the suggestion of the reviewer and rewrite this part accordingly. Standard well diameters are a few decimetres. This in turn would need a very fine mesh. To avoid this issue we used a larger diameter for the wells. To account for the unrealistic high diameter and thus the area of the "perforated production and injection zone" we choose to adjust the area via its length to a size that is in a realistic range. | Changes made in lines 129-132. |
|---|---|---|---|
| | ***Temperature:*** | | . |
| 6.4 | Line 97: The gradient's unit is wrong (should be 0.047 degC/m not per km).
Also, please briefly explain why the respective gradient and surface temperature have been chosen. Especially, since the gradient is very high and the surface temperature is very low. | Thanks for identifying this mistake. We corrected the typo 0.047°C/m.
Since we carried out a non-site specific numerical sensitivity study, we chose a realistic gradient that allows for electricity production at this depth. The surface temperature was chosen arbitrarily to be 0°C. This is in our opinion neither particularly high nor low, especially when considering that our numerical sensitivity study is not site specific. Nethertheless, the effect on the model results can be neglected, since a slightly increased surface temperature would alter not the temperature at target depth or the model results significantly. | Changes made in lines 84-86 141, and 142-144. |
| 6.5 | Line 105: This explanation of the model size should be move to the geometry section (2.2). The explanation itself is not really convincing: the model probably could have been extended to 10x10 km without significantly more cells, since no high resolution is required at the boundaries and far away from the wells.
Please at least mention/discuss possible effects here and in the discussion section. | In Line 103-104 of our manuscript, we describe that the temperature boundary conditions do not affect the model results, i.e. the size of the model domain does not affect the model results. The sentence in line 105-106 is thus obsolete. We will delete the last sentence. This solution also makes merging the description of the model geometry unnecessary.
The only limitation by the comparatively small model domain is that we cannot examine in all cases the complete geometry of the HDI (hundred degree isotherm). | Changes made in lines 150-153. |
| | ***Fluid flow:*** | | |
| 6.60 | Please explain the setups of the various background hydraulic gradients here or later (see next comment). | Please see our answer to **point 6.7.** | Changes made in lines 156-157. |
| 6.7 | Also please explain how the variation is implemented. Figure 1b is not doing a good job explaining the variation. Is the BHG varying from the center towards a certain | The BHGs are valid for the whole model domain, i.e. the BHG is not varied in the individual models, but interacts with the artificially introduced flow field. The BHG is applied as a pressure gradient on the model boundaries. We will explore this in more detail and improve Figure 1b. Please see also our answer to **point 1.2.** | Changed Figure 1b.
Changed lines 60-66 and 150-157. |

| | | | |
|---|---|---|---|
| | direction? Or from one "edge" of the model domain to the opposite one? Is the BHG a differential gradient in the reservoir or the entire cube? Since this seems to be such an important parameter, please try to be as precise as possible. Also, please provide some geological scenarios that justify the chosen variations in hydraulic gradient. | | |
| | *Scenarios:* | | . |
| **6.8** | Line 127: At 2-3 km burial depth, a matrix permeability of 10-11 m2 (10 Darcy) seems a bit high and probably impossible, when combined with 3% or 14% porosity. Please discuss or at least think about removing the high-perm-low-poro scenarios (or give an adequate geological scenario). In general, please consider giving some real world analogs/examples for the chosen poro-perm scenarios. The sandstone reservoir literature should be full of good examples. | We improve the method section to clarify this misunderstanding and add that the permeability values are not linked, respectively provided/controlled by the matrix porosity. We used instead a continuum approach (Berkowitz et al., 1988; Lege et al., 1996; Kolditz, 1997), that uses a replacement media for the fractures and which provides mean hydraulic properties of a given fracture system. This is in our opinion a justified assumption, since permeability is in consolidated sediments often to large parts provided by fractures (Bear, 1993; De Marsily, 1986; Hestir and Long, 1990; Nelson, 1985). Please see also our answer to **point 1.1.** | Changes made in lines 67-70, 78-86, and 176-182. |
| **6.9** | Line 145-146: It would be nice to have some real-world justification for the chosen fault permeabilities. There is a lot of literature available. | We accept the suggestions of the reviewer and justify the chosen parameter values in the introduction. Please see also our answer to **point 1.1.** | Changes made in lines 52-56. |
| **6.10** | Lines 149/150: Please provide some geological scenarios that justify the chosen variations in hydraulic gradient. | We follow the reviewer's suggestion and explore the topic in more depth in the introduction. Please, see our comment above to **point 1.2.** | Changes made in line 60-66. . |
| **7.** | **Results:** | | |
| **7.1** | Line 165/166: According to figures 2e & 2f, this is only true if the BHG is applied in the direction of the injection well (fig. 2f). | The reviewer is correct. Here we provide/describe the ranges of reservoir lifetimes observed in scenario 1, for different reservoir permeabilities. These ranges depend naturally also on the other parameters varied in our multi parameter sensitivity study. This is why we choose to | No changes made |

| | | present our results in different plots, e.g. lifetime vs, permeability, and lifetime vs direction of the hydraulic gradient. | |
|---|---|---|---|
| **7.2** | Line 180: This makes sense, but how realistic is it to have a rock/sediment with a permeability of 10-11 m² and a porosity of only 5% or 14%? | Please see our answer to your comments to Line 127 (**point 6.8**). | Changes made in in lines, 67-86, and 176-182. |
| **7.3** | Line 236: Why is the stabilization at 100°C? | In our study, we investigate the effect of multiple parameters; there are certain combination that can produce similar results, in this case the convergence to 100°C in Figure 2j, 6g, 7g, 8d, 8g, 9g. To analyse this in more depth, would require a different sensitivity study with a different setup.
We will also modify the sentence to: "In the presented model runs as shown in Figure 4, temperatures stabilize at a final temperature of about 100°C." | Changes made in line 318. |
| **7.4** | Line 237: Wouldn't you expect a significant effect of convective flow in a vertical fracture? | We assume that this question is likely caused by the fact that we were not clear enough about how permeability is implemented. See also our answer to **point 7.6 and 2.1**.
We will also rephrase the sentences in line 238-239 (discussion paper) to: "..., compared to the other directions, as common in fractured reservoirs (Figs. 1e, 5)."
We did not introduce additional vertical fractures in this scenario, but increase the fracture anisotropy in the given plane. This question would be necessary to answer if we would have used a discrete fracture model. We are convinced that this question will be answered after we improved the method section regarding the implementation of permeability and porosity.
Also in natural fracture systems the vertical extent of fractures is commonly restricted, i.e. many fractures stop at sedimentary contacts/layers and thus density-driven heat flow would be hindered in the vertical direction, as the reviewers agrees in **point 2.1.** | Changes made in lines 176-182, 321, and 534-538**.** |
| **7.5** | Line 253-254: Please rephrase or put more detail. What do you mean by: "a closed geothermal loop may not be feasible"? | We agree with the reviewer and rephrase the sentence to "…..the establishment of a closed geothermal system becomes unlikely." | Changes made in line 338-339. |
| **7.6** | Line 258: Not sure what we can really learn from this part, since many real-world projects have shown the significant impact of convection on the temperature field of fault-controlled reservoirs (e.g. Soultz-sous-Forets). | We discuss this limitation now. The main point will be: Whereas our models likely underestimate the lifetime of fault-related reservoirs that allow for convection, they allow for improved estimate of how strong the effect of convection should be to counteract the negative influence of the channelling effect. Thus, it shows the importance to know the budgets of both the channeling effect and the effect of density driven convection to make assumptions about their effect on the potential lifetime of a geothermal reservoir. | Changes made in lines 534-538. |

| | | Please see also our answer to **point 2.1** and to the lines 363-368 of our manuscript, where we refer to a real world example of this observation. | |
|---|---|---|---|
| **7.7** | Line 258f: What is the permeability of the matrix (host rock)? | See line 141f (Discussion paper). We agree with the reviewer and now repeat the value of the bulk permeability of the host rock in this section to improve readability.
Please note our answer to point 6.8 in which we clarify how permeability is implemented. | Changes made in line 347-348. |
| **7.8** | Line 291: "…BHG,  the temperature stays…" | We thank the reviewer and will correct the sentence. | Changes made in line 376. |
| **8.** | **Discussion:** | | . |
| **8.1** | Line 313-328: Maybe this part would be much better placed in the introduction and in some parts in the "Scenarios"-part (see previous comments on mentioning analogs etc). | Our study is a non-site-specific sensitivity study (with simplified models). We use this part as an introduction in the discussion section to show how parameters such as porosity and permeability can be highly variable. We discuss that even comparatively small variations of these parameters have a strong effect on a reservoir's performance. Thus, we think that the structure of the manuscript, as it is, is justified. We are aware, however, that it would be possible to tell the story in a different way.
However, now we will discuss in more detail the implication of figure 10 for our study and geothermal energy in general. Further, we will better explain in the Introduction the aim of our study. This will also include a point regards the variability of geological systems. See also our answer to **point 1.1.** | Changes made in lines 67-86. |
| **8.2** | Line 335: How does the bottomhole pressure impact the influence of the BHG? In particular in the low-permeability case? Please mention earlier (e.g. in the Methods or Scenarios section(s)). | The influence of the bottomhole pressure on the BHG depends on the ratio between both. If the bottomhole pressure is higher than the BHG it dominates and vice versa. The low permeable cases are due to high bottomhole pressures unrealistic, but allow to investigate the effect of other parameters like permeability contrasts. In our opinion, these points are preferably placed in the discussion section. We, however, will state that some of the scenarios are unrealistic and that both fluid systems interact and we will specify that point in the method and discussion section.
Please see also our answer to **point 3.3**. | Changes made in lines 159-164 and 228-235. |
| **8.3** | Line 335f: Here is the answer of the last comment: "the BHG is outperformed by the artificial flow field caused by the very high bottomhole pressure". Actually, the bottomhole pressures in the medium and low permeability cases are impossible in | We agree with the reviewer please see above.
The low permeability/ high fluid pressure models underestimate the effects of the background hydraulic gradient.
The importance of the findings of the low and medium permeability models as well as the use of the constant production and injection rates are justified above in **points 3.1, 3.3, 3.6.**
We hope that answers the questions raised by the reviewer. | Changes made in lines 159-164 and 228-235. |

| | | nature. The question is then, what is the meaning of the modelling results? An elegant way to avoid this problem would be to work with a constant draw-down instead. | | |
|---|---|---|---|---|
| 8.4 | Line 361: Please consider providing some geological scenarios for variations in BHG. | We assume that this is a misunderstanding. We have not introduced variations of the BGH within one individual model, but we assigned different BHG to individual models. We improved the method section to clarify this issue. Please see our comments to **points 1.2 and 6.7.** We also, as requested, provide improved introduction regarding the BHG. | Changes made in line 58-66 and 156-157. |
| 8.5 | Line 379f: "Notably, in the low and intermediate permeable models, where permeability contrasts are higher than 1 order of magnitude, none of the tested BHG configurations could compensate for the small volume". Or is this again related to the unnaturally high BHPs in the low and medium permeability scenarios? Please discuss. | We agree with the reviewer, and add that the unrealistic high bottomhole pressures do not allow the BHG to affect the system. We also correct the typo "higher than 1 order of magnitude" to "higher than 2 orders of magnitude" | Changes made in line 467. |
| 8.6 | Line 387: instead of "borecore": core from boreholes. | We thank the reviewer and we will correct the sentence accordingly. | Changes made in lines 475. |

**General Comments RC2:**

| | Figures: | | |
|---|---|---|---|
| **1.1** | Figures in general have a small scale for (small) colored dots and a (uselessly) large scale for the vertical. Furthermore, the colors used are the same. This is misleading the reader. My suggestion is to use different color codes for the two parameters (depth and BHG) and change the relative dimensions of the two scales, since the focus of the manuscript is on the BHG (color-coded dots). | We follow the reviewer's suggestions and will adjust the size of the legend for the HDI plots. Further, we'll try to find a different colour scale for the HDI. | We modified the figures as suggested. |
| **1.2** | Figs.2a-c (as well as other corresponding plots) either have inverted y-axis scale (sic!), or I did not understand the figure and/or the text (cfr. lines 161-165). This produced some initial misunderstanding of the work (the text is not properly describing what is presented in the figure). | The y-scales of each scatter plot show the time to thermal breakthrough, i.e., the time at which the production temperature reaches 100°C. There are no depth scales in the scatter plots. There are no inverted scales in any of the figures of our manuscript.
We assume, as the reviewer pointed out in **1.1**, that this misunderstanding is caused by the fact that the colours for the scatter plots overlap in parts with the colour code used for the figures showing the HDIs.
We will try to find a different colour scale for the figures that present the shape of the HDIs.
We are aware that the presentation of the results, owed to the multiple parameters we analysed, is somewhat unconventional. We will follow the suggestion to provide a short introduction/explanation on how to read the figures in the Method section (see also point **1.5, 1.6, and 2.8**). | Changes made in lines 220-226. |
| **1.3** | In many experiments the temperature stabilizes at around 100_C (Figs. 2j, 6g, 7g, 8d, 8g, 9g). The reason for this coincidence with the HDI not clear or explained. The author should justify this "convergence" in the various models. | The question why in some of our models the temperature converges to 100°C was also raised by reviewer 1 (**RC1s point 7.3**). The answer is that in these cases it is a coincidence and the result of a complex interplay between the chosen parameters, i.e. thermal gradient, surface temperature, porosity, permeability. | No changes made. |

| | | | |
|---|---|---|---|
| 1.4 | In Fig.1a the projection of the wells provides the impression that their trajectory is oblique. The Author should either correct the figure or describe the reason for oblique wells as well as quantify it. | The wells are indeed oblique. We will include the parameters in the method section. The reason for the inclined wells is that it allows us to keep the whole well within the damage zone of the fault in Scenario 5, which is also oblique, i.e. it dips. For comparability, we used the inclined wells consequently in all the other models. | Changes made in lines 134-136. |
| 1.5 | In the Figures the Authors should include the number of experiments represented (i.e. the number of dots in the single figure). | The reviewer is correct: The number provided in the figure captions could be misinterpreted. Indeed each of the corresponding plots contains the same results, i.e. the same number of dots. The difference is that in the different plots the same results are presented with different x-axis to explore the impact of the multiple parameters. We will clarify this issue as promised above (**point 1.2**) using a short introduction in the figure setup in the method section. We will also correct the figure captions, from e.g.: "Plots (a), (b), and (c) contain the results of 225 simulations." to "Plots (a), (b), and (c) each contain the results of the same 225 simulations." We think this is less confusing than presenting in each sub figure the same number of experiments. We hope the reviewer agrees with this option. | Changed figure captions for clarification (Figs. 2, 4, 5) |
| 1.6 | The author should discuss the case of a strong variation in the results (e.g. Fig. 1a, red dots for BHG=20 mm/m, at permeability 10-11). | We are convinced that the influence of the BHG is sufficiently discussed in the lines 351 to 368 (SED), where we describe that, at high permeabilities, the BHG can outperform the artificially-introduced flow field. We are convinced that the reviewer's question becomes obsolete with the new improved method section that introduces the setup of the scatter plots, i.e. that the scatter plots presented next to each other must be seen in combination and not as stand-alone results.

In this particular example cited by the reviewer, the corresponding Figures 2a, b, and c should read as follows. The thermal breakthroughs of less than 20 years, is according Figure 2a, at permeabilities of $10^{-11}$ m². The same data points in Figure 2b (red dots with lowest thermal breakthrough times) show that this short time to thermal breakthrough is observed for hydraulics that are directed southwards and have a magnitude (colour code red, as shown in the legend) of 20 mm m$^{-1}$. In Figure 2c, the same data points are plotted according to the porosities used in the models. Here, it shows that, in this case, porosity plays a minor role in determining the expected lifetime of the reservoir. | Included lines 220-226. |

| | | We additionally visualized this connection using horizontal lines that connect results of the same experiments. | |
|---|---|---|---|
| **1.7** | Numbers are very small to pretend some statistics (mean, sd), but they could mean unreliable results and should be discussed. | We agree that a further statistical evaluation is futile.
We also think that providing parameters like mean and standard deviation might not be the way to further investigate our results, because the present results are from different experiments with varied model parameters. This means, our experiments do not allow such statistics. | No changes made. |
| **1.8** | The "line connecting the same experiment" is not clear. In Fig. 2 the yellow line connect one yellow dot per figure, and it is easily understood. On the other hand, in other figures (e.g. Fig. 4a-b, green lines) the do connect multiple dots in the same figure. This is confusing: how many numerical experiments were responsible for each dot in each figure (I assumed one)? Maybe they partially overlap. | The reviewer is correct. Each dot in each panel is the result of one numerical experiment, and each of the according panels contains the same model results. Due to the large number of experiments, we unfortunately cannot avoid that, in some cases, the line connecting the dots, which belong to same experiment, also cross other points.
In this case (Figure 4a, b), it shows that the varied parameters (direction and magnitude of the BHG) does not influence the time to thermal breakthrough in the modelled time span. Consequently, the dominating parameter that determines the reservoir's lifetime in the low permeability series is the permeability contrast. Please see also lines 370f and 204f of the discussion paper.
Please see also our answer to **point 1.6**.**and 2.8.** | Included lines 220-226. |

**Specific comments R2:**

| **2.1** | In **lines 34-39** the Authors discuss the poor improvement in porosity due to the presence of fractures. This is true, but the author are not considering the main role provided by fractures in improving the effective porosity by connecting isolated pores, as is normally achieved in tight-gas reservoirs (gas-shale). In general, the manuscript is not discussing on the difference between total porosity and effective porosity. I guess that the porosity they consider in the numerical experiment is merely the effective one, and this should be | The second part of this question address how permeability is implemented in the models. The same issue was raised by reviewer 1 (**e.g. point 6.8**), it will be answered in the method section and we will clarify that the permeability in the models is not linked to matrix porosity. Instead we use the continuum approach (Berkowitz et al., 1988; Lege et al., 1996; Kolditz, 1997), in which replacement media is used to model fractures and provides mean hydraulic properties of a given fracture system. | Changes made in lines 42-45 and 176-182. |
|---|---|---|---|

| | | | |
|---|---|---|---|
| | clearly mentioned. On the other hand, a brief note on the role of influence of fractures on effective porosity is required to complete the introduction and the discussion paragraphs. | | |
| 2.2 | Equations **in Line 72, 81 ,82** seem correct, yet references for the general audience (as Solid Earth also has) are required. | We will follow the reviewer's request and will provide additional literature. | Included in lines 113 and 115. |
| 2.3 | **Line 91-92**. The limit to 20m is not easy to be understood (e.g. where these 20m were located along the well). Maybe a better way to express this correction would be to express it as a percentage of the well hole surface (the cylinder), or by presenting the equivalent reduction in permeability between the well cell and the surrounding ones in the mesh. | We thank the reviewer. A similar question was raised by reviewer 1 (point 6.3). We will follow the suggestion of the reviewers and rewrite this part accordingly. Standard well diameters are a few decimetres. This in turn would need a very fine mesh. To avoid this issue we use a larger diameter for the wells. To account for the unrealistic high diameter and thus the area of the "perforated production and injection zone" we choose to adjust the size of the area via its length to a size that is in a realistic range. | Corrections made in lines 130-132. |
| 2.4 | **Line 96**: I guess that the geothermal gradient is in reality expressed by m and not by km... | We thank the reviewer and correct the typo. | Changes made in line 141 |
| 2.5 | **Line 105 and 113**. I guess that "computational costs" really intends the more appropriate expression "computational time". It would be of interest to the readers to quantitatively justify this sentence: add in lines 69 71 information on the used computer platform and the approximated run-time for a single numerical experiment. **Line 126** and through all the experiments. | We will now provide information on the computer platform and the differences in the runtime for the fully- and unidirectional coupled experiments. | Changes made in lines 168-169. |
| 2.6 | My opinion is that a permeability of 10exponential-11 m2 is unfair to be reached in a reservoir at the used resolution of the model, with the exception of karst cavities. The Author might include here a descriptive | Both of the here addressed issues are also made by reviewer 1. The first point regarding the high permeabilities is answered by how permeability and porosity are implemented in the models. We are now aware that we were not clear enough about this and will now address this issue in the method section to avoid potential misunderstanding. | Changes made in lines 67-86 and 176-182. |

| | | | |
|---|---|---|---|
| | correspondence to the reservoir permeability (e.g. tight reservoir for 10 exponential -15 m2, medium-high permeable reservoir 10 exponential -13 m2, karst structures 10exponential-15 m2). | Please see our answer to your **point 2.1** and to e.g. **point 6.8** by reviewer 1.
This second question, which also ties directly in with a request of reviewer 1, is to provide some real world examples. We will improve the Introduction and try to find the best compromise between both suggestions. | |
| **2.7** | **Lines 143-147** more references on measure of permeability in fault core are important here (e.g. works by R.J Knipe and/or Q.J Fisher). | We will provide more literature on fault permeability, as also requested by reviewer 1 (**point 6.9**). We, however, prefer to present them in the introduction instead of placing them in the results. We hope the reviewer finds this compromise acceptable. | Changes made in line 52-56. |
| **2.8** | Line 161-165. As mentioned, the only way I found to correlate text and Fig2a-c is to invert the Y-axis scale. Anyhow the description, even with this correction, does not correlate for 10exponential-11 permeability experiments, that scatter results all along the entire span 0->200 a apparently without any rule (e.g. red dots). Did I understand properly the figure? If not, a more careful introduction to the figure and description might be necessary. | Please see our answer to your **points 1.2 and 1.6**. We will provide an introduction on how the plots are to read.
In detail, on the example of figure 2a-c. Each of the plots a), b), and c) contains the same model results for all combinations of the three porosities, the three permeabilities, the eight directions of the hydraulic gradient and of the 4 different magnitudes of the BHGs. As the reviewer agrees, these are a lot of parameters to visualize. Whereas this is a typical approach to plot results in multi parameter studies, it is not very common in the geosciences.
We decided to keep the y-axis constant, which shows the time to thermal breakthrough. The x-axis was used to plot the same data points in different ways, to produce the patterns that show the different effects of the different values (permeability, direction of the hydraulic background gradient, porosity). In consequence, it is important to see these plots as a whole, i.e. the value they provide only becomes apparent by looking at them in combination.
The connecting line is used to show the above issue and connects the same result of one experiment and allows to identify the parameter values for each individual model run. | Included lines 220-226. |
| **2.9** | In the Figs. the meaning of the represented surface is not completely described. The Authors refer to "HDI shape". I am not sure but I guess that, considering the experiments, these surfaces represent the envelope of the volume where the temperatures become lower than the HDI due to the successful heath extraction. An explanation on the meaning of the HDI shape | We accept this point and will improve the description of what the HDI actually is.
The reviewer is correct with the description. The HDI encloses the volume with temperatures lower than 100°C. | Changes made in line 150-152. |

| | | | |
|---|---|---|---|
| | is required in the text (and maybe in the caption for the fast readers…). | | |
| 2.10 | **Line 173-174** the probability concept should be better introduced. | We accept the suggestion and will describe in more detail that elongated ellipsoidal HDIs, where their direction is controlled by the HBG, may result in a reduced probability/chance that the injected cold fluid reaches the production well. | Changes were made to lines 252-253. |
| 2.11 | **Line 178:** I guess the Authors intend Fig.2g and not 2e. | We thank the reviewer to point out this mistake. We will also rephrase the according lines for better readability: ….”The two contrasting BHGs in Fig. 2g show, either fast (e), or almost no decrease (f) in production temperature”….. | Changes made in line 259. |
| 2.12 | **Line 198** "three series". This is not clear: I see in the figure3 different permeability (these are the three series), 4 permeability contrasts and 8 different orientation for BHG with 4 possible gradients, for total of 3x4x8x4=384 combinations. Then just three BHG shapes, but for the same permeability (same series). This might be confusing. A more complete description of the model procedure might help to understand the results. | We will improve the text. In detail in line 203 (discussion paper) we will directly refer to the according figures and model series when introducing the permeabilities. We will also mark in the figures the series to which they belong.  With regard to figures 4c, f, and i, we selected exemplarily three HDI shapes from the medium permeability models. Given the 300 individual model runs presented in this scenario, we decided to show these because they constitute a good compromise, i.e. they (1) are in a realistic permeability range and (2) clearly show the effect of the permeability contrasts. | We improved Figure 4 and changes made in lines 192-194, 281-284, and 825-830. |
| 2.13 | **Fig4b** is not clear, and in general figs 2, 4, 5 are not easy figures. Same color dots appear both on high and very low times to breakthrough. This could mean the excessive scattering of results, or that results are from experiments with different, not specified, parameters. | Regarding the Figures in general, please see our answer to your **points 1.6, and 2.8**. Regarding Figure 4b in particular please see our comment on your **point 1.8**. Figure 4b shows, in combination with Figure 4a, that the only controlling factor in this low permeability case is the permeability contrast in the reservoir, i.e. the orientation and magnitude of the BHG is of no importance. This correlation however is altered (Figure 4d, e and Figure 4 g, h), if the permeability of the reservoir layers increases, i.e. the BHG can, in these cases, compensate the limitations introduced by the permeability contrast. Please see lines 376f and results **3.2 Models of layered reservoirs in the discussion paper.** We assure the reviewer that all parameters are specified and can be picked in the figures as described in e.g. **point 2.8**. | We improved the figures and included lines 220-226. |
| 2.14 | I think to have properly understood the relations between the dots in Fig. 4a, b and | We assure the reviewer that we have discussed and tested many options to present the data, including the ones made by the reviewer, i.e., using different symbols and/or adding references. | We improved the figures and |

| | | | |
|---|---|---|---|
| | the reason for the limited connection presented in Figs. At the present stage, the figure is very difficult to be understood (also due to the high number of combinations in the experiments – i.e- the number of parameters used - and the limiting 2D of the journal pages…). The Author could try to improve the correlations by either using different symbols for each experiment (good luck, it would be a big effort with questionable results) or by adding a reference number to each dot. The diagrams have a relative small number of dots and al lot of empty space. A simpler alternative might be to add in the text the clear description of a correlation among dots as an example. There are also some evident overlap of dots (just comparing among figures) and this should be described (or slightly move one of the dots within the resolution of the results). | It, however, did not improve the figures, as expected by the reviewer. We found that the way we finally chose, as is common in multi-parameter studies, is probably the best. However, we will provide an additional section in the methods that helps to understand the concept of the figures. See also our answer to your **points 1.6, 1.8, 2.8, 2.13** above. | included lines 220-226. |
| 2.15 | **Line 215-220** again: the cited 70 years seems to correspond to 130 years in Fig.4d, second column. Is there again reversed the Y-axis scale? | Here we are writing about the range of observed lifetimes. The range of observed lifetimes, in this case, is indeed about 70 years, i.e. between 130 and 200 years. | No changes made. |
| 2.16 | **Line 235**. As previously mentioned. Why at 100_C? This should be justified by the Authors. | Please see our answer to your **point 1.3.** | No changes made. |
| 2.17 | **Fig.5** the origin of dots on top of the plots a-c (i.e. at >200a) is not clear. | In this, as in many other model runs, the production temperature did not fall below the 100°C threshold. In consequence, we decided to assign the results of expected lifetimes to be at least 200 years, i.e. the time modelled. | Changes made in line 217-218. |

| | | On the example of Figure 5. This the only model configuration that allowed a hydraulic connection between the wells because the fracture anisotropy is parallel to the well alignment. In the other cases, the fracture anisotropy, even the lowest, hinders the reinjected cold fluid reaching the production well, and consequently, under the applied model setup, the temperature stays above the threshold. | |
|---|---|---|---|
| 2.18 | **Line 264-265** Fig. 5g shows that temperatures stabilize at 100_C. How this happens at exactly the critical temperature chosen for the HDI? Is this input in the model? Some explanation is needed. | Please see our answer to your **point 1.3**. | No changes made. |
| 2.19 | **Lines 340-349** Here is perhaps the proper space to discuss the total porosity and the effective one I discussed above. As I understand, the chosen porosity is intended to be 100% effective. A sentence explaining this should be anyhow added to the article. | Please see also our answer to **point 6.8** from reviewer 1 and your **point 2.1**. | No changes made, see method in line 179-180. |
| 2.20 | **Line 371.** This assumption may be too forced, and I am sorry for the referenced articles. Secondary fractures and faulting allow permeability to take over thinner clay layers that lose their sealing property. This is more difficult in thicker clay layers. I understand that in the useful proposed model are necessary simplifications, but it is not the case for the complexity of real geothermal reservoirs. | We acknowledge that the reviewer agrees that this is an unfortunate but necessary restriction of the models. This is however, the concept of our study, i.e. to use simplified models and to show that, even with these simplifications, predictability of the modelled systems is extremely complex. This agrees also with the reviewer's **point 2.27** that our manuscript presents a first step in pointing out these difficulties.
We agree, of course, with the reviewer that fractures also propagate through "sealing" layers. Even though such softer layers hinder fracture propagation. For this reason we wrote,"restrict fluid flow across them" to avoid a too strong statement.
We will additionally rephrase the sentence in line 371 (discussion paper) to:
"(Sub)horizontal permeability contrasts can be caused by layering in sedimentary rocks and can span several orders of magnitude (Zhang, 2013), even though these sealing properties are altered or reduced by barren fractures."
We hope this is an acceptable compromise. | Changes made in lines 458 and 461. |
| 2.21 | **Line 399.** I do not see evidence in Fig. 5b to justify this sentence. At my sight, the resulting timings are fully independent from | The reviewer is correct. Indeed, we refer here to Figure 5a, b, and c. With this correction, the statement in line 399 is justified.
The sentence in Line 399-400 will be rephrased to: | Changes made in lines 488-489. |

| | | the BHG values (colored dots). May be the Authors are referring here on the BHG orientation of Fig.5c. | "Second, fracture anisotropy in the range of 1 order of magnitude, with respect to the bulk permeability, leads to either very short- or long-lived geothermal reservoirs, depending on the BHG properties and the orientation of fracture anisotropy (Fig. 5a, b, c)." | |
|---|---|---|---|---|
| 2.22 | **Line 423-414** Fractures and secondary faulting associated to faults have generally various angles to the faults and only a minority lies parallel to it (cfr. Riedel). This results in: fracture intersections, fracture opening by the stress induced from the kinematics along the fault (friction). These factors guarantee the higher permeability of fault damage zone to a certain extent, as described in the literature. To be explicit: "often-observed" of "fault-parallel fracture anisotropy" does not correspond to either field outcrops and cores across fault zones, apart from S-C structures, where in any case C planes are generally subordered in number to S ones, My suggestion is simply to eliminate the "often-observed" attribute. | We accept the reviewer's comment that our statement is eventually too strong. However, following the suggestion to delete "often-observed" makes the statement much stronger. We will modify the sentence to: "This typical characteristic of fault zones thus increases the chance of good hydraulic connection between injection- and production wells and is potentially further improved by fracture anisotropy in the damage zone, which is often (sub)parallel to the fault." | Corrections made in lines 504-505 |
| 2.23 | **Line 429** The previous concept is repeated here: useless redundancy and same comment. | We do not fully agree with the reviewer. In the lines 411 to 425, we discuss why faults have become recently prime targets in geothermics and the difficulties that have been reported. In lines 425 to 440, we discuss the results of our models and how they agree or disagree with common knowledge. We will slightly modify the sentence in line 425-426 to: "Our simplified models support these findings and show that faults, with damage zones that constitute positive permeability contrasts of just 2 orders of magnitude, exhibit these channelling effects (Fig. 6)." to make the structure of this section clearer. | Changes made in line 516. |
| 2.24 | **Line434-435** the use of the terms "opposed/opposite" to indicate opposite (!) | We accept the reviewer's suggestion and will rephrase the sentence accordingly to: | Changes made in lines 526-528. |

| | dipping is misleading. A rephrase would solve it. | "We observed that, when the BHG is oriented against the dip direction of a fault, the fault can be considered a more sustainable target for geothermal exploitation than a fault with a BHG oriented in dip direction (Figs. 7e, f, 8e, f)." | |
|---|---|---|---|
| 2.25 | **Lines 62, 442**: they were 1027 (from line 150). This is an interesting and serious number of runs and it would be effective to remark this number both in the introduction (say, "over one thousand numerical experiments") as well in the Conclusions "(1027)". My impression is that "large series" or "a series of" would be –alas – interpreted as much smaller number in present-day publish-or-perish scientific environment. | We very much appreciate that the reviewer values our work and we will stronger pronounce the number of experiments carried out by us. | Changes made in lines 14-15 94-95 540. |
| 2.26 | **Line 457**: This is not so simple. This sentence does not take in consideration the improvement of the effective porosity that is induced by fracturing that in turn may be enhanced by the oriented stress that develops in presence of strong BHG. Since the point about effective porosity changes is not taken into consideration in the presented models, my suggestion is to specify this in the sentence (referring to "in many cases" might be not sufficient). | The reviewer is correct that effective porosity is, in many cases, improved by fracturing. However, the reviewer accepts also that an investigation of this effect is not part of our experiments. We, here, refer to our model results that show that the positive effect of porosity has on heat capacity and thus on the reservoir lifetime is minor, compared to that of permeability and BHG.

We will rephrase the sentence accordingly to ", in many cases, the positive effect of porosity has on heat capacity and thus on the reservoir lifetime, is minor, compared to that of permeability and BHG....".

We hope that the reviewer can accept this solution.
Please see also our answer to **point 2.1,** in which we explain how porosity and permeability are implemented in the models. | Changes made in lines 176-182, and 555-557. |
| 2.27 | **Line 459-462**. On the contrary, results from this work well represent the first step to model real, complex geothermal reservoirs with their Stochastic modelling by adding in the mesh the proper random values! And I am sure that the "computational costs" at | We thank the reviewer for the positive evaluation and welcome the suggestion to extend the conclusions and end the manuscript as proposed with a positive outlook, i.e., how our findings help to improve geothermal exploration in the future.
We will include at the end of our manuscript:
"Our results show that realistic site-specific models are difficult to achieve, because parameters, such as permeability structure and BHG, are often poorly constrained but can have | Changes made in line 564-566. |

| | that stage will be an insignificant obstacle. This might be a further point and a better conclusion to your article (follow the Hollywood-movie style: end always your articles with a true, positive sentence on your results…). | unforeseeable large effects on the lifetime of geothermal systems. Thus our findings provide an important step forward to judge which parameters must be known to which degree to make site specific models as reliable and accurate as possible in the future, by implementing the controlling parameters in advanced stochastic models." | |
|---|---|---|---|

**General Comments RC3:**

| 1.1 | Model parameters, including selection of porosity-permeability combinations, length of model duration, selection of 100 C isotherm, are not sufficiently justified, and may not be relevant to operating geothermal fields. | We consider feasible ranges of all parameters, including values below and above typical benchmark parameters. This is prerequisite for a sensitivity study. From the combination of all the results, we determine particular parameters and their values that exert control on the geothermal reservoir. The necessity of this approach was provided and explained by us in our response to R1 point 3.3. | Changes made in lines 67-72, 78-86. |
|---|---|---|---|
| | | **Porosity and Permeability**
**Points 2.16, 2.20, 2.25, 2.32:**
In accordance to R1 and R2, we have now improved the introduction and introduce the values for permeability, porosity earlier in the text. We discuss how these compare to typical values in different geothermal settings. We now explain better the modelling approach that we use. | |
| | | **100°C and threshold**
**See also points 2.2, 2.10:**
A universally applicable (economic) threshold cannot exist, because of the different site-specific demands of geothermal power plants e.g., district heating, electricity generation, output, depth of the reservoir. The 100°C isotherm or the 100°C threshold must be arbitrary, at least to some degree, even not taking into account that it must somehow balance with the model duration. We chose 100°C because it is sometimes referred to as minimum temperature that allows electricity production with binary cycles (e.g., Bhatia, 2014; Buness et al., 2010; Erec, 2004; Huenges, 2010; Mergner et al., 2012). We have rephrased the sentence and improved references. | |
| | | **Model duration**
**points 2.9 and 2.18, 2.34:**
There are two points to be made here. Firstly, we do not investigate the lifespan of hydrothermal power plants, but rather the role of individual reservoir parameters on the thermal development of geothermal reservoirs (see line 53-54 SED). Secondly, there is a balance between threshold temperature and duration of the model. If, as requested by the reviewer, we had chosen 150°C, then the effect of the parameters on the thermal lifetime, would be less clearly shown. If we, in addition, had only run the models for 40-50 years, then the majority of the model runs would not reach the | |

| | | important/higher threshold, i.e. there would be no data to show. For instance in scenario 1 (Fig. 2a, b, c) and 2 (Fig.4a, b, d, e, g, h) we could not identify the impact of the different parameters.

We strongly disagree that this study is not relevant to operating fields. The model results give an indication of the importance of different hydrogeological parameters on the lifetime of geothermal reservoirs, and even if the modelled lifetime of the reservoir exceeds the lifetime of a geothermal power plant, the relative importance of different parameters remains the same. | |
|---|---|---|---|
| **1.2** | Use of references and citations is inconsistent. In some cases, statements with long lists of references are too vague to be useful (i.e. not clearly tied to particular geothermal fields or a specific type of inquiry (numerical, field, experimental: : :)) and in other cases the listed references do not seem appropriate for citing in their current context. | We feel that there is some room for improvement. However, we disagree that the references are too vague.
See our comments to your detailed criticism below. | See points below. |
| **1.3** | The structure of the paper fails to emphasize the role of BHG nor does it discuss enough real world scenarios were the impact of BHG, or even suspected impact of BHG, can be shown. As it stands, almost all of the conclusions are about BHG, but BHG only gets 3 lines in the introduction. | We strongly disagree. The results, the discussion section, and the conclusions contain a sufficient information about the BHG; in our opinion, balanced together with the other parameters. The only part that can be improved with regards to the BHG is the introduction. This was our answer to the reviews by R1 and R2.
That we cannot discuss the effect of BHG for a large amount of real world scenarios is because of, to our best knowledge, the lack of data and case studies in the literature. BHG has not been considered in equivalent studies in literature before our paper.
We appreciate that the reviewer likes our findings regarding the BHG. We will improve the introduction concerning the BHG. We, however, disagree that almost all of the conclusions are about the BHG, all of the conclusion points emphasize the important role of permeability and permeability heterogeneity as well. The BHG - even though you agree that this is an important parameter- is an underestimated parameter. It is, however, still just one of the parameters that we investigated in our manuscript, and its ranking in the modelling, as a whole, needs to be understood. | Changes made in lines 58-66. |

**Specific comments R3:**

| | | | |
|---|---|---|---|
| 2.1 | **Line 11**: This sentence neglects economic factors. Rather than "can be exploited" maybe describe geologic factors influencing economic viability, as you do in the introduction. | Our manuscript considers geological reasons for geothermal lifetime. Our manuscript does not focus on economic factors and we have therefore removed any reference to this subject in the introduction. | Changes made in line 30, 93, and 290. |
| 2.2 | **Line 17**: 100°C isotherm is not well justified. See additional comments below. | This is the abstract of our manuscript, we here report solely the threshold we use and are convinced that any justification of the 100°C isotherm at this place would be misplaced.
See also point 1.1. | No changes in the abstract.
The 100°C isotherm is justified in lines 90-94 |
| 2.3 | **Line 29**: The first few lines of this paragraph make it seem like these references pertain to hydrothermal settings specifically. In this current configuration, Laubach et al. (2009) does not seem like an appropriate reference as they do not describe fracture patterns in hydrothermal systems, nor do they explicitly describe the impact of fractures on permeability or volume (other than tangentially) but rather compare fracture and mechanical stratigraphy. | This line introduces the difficulties to predict reservoir properties. This is the case of permeability. Permeability is commonly provided by fractures. In Laubach et al. (2009), they point out that fracture patterns are difficult to predict (and therefore also permeability). Thus, we are convinced that citing Laubach et. al., 2009 here is justified.

For instance, Laubach et al. (2009) wrote:
"*In subsurface studies, current mechanical stratigraphy is generally measurable, but because of inherent limitations of sampling, fracture stratigraphy is commonly incompletely known. To accurately predict fractures in diagenetically and structurally complex settings, we need to use evidence of loading and mechanical property history as well as current mechanical states.*" | No changes made. |
| 2.4 | **Line 32**: Manning and Ingebrigtsen (1999) concerns theoretical permeability at the crustal scale and in metamorphic rocks in particular. The link between this reference and the statement are again tenuous unless more clearly explained. | Here we write that permeability and porosity in general are rock properties that are highly heterogeneous, independent of rock type.

Manning and Ingebritsen (1999) wrote:
"*Near the Earth's surface, permeability exhibits extreme spatial variability (heterogeneity) and anisotropy, both among geologic units and within particular units*".
Thus, in our opinion, the reference is justified.

However, we will rephrase the sentence accordingly: | No changes made. |

| | | They are, independent of rock type, often highly heterogeneous because of layering, localized fracturing, and diagenesis (e.g., Aragón-Aguilar et al., 2017; De Marsily, 1986; Lee and Farmer, 1993; Manning and Ingebritsen, 1999; Zhang, 2013). | |
|---|---|---|---|
| **2.5** | **Line 37-39:** The logic here is odd. You describe high porosity in sedimentary geothermal systems, then say fracture porosity in sedimentary rocks is low (are dam sites really the best analog, i.e. Snow, 1968?), but that fractures dominate geothermal systems.
Separately these statements may all be true, but fractures commonly dominate in geothermal systems because geothermal systems are commonly not hosted in sedimentary rocks.
Also, you may want to specify "clastic" sedimentary reservoirs, as fractures can be very significant contributors in carbonate rocks. | 2.5.1: We do not understand the logic of the reviewer's comment. We wrote that fractures have a dominant control on rock permeability in geothermal reservoirs, even though their contribution to bulk porosity is negligible compared to matrix porosity. We are convinced that we have communicated this correctly in our discussion paper.

*Snow (1968) is highly appropriate in this case, because*
*Snow (1968) analysed fracture porosity in different rock types,*
*The fact Snow (1968) used outcrops at dam sites is highly relevant here, because the known intrinsic permeability at the dam sites were an asset to calculate the fracture porosity.*

2.5.2: We strongly disagree that geothermal systems are commonly not hosted in sedimentary rocks. There is a large number of examples worldwide for geothermal systems in sedimentary rocks (see Moeck, 2014). We also strongly disagree with the reviewer's point that fractures only play a minor role in geothermal systems hosted in sediments. For instance in the Upper Rhine Graben, the permeability is controlled by fractures in lithified sedimentary rocks (Meixner et al., 2014; Egert et al., 2018).

2.5.3: We disagree. This study could be used for both clastic and carbonate reservoirs. The range of parameters used in this study covers both cases. Fractures dominate most deep geothermal systems. | No changes made. |
| 2.6 | Line 42-43: The statement about specific failures needs referencing. | We are aware of this issue. However, this is tricky, since such negative examples are commonly not published in scientific literature. In our experience links to webpages on failed projects disappeared over time. Nevertheless, we will provide links to websites, if possible. | Changes made in lines 42-44 |
| **2.7** | **Line 45:** Beall (1994) does not appear to be about declines in production fluids nor fault damage zones, but rather to be about tracer tests and what can be learned about fluid saturations. | We have deleted the reference to Beall (1994). | Changes made in line 47-48. |

| 2.8 | **Line 48-50:** BHG is a huge part of your overall paper but has a tiny role in the introduction. This should be much larger, with specific examples of where it has impacted production. It could be your primary hypothesis and seems like the major contribution, but it is not firmly established in the introduction. As it stands, the introduction does not lay the necessary foundation for the paper, not establish a clear hypothesis, but it could be reworded to emphasis BHG (see comments about 363-368). | Please see our answer to your general comment (point 1.3) above.

In accordance to R1 and R2, we have improved the introduction regarding the BHG. However, we are convinced that, even though the BGH is important, that the introduction as well as the other parts of the manuscript should remain balanced regarding the investigated parameters.

We strongly disagree that the aim of our manuscript was not sufficiently communicated.
In lines 51-56 (discussion paper), we did provide the objective of our manuscript.

We, as requested by R1 and R2, have modified this part and included more details. | Changes made in line 58-66. |
| :-- | :-- | :-- | :-- |
| 2.9 | **Line 58:** The lifespan of 200 years in not well justified. This is longer than the nominal lifespan of geothermal powerplants (which may be closer to 30-50 years). Furthermore, most of your graphs show major deviations between scenarios early in the life of the model. I'd change the approach and the figures (graphs) to emphasize time frames that are more relevant to plant economics. | See our answer to point 1.1. | No changes made. See lines 67-72 for the concept of our study. |
| 2.10 | **Line 61**: Regarding 100°C as a threshold. On cursory examination, I did not find reference to this number (which seems very low and rarely economic unless the system is particularly shallow, productive, or in a great market) in the DiPippo volume. Instead, look into Bertani (2005) for some examples of typical producing (and presumably economic) values. Furthermore, I would expect major economic and efficiency loss well before your | We do not consider efficiency loss. We also do not carry out an economic feasibility study; see our objective. In our introduction (discussion paper), we communicate that we carried out a sensitivity study in which we investigate the influence of petrophysical and other parameters on the thermal development of geothermal reservoirs. Thus, the points addressed by the reviewer are not the focus of our manuscript.
See our answer to point 1.1. | Changes made in lines 90-94. |

| | | | |
|---|---|---|---|
| | production temperature declined from 150 to 100°C.
Bertani, R. (2005). World geothermal power generation in the period 2001–2005. Geothermics, 34(6), 651-690. 10.1016/j.geothermics.2005.09.00_ | | |
| 2.11 | **Line 66-68:** Consider emphasizing BHG instead of all the others. | We have, according to the comments by R1 and R2, rewritten the last section of the introduction, with focus on the objective of our study. In our opinion, the presentation and discussion of the results is well balanced concerning the different investigated parameters. | No changes made. |
| 2.12 | **Line 94-95:** The issue with well spacing seems to distract from BHG, until you specifically related the impact of BHG on effective well spacing. The introduction of parameters overall could take more care. | With our manuscript, we do not concentrate solely on the BHG. We present a sensitivity study in which we examine different parameters for their importance. One is well spacing. No changes needed.
Again, the reviewer draws all the attention to the BHG. In addition, the effect of the BHG on well distance is made. We described it in short but appropriately and with the possible details in lines 187 – 197 (discussion paper). | No changes made.

Impact of BHG is described in lines 194-197 (SED).

Parameters are introduced in line 126-129 and 185-187 (SED). |
| 2.13 | Line 97: Change to 0.047C/m-1 | We thank the reviewer and corrected the typo. | Changes made in line 141. |
| 2.14 | **Line 97**: Is a linear gradient throughout justified? In higher permeability systems you may expect isothermal reservoirs. | Numerous studies have shown that a linear geothermal gradient is a good first order approximation for temperatures that are determined by heat conduction only. The initial temperatures in the model represent temperatures that are undisturbed by fluid flow, and therefore can be represented by a linear geothermal gradient. In some high permeability systems, thermal convection or topography-driven flow could affect background temperatures to an unknown degree. However, the focus of the paper was to explore the effect of induced fluid flow between the injection and production well on subsurface temperatures. Using a different initial geothermal gradient for different parameter sets would make it difficult to compare the different model runs. | Changes made in lines 82-86 and 143-144. |
| 2.15 | **Line 117-119:** You have a high geothermal gradient given limited vertical advection. | Our model scenarios describe both situations, i.e., we have model runs for geothermal reservoirs with fracture anisotropy, faults and for layered sedimentary aquifers. The | No changes made. |

| | | | |
|---|---|---|---|
| | Perhaps this study really is best described as analogous to hot sedimentary aquifers, rather than more conventional fault-fracture hydrothermal systems? I don't recall seeing this distinction. | geothermal gradient that we use is relatively high, but not unusual. The reason for not varying the geothermal gradient for the different model scenarios is discussed in the reply to the previous point. | |
| 2.16 | **Line 127-129:** Is the combination of porosity of 14% and a permeability of 10-15m2 realistic? | To carry out a sensitivity study, we also need to combine different parameter values, even if they are sometimes unrealistic. This is inevitable in a one at a time sensitivity study. The base case value of a permeability of 10-13 m2 and a porosity of 14% is certainly realistic. We did not consider co-varying porosity and permeability in our sensitivity study.
We understood also from the comments from R1 and R2 that we had to improve our methods section regarding this matter. We have now modified it and describe how permeability and porosity is implemented. | Changes made in lines 78-86 and 176-182. |
| 2.17 | **Line 140:** A 7 m wide fault core is quite large. Can you include references to justify this model parameter? | 7m is wide, but not unusual; see for instance Childs et al. (2009). Furthermore, we chose to model the fault core with this thickness to avoid the high computational cost of very fine meshes. At any rate, the thickness of the fault core is somewhat irrelevant, because the fault core was modeled as an impermeable unit. | No changes made |
| 2.18 | **Line 154:** Again, the model time of 200 years, while perhaps arbitrary, is not particularly relevant to producing geothermal fields. | Please see our answer to point 1.1. | No changes made. For the concept of our study see lines 67-71. |
| 2.19 | **Line 193-197:** This is an interesting finding, but it is lost in the paper because the structure is not set up as a test of the influence of BHG compared to other parameters (see lines 66-68). Couching this section in terms of BHG would bring more coherence to the results and discussion. | We thank the reviewer.
We investigated many more parameters and we do not agree that this point is lost. Instead, we feel that our manuscript is well balanced when discussing the contributions of BHG but also the other parameters that were included in the sensitivity study.
see also point 2.11. | No changes made. |
| 2.20 | **Line 202**: 10-15 m2 seems very low for a sandstone with 14% porosity. Better geologic constraints on parameter space would make the results more defensible (see notes Line 720). | Please see our answers to points 1.1 and 2.16. | Changes made in line 78-86, and 176-182. |

| | | | |
|---|---|---|---|
| 2.21 | **Line 317:** There seems to be a disconnect between statement and reference here. I don't think Alava et al. (2009) discuss porosity or permeability, and if it is a different parameter they describe it should perhaps be clearly specified separately instead of grouped with other references. | We have rephrased this part accordingly.
Instead of:
The variability of these and other petrophysical parameters increases with scale (Alava et al., 2009; Freudenthal, 1968; Krumbholz et al., 2014a).
We now write:
The variability of these (Freudenthal, 1968; Krumbholz et al., 2014a) and other (petro)physical parameters (Alava et al., 2009; Lobo-Guerrero and Vallejo, 2006) increases with scale. | Changes made in lines 402-404. |
| 2.22 | **Line 337**: Although bottom hole pressures exceeding lithostatic may not be unreasonable, it is not clear that your model responds to these conditions by fracturing, nor would this condition be favorable (or even permissible) in a permitted injection well. Constraining your model space to geologically reasonable conditions would make the results more useful. | The model does not include any fracturing, i.e., lithostatic pore pressures only affect fluid flow, and not permeability or porosity.
Constraining the parameter space to sub-lithostatic pore pressures would result in a loss of information, because either parameters would have to be varied together (i.e., adjusting injection rate along with permeability) which would make it much more difficult to compare models and to isolate the effect of a single parameter | Changes made in lines 178-179.

See additionally lines 159-164 and 176-182. |
| 2.23 | **Line 342:** Aren't pores and fractures always filled with fluid? | We agree with the reviewer and will delete "commonly" in Line 342. | Changes made in line 429 |
| 2.24 | **Line 342**. "Since pore space often exceeds: : :" is not needed in this argument, as you say "high porosity" later in the sentence. The "since" statement is distracting, as there are many counter examples. | We will rephrase the sentence. | Changes made in lines 429-430. |
| 2.25 | **Line 348**. Again, regarding parameter space, if 10-13 m2 is the threshold, why bother with the very low permeability cases? | See our answer to point 1.1. | Changes made in lines 82-86, 176-182, and 228-235. |
| 2.26 | **Line 363-368**: This passage makes the point that your models considering BHG are important, but it needs to be expanded, and more rigorously explored and cited (there should be many examples of fields that target | Regarding the many examples: we are not aware of many published examples of geothermal fields that discuss or report BHG. See also our answer to point 1.3. However, we will improve the introduction regarding the BHG. The BHG, as our study shows, cannot be analysed or ranked as a standalone parameter, it must be seen in combination with other parameters. | Changes made in lines 58-66 |

| | | | |
|---|---|---|---|
| | outflow zones for reinjection and upflow zones for production). I'd also consider moving a version of this into the introduction when you describe the importance of BHG. | | |
| 2.27 | **Line 388**: Check "metre" for journal style. | We used British style English throughout the manuscript, as allowed by the Journal. | No changes made |
| 2.28 | **Line 406**: "scales" to "scale" | Done | Changes made in line 496 |
| 2.29 | **Line 411**: I would either cite or change this first statement. | The statement is, in our opinion, sufficiently referenced after the following sentence. See Line 412-413 in discussion paper. | No changes were made. |
| 2.30 | **Line 411-424**: Another and significant reason there is an interest in fault zones is that fault zones are fundamental parts in many producing geothermal fields because they provide the necessary vertical permeability and advection of heat and fluid so that high temperatures are shallow enough to be economically exploited. I think your passage misses this by focusing on the complexities of faults instead of the constraint that many fields and models will by necessity involve faults. | We thank the reviewer for the suggestion and will add a statement about faults as thermal anomalies.
However, we consider the effects of the fault on the reservoir itself and do not consider the possible thermal anomaly that allows for a shallower exploitation (see line 117-118 SED). | Changes made in lines 52-56, 502, and 534-536 |
| 2.31 | **Line 439-441**. This passage is probably not necessary. | We disagree, we think it is important to discuss or least mention the restriction of our study. | No changes made |
| 2.32 | **Line 445.** Although the ranges may be real, the combination of ranges seem less plausible. | The combination of ranges may seem less plausible, but this combination was necessary to see the effects of individual parameters, and is a standard approach in one-at-a-time sensitivity analysis. See point 1.1. | Changes made in lines 67-71, 78-86, and 176-182. |
| 2.33 | **Line 472:** There is an extra space resulting in a broken link. | Corrected | Changes made in line 575. |
| 2.34 | **Line 648** (Figure 2 g). Please consider a shorter time span and temperature range. | See our answer to point 1.1. | No changes made. |

| | | | |
|---|---|---|---|
| | The timespan of 200 years and wide range in T (40-180°C) masks the more relevant changes early in the lifespan of a well or geothermal field. Furthermore, smaller drops in temperature would nonetheless have major impacts on plant efficiency. This comment applies even more to your fault-controlled models that show major changes in the first few years. | | |
| 2.35 | **Line 720** (Figure 10). It would be nice to see these plotted together as x-y, so you could support your use of 14% porosity and low permeability. Because this is described as a more generic model, might it also make sense to show values from other geothermal fields producing in sedimentary basins? | This is not possible, because the data are derived from several publications. Same region, but different places. In addition, most of the data are not linked (with the exception of Bauer et. al. (2018)). The purpose of this figure is to show just how variable rock properties are. See point 1.1. | No changes made. |

**Additional author changes**

| 1 | Line 1     | Included porosity to the title.                                  |
|---|------------|-----------------------------------------------------------------|
| 2 | Lines 100  | Corrected a typo                                                |
| 3 | Line 189   | Corrected a typo                                                |
| 4 | Line 815   | Modified Figure 3. Icons for E and W oriented BHGs were missing. |

Some further small changes by the authors, that are not requested by the reviewers, comprise only improvements of wording and are marked but not further annotated in the revised manuscript.

[revised manuscript text omitted]

Kommentiert [A26]: R1 – point 6.2

Kommentiert [A27]: R1 – point 6.3
R2 – point 2.3

Kommentiert [A28]: R2 – point 1.4

Kommentiert [A29]: R3 – point 2.12

Kommentiert [A30]: R1 – point 6.4;
R2 – 2.4
R3 – 2.13

Kommentiert [A31]: R1 – points 2.1, 6.4
R3 – point 2.14

Kommentiert [A32]: R1 – points 6.2 / 6.5

R2 – point 2.9

of the reservoir volume with temperatures lower than 100°C, can be examined in all cases.

**2.4 Fluid flow, permeability, and porosity**

The upper and lower model boundaries were closed to fluid flow. A BHG was simulated in the model, which was varied in magnitude and direction in different model runs . The BHGas applied as a pressure gradient on the model boundaries from different directions and are thus valid for the whole model domain (Fig. 1b).
We applied a specified flow rate of 75 l s⁻¹ that was distributed over a cylindrical body that represents the active part of the injection- and production well. The BHG and the artificial flow field introduced by injection and production wells can interact. We decided to use a fixed flow rate in our models, because it warrants, in contrast to the use of a fixed draw-down pressure, comparability of the models, because the amount of injected cold fluid is constant and thus achieves flow velocities that are not a function of the bottomhole pressure. Second, a fixed flow rate allows to identify the effect of the tested petrophysical and structural parameters by providing the necessary fluid flux, i.e. it avoids extremely low flow rates. A further effect is that the relation between bottomhole pressure and BHG is only controlled by permeability.
The temperature of the reinjected fluid was set to 40°C. The density and viscosity of the fluid  were assumed constant (Table 1), which means that fluid flow directly affects temperature, but changes in model temperature  do not change fluid density and cause density-driven fluid flow. This simplification avoids thermal convection and reduces computational time significantly from about 500 min to about 6 min for each model (PC platform configuration: Intel Xeon E31225 with clock rate:3.1 GHz and 8 GB RAM). In addition, thermal convection is unlikely to occur in sedimentary settings, because it requires thick homogeneous and highly permeable formations, whereas the establishment of convection cells is efficiently hindered by thin low-permeability layers that are a common feature in most sedimentary rocks (Bjørlykke et al., 1988; Moeck, 2014). The exception may be thermal convection in large, steep, continuous fault zones (Simms and Garven, 2004), which we  do not investigate here. Moreover, thermal convection generates fluid flux that  is commonly lower than  topography-driven flow (Garven, 1995), and are also lower than the flow regimes induced by the injection and production wells in the model domain.
Permeability was implemented using the continuum approach, which is, for sufficiently large volumes, a reasonable approximation (e.g. Berkowitz et al., 1988). In the continuum approach, hydraulic properties are assigned to a replacement media which has the mean hydraulic properties of a given fracture system. In our study, the parameters porosity and permeability are not coupled, i.e., because we vary each parameter separately. Therefore, we do not consider the role of effective porosity. Lithostatic pore pressures affect only fluid flow, but not permeability or porosity. Porosity controls the heat capacity of a given volume. Since fracture porosity is typically not higher than 0.001 % (e.g., Snow, 1968; van Golf-Racht, 1982) its contribution to heat capacity it can be considered neglectable.

**Kommentiert [A33]:** R1 – points 6.6 / 6.7, 8.4

**Kommentiert [A34]:** R1 – points 3.1, 3.2, 3.4, 3.5, 3.6, 8.2, 8.3

**Kommentiert [A35]:** R2 – point 2.5

**Kommentiert [A36]:** R1 – points 6.8 / 7.2 / 7.4
R2 – point 2.1, 2.6, 2.26
R3 – points 1.1, 2.16, 2.20, 2.25, 2.32

**Kommentiert [A37]:** R2 – point 2.1
R3 – point 2.22, 2.25

**2.5 Scenarios**

In the following, we define the basic model properties. Homogenous models do not include any internal structure; isotropic
models do not contain fracture anisotropy. Four basic scenarios were investigated (Fig. 1c–f). Material properties used for all
models are listed in Table 1.

In the first scenario (Fig. 1c), the reservoir is homogenous and isotropic. We evaluate the time to thermal breakthrough for all
combinations of three porosity values ($\theta$ = 3, 14, and 25%) and three different permeabilities ($\kappa$ = $10^{-15}$, $10^{-13}$, and $10^{-11}$ m$^2$).
For the combination of 14% porosity and permeability of $10^{-13}$ m$^2$, we tested the effect of the distance between injection- and
production wells.

In the second scenario (Fig. 1d), we introduced five horizontal confining layers, each a 100 m thick, at intervals of 300 m,
into the model volume. The production- and injection wells were placed in a 300 m-thick reservoir. This scenario comprises
three series with different reservoir permeabilities ($\kappa$ = $10^{-15}$, $10^{-13}$, and $10^{-11}$ m$^2$). For each of these
series, we set the permeability of the horizontal confining layers to be 1 to 4 orders of magnitude lower than that of the
reservoir. All units were assigned porosities of 14%.

In the third scenario (Fig. 1e), the model had a porosity of 14%, and a permeability of $10^{-13}$ m$^2$. We introduced vertical fracture
anisotropy that strikes N–S, NE–SW, E–W, and SE–NW and has 1, 2, and 3 orders of magnitude higher fracture permeability
compared to the other directions, in an otherwise homogenous media.

To all possible variations of different parameters in these three scenarios, we applied BHGs of 0, 1, 5, and 20 mm m$^{-1}$ and
varied the BHG direction from 0° to 315° in 45° steps (Fig. 1b).

In the fourth scenario (Fig. 1f), we tested the effect of a N–S striking, 60° westward-dipping fault, which consists of up to three
parts; a 7 m wide fault core and two 40 m wide damage zones. We placed both wells in the western damage zone. We assigned
a porosity of 14% to the entire model domain. The permeability of the host model volume, representing the host rock, was set
at $10^{-13}$ m$^2$. In the first sub-scenario, the fault was modelled as a single structure, i.e., only as a damage zone, with a
permeability increased by 2 orders of magnitude compared to the host rock. In the second sub-scenario, we simulated a fault
that consists of two damage zones and a fault core. The permeability of the damage zones was set to be 2 orders of magnitude
higher ($10^{-11}$ m$^2$) than the host rock ($10^{-13}$ m$^2$) and the permeability of the fault core was set to be 5 orders of magnitude lower
($10^{-18}$ m$^2$) than the host rock. Both sub-scenarios were modelled without and with fracture anisotropy within the damage zones.
In the latter case, we introduced fracture anisotropy parallel to the fault surface, with permeability 1 order of magnitude higher
($10^{-10}$ m$^2$), compared to all other directions.

In this fourth scenario, the orientations of the BHGs were 0°, 90°, 180°, and 270°, with simulated magnitudes of 0, 1, 5,
and 20 mm m$^{-1}$.

In total, we modelled 1027 experiments with increasing geological complexity. Note that since the range of permeabilities
analysed was large, we kept other parameters, including the fluid injection rate, constant, to allow different models to be
comparable.

**Kommentiert [A38]:** AC – point 3

Corrected a typo

**Kommentiert [A39]:** R2 – point 2.12

**Kommentiert [A40]:** R3 – point 2.17
No changes made

[revised manuscript text omitted]